# CA²P: Cache-Augmented Code-as-Policies for Open-Domain Embodied Tasks

## Abstract

Embodied agents deployed in open-domain environments must continuously handle unpredictable tasks beyond predefined action policies. Such tasks are often given as natural language instructions, and recent progress in code-writing large language models (CodeLLMs) has inspired the Code-as-Policies (CaP) paradigm, where instructions are translated into executable control code when issued. However, generating full code from scratch for each instruction incurs high latency and inconsistency, limiting CaP's practicality in real-world, time-sensitive scenarios. To address these limitations, we present CA²P, a Cache-Augmented Code-as-Policies framework that improves CodeLLM-based robotic programming by introducing function-level key-value (KV) caching, a repurposed and extended form of the native KV caching mechanism tailored for function reuse, together with cache-augmented code policy synthesis. CA²P decomposes previously generated and validated code policies and stores them as function-level KV caches, supporting efficient compositional programming, where new policies are synthesized by invoking cached functions directly through their KV states. Furthermore, by revisiting and editing cached functions within their KV states, CA²P provides cache-refactoring, thereby enabling efficient synthesis of task-specific code policies without the need for full regeneration. Evaluated on ALFRED, TEACh, and RLBench benchmarks together with real-world robot manipulation, CA²P achieves the best trade-off between robustness and latency, with $19.80\%$ higher task success rate and $2.91\times$ faster policy synthesis than the CaP baseline.

## 1 Introduction

Embodied agents, including mobile and manipulator robots, operate in ever-changing environments and continuously encounter novel tasks. To act effectively, agents must ground natural language instructions in perception and transform them into action policies that can handle open-ended, dynamic tasks with responsiveness and adaptability. Recent advances in large language models with code-writing capabilities (CodeLLMs) have inspired the paradigm of Code-as-Policies (CaP), in which executable control code is generated from instructions using a predefined set of APIs (Liang et al., 2023; Vemprala et al., 2023; Huang et al., 2023a;b; Burns et al., 2024). By leveraging the generalization capabilities of CodeLLMs, the CaP paradigm complements task-specific policy learning and provides a more flexible means of control. However, its practicality remains limited in real-world deployments. Each new task or environmental change requires the full generation of control code from scratch, often comprising hundreds to thousands of tokens. This results in high inference latency, which undermines the immediacy required for responsive control. Furthermore, even semantically similar instructions can produce divergent control logic, leading to inconsistent behavior.

Promising directions for mitigating these challenges involve memory-based approaches that reuse prior experiences, together with key-value (KV) caching mechanisms that minimize redundant attention computations. However, existing memory-based approaches in embodied agents primarily operate at the text level, offering only limited improvements in computational efficiency. Likewise, prior work on KV caching does not support function-level adaptation and is therefore inadequate for the dynamic and evolving nature of open-domain environments. To this end, we propose CA²P, a Cache-Augmented Code-as-Policies framework that enhances CodeLLM-based robotic programming via function-level KV caching. CA²P repurposes the native KV caching mechanism to enable function-level code reuse, forming the basis for cache-augmented code policy synthesis. Akin to the skill-based

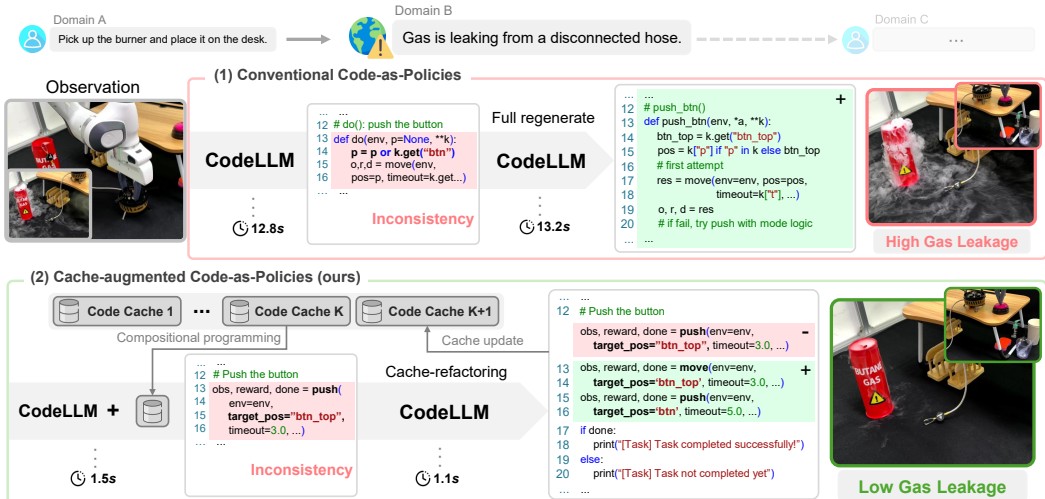

Figure 1: Illustration of $\text{CA}^2\text{P}$ in an open-domain scenario involving gas management. (1) Conventional CaP suffers from high latency; delayed responses cause gas leakage, cascading into further disruptions in subsequent tasks. (2) $\text{CA}^2\text{P}$ enables rapid and reliable code policy synthesis, adapting control code in time to minimize leakage and promptly resume remaining tasks.

paradigm in robotics, where predefined skills, represented as action sequences, are combined and adapted to meet environment-specific constraints and task variations (Nachum et al., 2018; Nyga et al., 2018; Rana et al., 2023), our framework follows the principle of reusing prior building blocks. However, $\text{CA}^2\text{P}$ focuses on efficient policy construction and improved responsiveness by directly operating on KV states, enabling targeted in-place modifications without full regeneration.

Figure 1 illustrates the core distinction between conventional CaP and $\text{CA}^2\text{P}$. While conventional CaP generates full control code from scratch for every new instruction, $\text{CA}^2\text{P}$ leverages cached functions to enable more efficient and reliable code policy synthesis in open-domain environments. To effectively handle unpredictable situations, $\text{CA}^2\text{P}$ performs compositional programming and cache-refactoring, which operate in a complementary manner to support cache-augmented code policy synthesis. From earlier task success, previously generated code policies are decomposed into function-level KV caches and stored in a two-tier code cache, where lightweight references are provided and complete implementations are retained. Building on cached functions, new code policies are rapidly synthesized by composing function calls directly through their KV states, thereby bypassing full token-level generation. On demand, reasoning-based cache-refactoring is performed by editing decomposed sub-policies within KV states, allowing efficient task- and environment-specific adaptation without full regeneration. Consequently, our $\text{CA}^2\text{P}$ framework reduces CaP's overhead and improves its practicality in open-domain environments.

We evaluate $\text{CA}^2\text{P}$ in diverse open-domain scenarios using embodied benchmarks, including AL-FRED (Shridhar et al., 2020), TEACh (Padmakumar et al., 2022), and RLBench (James et al., 2020), as well as in real-world robotic manipulation tasks. Experimental results show that $\text{CA}^2\text{P}$ achieves a superior trade-off between robustness and latency compared to CaP baselines, improving task success rates by $19.80\%$ and accelerating policy synthesis by $2.91\times$ on average (Table 1 and Table 2). These results highlight $\text{CA}^2\text{P}$'s ability to deliver fast, reliable, and consistent policy synthesis, underscoring its practical advantage for embodied agents in open-domain environments.

Our contributions are summarized as follows:

- We present the $\text{CA}^2\text{P}$ framework that improves CodeLLM-based robotic programming through function-level KV caching, achieving greater computational efficiency and behavioral consistency, which in turn strengthens adaptability in open-domain environments.
- We introduce cache-refactoring, a reasoning-based code editing mechanism that enables targeted modifications within cached KV states during policy synthesis, allowing $\text{CA}^2\text{P}$ to adapt to task-specific requirements without full regeneration.

- We show that $\mathrm{CA}^2\mathrm{P}$ achieves the best trade-off between robustness and latency in diverse open-domain scenarios, including real-world robotic manipulation.

## 2 RELATED WORK AND PROBLEM FORMULATION

**Related work.**

Recent advances in embodied control increasingly leverage large language models (LLMs) for task planning (Huang et al., 2022b; Brohan et al., 2023; Song et al., 2023), with a growing shift toward using CodeLLMs to generate executable control code policies (Nijkamp et al., 2022; Chen et al., 2021; Roziere et al., 2023; Hui et al., 2024). In parallel, memory-based approaches explore experience reuse (Wang et al., 2024; Xu et al., 2025; Shinn et al., 2024) for long-horizon reasoning and task adaptation, with emerging efforts integrating memory into CaP (Sarch et al., 2023; 2024; Tziafas & Kasaei, 2024). KV caching mechanisms have also been introduced to accelerate generation (Chan et al., 2025; Jin et al., 2024; Lu et al., 2024) by reusing attention states, with recent extensions supporting dynamic settings through adaptive and modular strategies (Yao et al., 2025; Hu et al., 2024; Gim et al., 2024). Additional advances include infilling methods for fill-in-the-middle (FIM) tasks (Bavarian et al., 2022; Guo et al., 2025; He et al., 2024), which enable the insertion of new code lines into cached sequences. Building on these trends, this work introduces function-level KV caching and efficient code policy synthesis, designed to support responsive embodied control. A detailed discussion of related work is provided in Appendix A.

**Problem formulation.** We formulate the open-domain embodied task as a tuple $(\mathcal{D}, \mathcal{S}, \mathcal{A}, \mathcal{F})$, where $\mathcal{D}$ is the set of domains, $\mathcal{S}$ the state space, $\mathcal{A}$ the action space, and $\mathcal{F}$ the domain mapping function. Due to partial observability (Sutton & Barto, 2018), at each timestep $t$, the agent perceives an observation $o_t \in \mathcal{O}$ that provides incomplete information about the true state $s_t \in \mathcal{S}$. The function $\mathcal{F}(d) = \{\Omega_d, \mathcal{G}_d, \mathcal{P}_d\}$ (Hallak et al., 2015) maps each domain $d$ to its observation function $\Omega_d : \mathcal{S} \times \mathcal{A} \rightarrow \mathcal{O}$, a set of goal states $\mathcal{G}_d \subset \mathcal{S}$, and transition function $\mathcal{P}_d : \mathcal{S} \times \mathcal{A} \rightarrow \mathcal{S}$. The target goal state $g_d \in \mathcal{G}_d$ is defined by a natural language instruction set $\mathcal{T}_d$, where each instruction $\tau_d \sim \mathcal{T}_d$ specifies a corresponding goal condition. In open-domain settings, $\mathcal{D}$ is neither fixed nor known in advance, requiring the agent to continually adapt to unpredictable tasks and ever-changing environments. Instead of modeling the policy as a direct mapping from $(o_t, \tau_d)$ to $a_t$, we employ a CodeLLM $\pi_\theta$ to generate an executable code policy $\pi_{\mathrm{code}}$ that specifies this mapping. Our objective is to optimize $\pi_\theta$ such that the generated $\pi_{\mathrm{code}}$ maximizes task success rate (SR) while minimizing policy synthesis latency (PSL), with code similarity (CSIM) as a regularizer for consistent behavior across tasks:

$$\pi_\theta^* = \arg\max_{\pi_\theta} \mathbb{E}_{d \sim \mathcal{D}} \mathbb{E}_{\pi_{\mathrm{code}} \sim \pi_\theta(\cdot | \tau_d, o_t)} \Big[ \mathrm{SR}(\mathrm{Exec}(\pi_{\mathrm{code}}), g_d) - \eta \cdot \mathrm{PSL}(\pi_\theta) + \mu \cdot \mathrm{CSIM}(\pi_\theta) \Big] \quad (1)$$

where Exec denotes the trace of program execution in domain $d$, and $\eta$ and $\mu$ are weighting factors.

## 3 $\mathrm{CA}^2\mathrm{P}$: CACHE-AUGMENTED CODE-AS-POLICIES

As illustrated in Figure 2, $\mathrm{CA}^2\mathrm{P}$ is built on two core mechanisms: (i) function-level KV caching and (ii) cache-augmented code policy synthesis. In (i) *function-level KV caching*, previously generated and validated code policies are decomposed into callable functions and stored in a two-tier code cache, with each tier maintaining both textual and KV representations. The code cache organizes function-level entries into an interface tier and an implementation tier, facilitating lightweight code generation and in-place editing during code policy synthesis. To ensure reusability across open-domain tasks, our framework dynamically manages cached functions not only based on recent usage patterns but also through a semantic-aware strategy that accounts for functional diversity. In (ii) *cache-augmented code policy synthesis*, new policies are synthesized through efficient compositional programming and responsive cache-refactoring. Given a task instruction, new code policies are constructed by either composing cached functions via their KV states or revisiting them for in-place editing. The former enables efficient assembly without full regeneration, while the latter allows targeted revisions such as adjusting values, modifying control flow, or correcting execution errors for rapid task-specific adaptation. These mechanisms significantly reduce the overhead of conventional CaP pipelines, allowing fast and reliable code policy synthesis in real-world, open-domain environments.

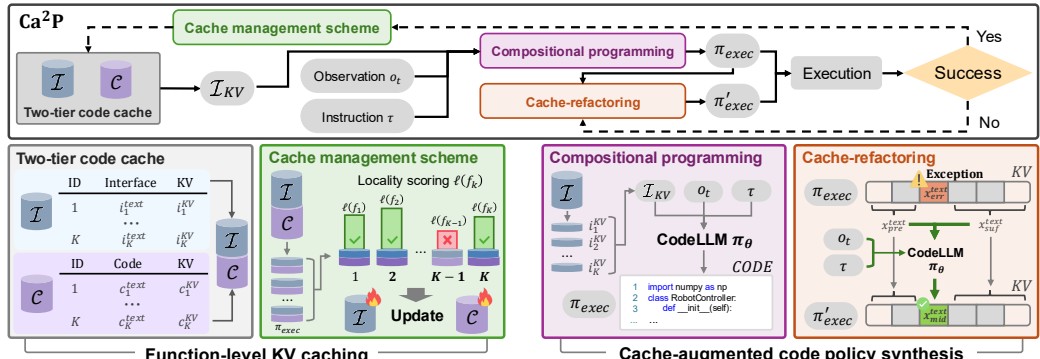

Figure 2: Overall architecture of CA²P. **Top:** End-to-end robotic programming workflow. **Bottom:** Process of function-level KV caching and cache-augmented code policy synthesis.

### 3.1 FUNCTION-LEVEL KV CACHING

As illustrated in Figure 2, we implement a two-tier hierarchical code cache $\mathcal{H}$ consisting of a Function-Interface tier $\mathcal{I}$ and a Function-Code tier $\mathcal{C}$, both indexed by the function identifier $f_k$.

$$\mathcal{H} = (\mathcal{I}, \mathcal{C}), \quad \mathcal{I} = \{(f_k, i_k^{\text{text}}, i_k^{\text{KV}})\}_{k=1}^K, \quad \mathcal{C} = \{(f_k, c_k^{\text{text}}, c_k^{\text{KV}})\}_{k=1}^K \quad (2)$$

The native KV caching mechanism is restructured into this two-tier hierarchical design, enabling efficient function reuse during inference. $\mathcal{I}$ maintains callable signatures along with semantic metadata abstracting full implementations to the function interfaces with KV states for lightweight cache-augmented code generation. When available, $\mathcal{C}$ preserves the corresponding validated implementations that are directly executable and editable in-place within their KV states. To support continual reuse during runtime, each function entry is scored based on recency and invocation frequency, with consideration of conditional co-occurrence in execution traces and semantic functional diversity.

**Two-tier code cache.** The first tier, Function-Interface $\mathcal{I}$, stores entries containing function interfaces $i_k^{\text{text}}$ in text form, which define the function name and typed parameters, paired with their KV states $i_k^{\text{KV}}$. Each entry is modularized at the function level, allowing its $i_k^{\text{KV}}$ to be reinjected into the CodeLLM $\pi_\theta$ without recomputing cross-attention. This design reduces redundant attention computation for reused interfaces and mitigates the *attention sink* problem when handling multiple cached functions, ensuring that cached interfaces remain independent and position-agnostic (Yao et al., 2025; Hu et al., 2024; Gim et al., 2024). The second tier, Function-Code $\mathcal{C}$, stores validated implementations $c_k^{\text{text}}$ along with their cached KV states $c_k^{\text{KV}}$, also indexed by $f_k$. These entries are excluded from direct injection into $\pi_\theta$ during generation to avoid redundant reasoning over complex implementations. Instead, they are linked at execution time to assemble the final program, while their KV states remain accessible for efficient code policy editing.

**Cache management scheme.** After executing each task, all successfully invoked functions, including newly generated ones, are decomposed into $\mathcal{I}$ and $\mathcal{C}$ entries and stored in $\mathcal{H}$. To manage $\mathcal{H}$ under limited GPU memory, we assign a locality score $\ell(f_k)$ to each function $f_k$, indicating its potential for future reuse. Low-scored entries, particularly $i_k^{\text{KV}}$ in $\mathcal{C}$, are offloaded to DRAM, while high-scored entries are retained on the GPU to support fast revisit during code policy synthesis.

$$\ell(f_k) = (1 - \ell_{\text{curr}}(f_k)) \cdot (\alpha \cdot \ell_{\text{freq}}(f_k) + \beta \cdot \sum_j \ell_{\text{asso}}(f_j \mid f_k) + \gamma \cdot \ell_{\text{sema}}(f_k)) + \ell_{\text{curr}}(f_k) \quad (3)$$

Here, $\ell_{\text{curr}}(f_k) \in \{0, 1\}$ is a binary indicator denoting whether $f_k$ was recently used, enforcing immediate retention. The composite score integrates: (1) usage frequency $\ell_{\text{freq}}$, (2) conditional association $\ell_{\text{asso}}$ with co-invoked functions, and (3) semantic relevance $\ell_{\text{sema}}$ based on perplexity (Jelinek et al., 1977). Each component is normalized to $[0, 1]$ and weighted by coefficients $\alpha$, $\beta$, and $\gamma$ satisfying $\alpha + \beta + \gamma = 1$. This scoring captures both short-term recency and long-term functional utility while promoting semantic diversity of cached functions. It enables the agent to maintain a compact yet adaptive code cache that remains responsive to the unpredictability of open-domain tasks.

## 3.2 Cache-augmented code policy synthesis

As show in Figure 2, leveraging the two-tier code cache $\mathcal{H}$, $\mathrm{CA^2P}$ synthesizes new code policies through two complementary modes: compositional programming and cache-refactoring. In compositional programming, $\mathrm{CA^2P}$ generates a new code policy $\pi_{\mathrm{code}}$ by invoking cached functions via their KV states stored in the Function-Interface tier $\mathcal{I}$. This allows executable policies to be assembled compositionally from previously cached functions. When task-specific revisions are required due to unexpected changes or execution errors, $\mathrm{CA^2P}$ further enters the cache-refactoring mode. It revisits $\pi_{\mathrm{code}}$ along with the corresponding implementations in the Function-Code tier $\mathcal{C}$, and applies targeted modifications directly within their KV states. This mechanism enables efficient task-specific adaptation while avoiding redundant regeneration from scratch.

**Compositional programming.** Given current observation $o_t$, task instruction $\tau$, and cached interface KV states $i_k^{\mathrm{KV}}$ in $\mathcal{I}$, the CodeLLM $\pi_\theta$ generates code policy $\pi_{\mathrm{code}}$.

$$\pi_{\mathrm{code}} \sim \pi_\theta(\cdot \mid o_t, \tau, \mathcal{I}_{\mathrm{KV}}), \quad \mathcal{I}_{\mathrm{KV}} = \{i_k^{\mathrm{KV}} \mid (f_k, i_k^{\mathrm{text}}, i_k^{\mathrm{KV}}) \in \mathcal{I}\} \tag{4}$$

Here, $\mathcal{I}_{\mathrm{KV}}$ denotes the set of interface KV states, which are concatenated and injected into $\pi_\theta$ as additional context during generation. Let $\mathrm{Call}(\pi_{\mathrm{code}})$ denote the set of function identifiers that appear as function calls in $\pi_{\mathrm{code}}$. The final executable program $\pi_{\mathrm{exec}}$ is constructed by linking the corresponding cached implementations from the Function-Code tier $\mathcal{C}$.

$$\pi_{\mathrm{exec}} = \pi_{\mathrm{code}} \| \{c_k^{\mathrm{text}} \mid i_k^{\mathrm{text}} \in \mathrm{Call}(\pi_{\mathrm{code}}), (f_k, i_k^{\mathrm{text}}, i_k^{\mathrm{KV}}) \in \mathcal{I}, (f_k, c_k^{\mathrm{text}}, c_k^{\mathrm{KV}}) \in \mathcal{C}\} \tag{5}$$

Here, $\|$ denotes the concatenation of the generated $\pi_{\mathrm{code}}$ with the corresponding cached function $c_k^{\mathrm{text}}$ from $\mathcal{C}$, prior to execution by the program executor (e.g., a Python interpreter).

**Cache-refactoring.** Cache-refactoring is applied either when exception flags are propagated during compositional programming, indicating potentially faulty code spans, or when an exception with a stack trace is raised during the execution of $\pi_{\mathrm{exec}}$. In both cases, the exception $E$ is either attached as an inline comment or explicitly provided, and the target span within $\pi_{\mathrm{exec}}$ is localized. Then, $\pi_{\mathrm{exec}}$ is split into three parts: $\pi_{\mathrm{exec}} = [x_{\mathrm{pre}}^{\mathrm{text}} \| x_{\mathrm{err}}^{\mathrm{text}} \| x_{\mathrm{suf}}^{\mathrm{text}}]$, where $x_{\mathrm{pre}}^{\mathrm{text}}$ and $x_{\mathrm{suf}}^{\mathrm{text}}$ denote the preserved prefix and suffix, and $x_{\mathrm{err}}^{\mathrm{text}}$ corresponds to the erroneous span responsible for $E$. cache-refactoring is then formulated as an FIM task (Bavarian et al., 2022), guided by Chain-of-Thought (CoT) (Wei et al., 2022) that identifies the root cause of $E$ and generates a corrective patch for $x_{\mathrm{err}}^{\mathrm{text}}$. The correction is conducted directly within the KV states, based on the preserved KV contexts $x_{\mathrm{pre}}^{\mathrm{KV}}$ and $x_{\mathrm{suf}}^{\mathrm{KV}}$. Since KV states are generated token-by-token by $\pi_\theta$ during code generation, the alignment between textual spans $x_{\mathrm{pre}}^{\mathrm{text}}, x_{\mathrm{suf}}^{\mathrm{text}}$ and their corresponding KV states $x_{\mathrm{pre}}^{\mathrm{KV}}, x_{\mathrm{suf}}^{\mathrm{KV}}$ can be directly determined without decoding, as positional correspondence is inherently preserved.

$$\pi'_{\mathrm{exec}} = [x_{\mathrm{pre}}^{\mathrm{text}} \| x_{\mathrm{mid}}^{\mathrm{text}} = \pi_\theta(x_{\mathrm{pre}}^{\mathrm{KV}}, x_{\mathrm{suf}}^{\mathrm{KV}}, \mathrm{CoT}(\cdot, E)) \| x_{\mathrm{suf}}^{\mathrm{text}}] \tag{6}$$

Here, $\pi_\theta$ generates a new middle span $x_{\mathrm{mid}}^{\mathrm{text}}$ to bridge the prefix and suffix, resulting in the updated executable program $\pi'_{\mathrm{exec}}$. The CoT trace is used during the refactoring process and discarded before execution, ensuring that only the corrected code is executed. The *generate* method corresponds to compositional programming, and the *edit* method corresponds to cache-refactoring.

# 4 Evaluation

## 4.1 Experimental setting

**Environments.** We evaluate $\mathrm{CA^2P}$ on widely used embodied benchmarks, including AL-FRED (Shridhar et al., 2020), TEACh (Padmakumar et al., 2022), and RLBench (James et al., 2020). To assess its practical applicability, we also conduct real-world robotic manipulation experiments. To reflect the complexity of open-domain deployment, we construct 3 long-horizon evaluation scenarios for each benchmark, characterized by increasing environmental dynamics and uncertainty. In the first scenario, *Open-Composition*, tasks follow a curriculum-style progression with increasing difficulty, where later tasks are constructed by composing earlier ones into more complex forms. The second scenario, *Open-Perturbation*, introduces observation-level dynamics, where object states (e.g., open vs. closed) may change unpredictably during execution. The third scenario, *Open-Evolution*, further increases complexity via unpredictable variations in both observations and goal conditions.

---

**Algorithm 1** Procedure of $\text{CA}^2\text{P}$: (A) task execution loop and (B) built-in methods

---

**(A) Task Execution Loop**
Agent $\text{CA}^2\text{P}$; Environment $env$

1: $scenario\_done \leftarrow$ False
2: **while** not $scenario\_done$ **do**
3:    $t \leftarrow 0; (o_t, \tau) \leftarrow env.\text{reset}()$
4:    $\pi_{\text{exec}} \leftarrow \text{CA}^2\text{P}.\text{generate}(o_t, \tau)$
5:    $[a_0, a_1, \dots] \leftarrow \text{Exec}(\pi_{\text{exec}})$
6:    info $\leftarrow \{\text{``is\_done''} : \text{False}\}$
7:    **while** not info[``is\_done''] **do**
8:      **try**:
9:        $(o_{t+1}, \text{info}) \leftarrow env.\text{step}(a_t)$
10:      **except** Exception as $E$:
11:        $\pi_{\text{exec}} \leftarrow \text{CA}^2\text{P}.\text{edit}(o_{t+1}, \tau, E, \pi_{\text{exec}})$
12:        $[a_t, a_{t+1}, \dots] \leftarrow \text{Exec}(\pi_{\text{exec}})$
13:        **continue**
14:      $o_t \leftarrow o_{t+1}; t \leftarrow t + 1$
15:    **end while**
16:    **if** info[``is\_success''] **then**
17:      $\text{CA}^2\text{P}.\text{update}(\pi_{\text{exec}})$
18:    **end if**
19:    $scenario\_done \leftarrow$ info[``scenario\_done'']
20: **end while**

**(B) $\text{CA}^2\text{P}$ Built-in Methods**
Two-tier code cache $\mathcal{H} = (\mathcal{I}, \mathcal{C})$
**def** generate(self, $o_t, \tau$):
   # compositional programming
1: Generate $\pi_{\text{code}}$ using Eq.(4)
2: Get $\pi_{\text{exec}}$ using Eq.(5)
3: $\pi_{\text{exec}} \leftarrow \text{self.edit}(o_t, \tau, E, \pi_{\text{exec}})$ **if** $E$ in $\pi_{\text{exec}}$ **else** $\pi_{\text{exec}}$
4: **return** $\pi_{\text{exec}}$

**def** edit(self, $o_t, \tau, E, \pi_{\text{exec}}$):
   # cache-refactoring
1: Split $\pi_{\text{exec}} = [x_{\text{pre}}^{\text{text}} \| x_{\text{err}}^{\text{text}} \| x_{\text{suf}}^{\text{text}}]$
2: $x_{\text{mid}}^{\text{text}} \leftarrow \pi_\theta(x_{\text{pre}}^{\text{KV}}, x_{\text{suf}}^{\text{KV}}, \text{CoT}(o_t, \tau, E))$
3: $\pi'_{\text{exec}} \leftarrow [x_{\text{pre}}^{\text{text}} \| x_{\text{mid}}^{\text{text}} \| x_{\text{suf}}^{\text{text}}]$    *cf. Eq.(6)*
4: **return** $\pi'_{\text{exec}}$

**def** update(self, $\pi_{\text{exec}}$):
   # cache management scheme
1: Decompose $\pi_{\text{exec}}$ into $\mathcal{I}$- and $\mathcal{C}$-entries
2: Assign the locallity score $\ell(f_k)$ using Eq.(3)
3: Update $\mathcal{H}$ under limited GPU memory
4: **return** 0

---

**Tasks.** In each scenario, agents encounter a continual stream of tasks, categorized into simple tasks that require shallow logic and complex tasks that demand subtask composition or adaptive reasoning. For example, in ALFRED, tasks such as *move an item* are simple, while tasks such as *put items in correct places* require reasoning on multiple spatial and temporal dependencies. In the real-world setting, we use complex tasks such as *organizing an office desk* and *preparing a cooking workstation*, which involve manipulating multiple objects with varying attributes and spatial constraints.

**Baselines.** We compare $\text{CA}^2\text{P}$ against nine competitive baselines, categorized into three groups based on their strategy for leveraging the CodeLLM during code policy synthesis. (1) General CodeLLM-based programming methods such as **CaP** (Liang et al., 2023) and **SCoT** (Li et al., 2025a) directly generate code policies from natural language instructions. (2) Memory-based approaches such as **HELPER** (Sarch et al., 2023), **LRLL** (Tziafas & Kasaei, 2024), and **PromptBook** (Arenas et al., 2024) retrieve textual programming artifacts as additional inputs to the CodeLLM. (3) KV caching methods such as **CAG** (Chan et al., 2025), **RAGCache** (Jin et al., 2024), **PromptCache** (Gim et al., 2024), and **EPIC** (Hu et al., 2024) accelerate inference by reusing cached attention states. To ensure a fair comparison, all baselines are given the same prior information derived from external oracle policies for basic functions that are entirely unrelated to any task in the benchmarks. This prior information is supplied in the format appropriate for each method: prompt-based baselines receive it as prompts, memory-based baselines receive it as memory entries, and both the KV cache baselines and $\text{CA}^2\text{P}$ receive it in KV cache form to establish an equivalent initialization.

**Metrics.** We evaluate performance across four complementary dimensions: (1) Task accuracy, measured by task success rate (SR) and goal condition success rate (GC) (Shridhar et al., 2020). (2) Computational efficiency, measured by policy synthesis latency (PSL) in seconds and number of generated tokens (NGT). (3) Behavioral consistency, measured by code similarity (CSIM) (Chon et al., 2024), forward transfer (FWT), and backward transfer (BWT) (Lopez-Paz & Ranzato, 2017). (4) Memory efficiency, measured by cache hit rate (HR) and GPU memory usage (MU). We also report a composite metric (Rank) that captures the trade-off between SR and PSL.

**Implementation.** All experiments are conducted in Python 3.9 using Qwen2.5-Coder-14B (Hui et al., 2024) as the default CodeLLM, accessed via HuggingFace (Wolf et al., 2019), unless otherwise specified. For fair comparison, all baselines use the same CodeLLM configuration and run on off-the-shelf NVIDIA RTX 4090 GPUs. Experimental details are provided in Appendix B.

Table 1: Performance on open-domain embodied tasks across simulation benchmarks. Results are averaged over three seeds, with standard deviations indicating consistency across runs.

| **ALFRED** | *Open-Composition* | | | *Open-Perturbation* | | | *Open-Evolution* | | |
|---|---|---|---|---|---|---|---|---|---|
| Methods | SR (↑) | PSL (↓) | Rank (↓) | SR (↑) | PSL (↓) | Rank (↓) | SR (↑) | PSL (↓) | Rank (↓) |
| CaP | 46.48±1.87 | 16.21±1.03 | 8 (0.53) | 42.23±0.12 | 16.15±0.23 | 8 (0.53) | 37.22±1.72 | 16.28±1.20 | 8 (0.50) |
| SCoT | 47.93±0.87 | 30.02±3.38 | 10 (0.27) | 43.46±1.06 | 32.39±1.12 | 10 (0.23) | 39.00±0.98 | 32.04±2.10 | 10 (0.22) |
| HELPER | 53.83±0.39 | 16.82±1.20 | 6 (0.55) | 52.49±0.24 | 15.94±0.69 | 3 (0.58) | 47.75±1.08 | 16.42±0.92 | 3 (0.55) |
| LRLL | 56.28±3.20 | 16.02±2.02 | 2 (0.58) | 54.70±0.87 | 16.40±1.24 | 2 (0.59) | 51.60±0.30 | 16.63±1.03 | 2 (0.57) |
| PromptBook | 53.94±3.96 | 31.34±1.82 | 9 (0.27) | 52.63±1.09 | 33.29±1.52 | 9 (0.26) | 50.49±0.51 | 33.49±2.07 | 9 (0.25) |
| CAG | 38.24±0.04 | 12.12±0.62 | 3 (0.57) | 36.03±0.32 | 12.38±0.92 | 4 (0.57) | 32.18±1.12 | 12.39±0.92 | 4 (0.55) |
| RAGCache | 39.29±1.70 | 13.48±0.93 | 7 (0.55) | 36.33±0.02 | 13.15±0.12 | 6 (0.56) | 33.83±1.53 | 13.02±1.15 | 5 (0.55) |
| EPIC | 43.38±1.02 | 14.02±1.04 | 5 (0.56) | 37.08±1.02 | 13.74±0.69 | 7 (0.55) | 34.23±0.23 | 13.49±1.14 | 7 (0.54) |
| PromptCache | 40.37±0.37 | 12.82±0.86 | 4 (0.56) | 36.92±2.22 | 12.93±1.48 | 5 (0.56) | 34.14±0.82 | 13.18±1.15 | 6 (0.54) |
| CA$^2$P (ours) | **61.58**±1.86 | **5.82**±0.57 | **1 (0.81)** | **57.23**±0.23 | **6.32**±0.51 | **1 (0.79)** | **55.89**±0.85 | **6.29**±1.25 | **1 (0.78)** |

| **TEACh** | *Open-Composition* | | | *Open-Perturbation* | | | *Open-Evolution* | | |
|---|---|---|---|---|---|---|---|---|---|
| Methods | SR (↑) | PSL (↓) | Rank (↓) | SR (↑) | PSL (↓) | Rank (↓) | SR (↑) | PSL (↓) | Rank (↓) |
| CaP | 43.92±0.81 | 17.29±0.73 | 5 (0.53) | 42.21±0.28 | 17.23±0.39 | 5 (0.51) | 38.07±0.12 | 17.03±1.20 | 4 (0.52) |
| SCoT | 44.03±1.47 | 33.00±0.91 | 9 (0.28) | 42.49±1.02 | 33.94±1.92 | 9 (0.23) | 37.91±0.41 | 35.03±1.58 | 9 (0.22) |
| HELPER | 47.14±0.63 | 17.31±1.08 | 3 (0.55) | 44.05±0.29 | 17.39±1.02 | 4 (0.52) | 40.00±0.38 | 17.48±0.71 | 3 (0.52) |
| LRLL | 48.99±1.73 | 17.82±0.82 | 2 (0.55) | 46.78±0.91 | 17.93±1.22 | 3 (0.52) | 41.85±0.02 | 18.03±0.39 | 2 (0.52) |
| PromptBook | 47.08±2.24 | 36.92±1.47 | 10 (0.24) | 44.85±1.00 | 35.04±1.24 | 10 (0.22) | 41.07±0.32 | 36.82±0.49 | 10 (0.21) |
| CAG | 35.03±2.01 | 14.23±1.28 | 4 (0.53) | 33.89±0.92 | 14.02±0.24 | 2 (0.53) | 30.58±0.33 | 14.82±0.11 | 5 (0.52) |
| RAGCache | 35.23±0.05 | 14.97±0.15 | 7 (0.52) | 34.08±0.04 | 15.51±0.48 | 7 (0.50) | 29.45±1.23 | 16.07±0.29 | 8 (0.50) |
| EPIC | 36.39±0.64 | 15.43±0.44 | 8 (0.52) | 34.39±0.51 | 15.66±0.48 | 8 (0.50) | 33.92±1.52 | 17.18±0.54 | 7 (0.50) |
| PromptCache | 36.15±0.86 | 14.88±0.29 | 6 (0.53) | 33.12±0.12 | 14.94±1.31 | 6 (0.51) | 31.07±1.49 | 16.32±0.24 | 6 (0.50) |
| CA$^2$P (ours) | **58.59**±0.39 | **5.23**±0.09 | **1 (0.79)** | **56.26**±1.09 | **5.61**±1.83 | **1 (0.78)** | **52.93**±0.62 | **7.03**±0.82 | **1 (0.76)** |

| **RLBench** | *Open-Composition* | | | *Open-Perturbation* | | | *Open-Evolution* | | |
|---|---|---|---|---|---|---|---|---|---|
| Methods | SR (↑) | PSL (↓) | Rank (↓) | SR (↑) | PSL (↓) | Rank (↓) | SR (↑) | PSL (↓) | Rank (↓) |
| CaP | 31.23±3.32 | 11.39±0.53 | 8 (0.45) | 30.46±0.02 | 11.23±0.93 | 7 (0.46) | 26.84±1.20 | 11.23±0.24 | 7 (0.48) |
| SCoT | 32.32±2.22 | 22.03±2.03 | 9 (0.20) | 30.32±0.28 | 21.37±1.84 | 9 (0.22) | 27.11±0.37 | 22.83±1.42 | 9 (0.21) |
| HELPER | 35.42±1.32 | 11.48±0.23 | 7 (0.47) | 34.84±0.03 | 11.38±0.06 | 6 (0.48) | 31.12±1.08 | 12.16±1.42 | 8 (0.48) |
| LRLL | 36.11±0.00 | 10.83±0.82 | 4 (0.49) | 33.82±0.02 | 12.01±0.11 | 8 (0.46) | 32.45±0.94 | 12.17±2.01 | 6 (0.48) |
| PromptBook | 35.93±1.12 | 23.49±0.63 | 10 (0.18) | 33.03±0.67 | 24.03±3.89 | 10 (0.17) | 31.65±3.32 | 25.93±2.09 | 10 (0.16) |
| CAG | 26.37±2.03 | 8.53±0.83 | 2 (0.50) | 24.89±1.21 | 8.39±0.37 | 2 (0.50) | 22.82±0.73 | 8.08±1.91 | 2 (0.53) |
| RAGCache | 27.26±1.08 | 9.09±0.63 | 6 (0.49) | 27.03±1.04 | 9.28±0.66 | 5 (0.49) | 23.79±1.20 | 9.02±0.18 | 5 (0.51) |
| EPIC | 29.24±2.88 | 9.48±1.20 | 5 (0.49) | 29.04±0.83 | 9.48±0.32 | 4 (0.50) | 25.85±1.25 | 9.28±0.47 | 3 (0.52) |
| PromptCache | 28.33±1.73 | 8.99±0.39 | 3 (0.50) | 27.41±0.81 | 8.97±0.35 | 3 (0.50) | 25.03±0.85 | 9.12±0.65 | 4 (0.52) |
| CA$^2$P (ours) | **45.91**±4.37 | **3.24**±0.65 | **1 (0.73)** | **42.82**±2.64 | **3.47**±0.16 | **1 (0.71)** | **33.98**±1.27 | **4.39**±0.43 | **1 (0.67)** |

## 4.2 MAIN RESULT

**Benchmarks.** Table 1 presents a comparison between CA$^2$P and 9 competitive baselines for code policy synthesis in open-domain tasks, evaluated on 3 benchmarks, each with 3 open-domain scenarios. Across all benchmarks, CA$^2$P consistently outperforms the baselines, achieving the best trade-off between SR and PSL. On average, it achieves a $19.80\%$ higher SR and a $2.91\times$ faster in PSL compared to CaP. On ALFRED, CA$^2$P achieves a $4.04\%$ higher SR and a $2.67\times$ reduction in PSL compared to HELPER, the second-best baseline in Rank. On TEACh, which involves interactive tasks and frequent instruction changes, CA$^2$P achieves a $14.35\%$ higher SR and a $2.82\times$ reduction in PSL compared to LRLL. These results highlight the effectiveness of our cache-refactoring mechanism in enabling both accurate and efficient policy synthesis. Compared to regeneration-based CaP baselines, cache-refactoring synthesizes more reliable code policies, leading to increased retention of validated functions in the code cache. This iterative accumulation promotes better adaptation to future tasks through improved function reuse. On RLBench, which emphasizes low-level manipulation skills and fine-grained control, CA$^2$P achieves a $16.21\%$ higher SR and a $2.30\times$ reduction in PSL compared to CAG. Unlike ALFRED and TEACh, RLBench tasks often involve loop structures in the generated code policies, leading to generally lower PSL across all methods. The requirement for precise control makes RLBench particularly suitable for showcasing the strength of CA$^2$P's cache-refactoring, which enables fine-grained and efficient code-level adaptation. Specifically, in *Open-Composition*, CA$^2$P leverages compositional programming followed by cache-refactoring to adjust function parameters or update specific variable values, resulting in improved SR despite increasing task complexity. In *Open-Perturbation*, cache-refactoring enables CA$^2$P to remain robust under changing object states by efficiently responding to API-level exceptions triggered when planned actions fail to produce effects, maintaining higher SR and faster PSL. In *Open-Evolution*, CA$^2$P adapts quickly to dynamic changes

Figure 3: Qualitative examples of open-domain embodied task execution in real-world settings

in both observations and goals by replacing initially selected functions with alternatives better suited to the current environmental context, outperforming baselines in both SR and PSL.

**Robot tests.** In Table 2, we test CA$^2$P on real-world manipulation with a 7-DoF Franka Emika Research 3, evaluating transferability of the code cache from simulation to the physical robot. To build the code cache, the agent first solves simpler tasks in RLBench, then transfers to open-domain tasks involving mechanical failures and safety hazards. These include scenarios such as an office desk with diverse objects and complex spatial relations not seen in RLBench, and a cooking workstation where gas hoses disconnect mid-operation, requiring the agent to synthesize and deploy corrective code under time pressure. In both the office desk and cooking workstation scenarios, CA$^2$P consistently outperforms all baselines, achieving up to $37.04\%$ higher SR than LRLL, reducing PSL by $2.88\times$ than RAGCache, and showing more reliable performance across 9 trials.

Figure 3 complements Table 2 by demonstrating CA$^2$P's practicality through real-world experimental examples of compositional programming and cache-refactoring. CA$^2$P synthesizes new code policies via compositional programming by reusing cached functions within a few seconds. For task- or environment-specific adaptation, it performs cache-refactoring by revisiting and editing the corresponding KV states in place. For example, when an exception is triggered due to a key error-caused by misinterpreting the output of the perception API and selecting an inconsistent object-region value (e.g., drawer_high) as a parameter-CA$^2$P replaces the previously selected function with a more appropriate one aligned with the correct parameter (e.g., drawer_top). As a result, the agent rapidly grounds its code policy in the current context, enabling reliable task completion.

## 4.3 ANALYSIS AND ABLATION

**Analysis on code cache warm-up.** Figure 4, including subfigures (a)–(d), shows the bootstrapping performance of the code cache over a stream of 40 open-domain tasks. This experiment is designed as an independent warm-up analysis to observe how each method behaves when starting from an empty cache and how many tasks are required before the performance becomes stable. To ensure fairness, all methods-including those that use KV caching-begin with an empty cache so that the warm-up characteristics can be directly compared. In (a), CA$^2$P consistently improves and maintains a higher SR, while (b) shows a rapid reduction in PSL compared to other baselines. It also maintains

Table 2: Performance on open-domain embodied tasks in real-world robotic manipulation

| Real-world | Office Desk Rearrangement | | | Cooking Workstation Preparation | | |
|---|---|---|---|---|---|---|
| Methods | SR ($\uparrow$) | PSL ($\downarrow$) | Rank ($\downarrow$) | SR ($\uparrow$) | PSL ($\downarrow$) | Rank ($\downarrow$) |
| CaP | $33.33\pm0.00$ | $11.94\pm1.17$ | 4 (0.19) | $51.85\pm6.42$ | $13.07\pm1.09$ | 4 (0.26) |
| LRLL | $55.56\pm11.11$ | $12.31\pm0.74$ | 3 (0.28) | $55.56\pm0.00$ | $12.65\pm1.14$ | 3 (0.30) |
| RAGCache | $44.44\pm22.22$ | $9.08\pm0.74$ | 2 (0.41) | $37.04\pm6.42$ | $9.63\pm1.01$ | 2 (0.35) |
| CA$^2$P (ours) | $\mathbf{77.78}\pm11.11$ | $\mathbf{3.80}\pm0.31$ | **1 (0.89)** | $\mathbf{81.48}\pm12.83$ | $\mathbf{2.85}\pm0.54$ | **1 (0.91)** |

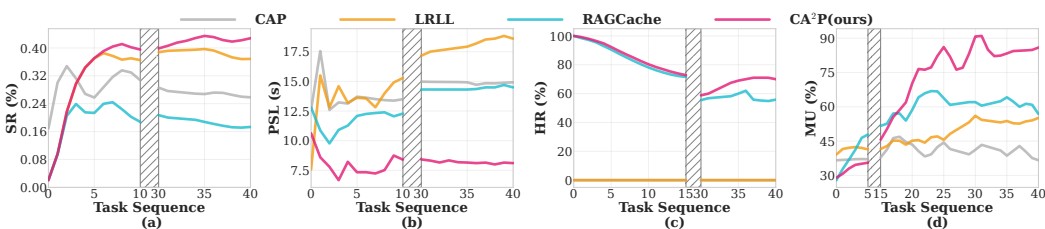

Figure 4: Analysis on code cache warm-up, with SR, PSL, HR, and MU over 40 tasks

a higher HR than RAGCache, as shown in (c), and achieves more efficient MU, as shown in (d), demonstrating the effectiveness of our code cache management strategy.

Table 3: Analysis on behavioral and code-level consistency across continual task phases

| RLBench | *Initial* (after task IDs 1–9) | | | | *Middle* (after task IDs 10–18) | | | | *Final* (after task IDs 19–27) | | | |
|---|---|---|---|---|---|---|---|---|---|---|---|---|
| Method | SR (↑) | FWT | BWT | CSIM | SR (↑) | FWT | BWT | CSIM | SR (↑) | FWT | BWT | CSIM |
| CaP | 33.33 | 0.00 | 0.00 | 57.87±28.03 | 27.78 | 0.00 | 0.00 | 69.06±28.93 | 37.33 | – | 0.00 | 63.44±27.81 |
| LRLL | 33.33 | -6.25 | 5.56 | 72.17±14.84 | 33.33 | 0.00 | 5.56 | 52.87±21.88 | 40.00 | – | 4.00 | 48.25±24.98 |
| RAGCache | 33.33 | 0.00 | 0.00 | 34.12±1.36 | 33.33 | 0.00 | 0.00 | 27.77±8.49 | 36.44 | – | -2.00 | 35.74±13.69 |
| CA$^2$P (ours) | **44.44** | 2.08 | 0.00 | 65.63±5.03 | **38.89** | 4.76 | 0.00 | 57.58±11.87 | **46.00** | – | 2.00 | 55.43±8.98 |

**Analysis on behavior consistency.** Table 3 shows snapshot evaluations of the code cache at the *Initial*, *Middle*, and *Final* phases during a continual task stream of 25 tasks. At each phase, the entire task set is re-evaluated while preserving the current state of the code cache to assess the behavioral consistency of the accumulated code, reporting CSIM, FWT, BWT, and SR. The results show that CA$^2$P not only improves SR but also maintains consistent code structure across tasks; it achieves positive FWT across phases without forgetting (non-negative BWT). This demonstrates that CA$^2$P supports consistent behavior by transferring function-level knowledge in open-domain tasks.

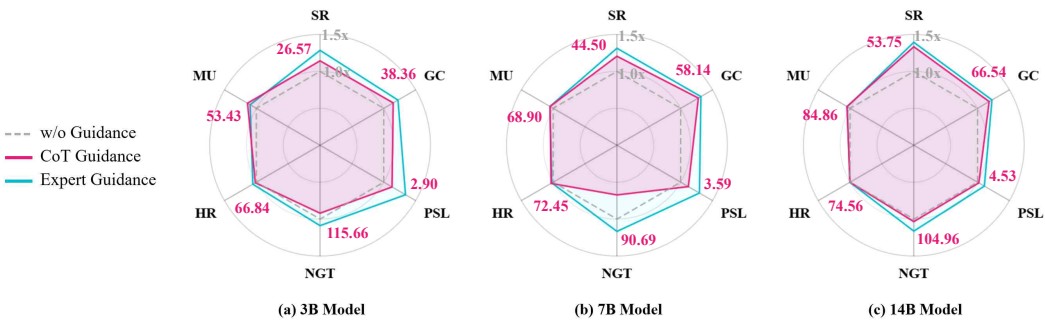

Figure 5: Ablation on model size and cache-refactoring. Evaluation of CodeLLMs with different scales (3B, 7B, 14B), contrasting cache-refactoring against ablated settings.

**Ablation on model size and cache-refactoring.** Figure 5 compares CA$^2$P across different CodeLLM sizes and cache-refactoring configurations. We evaluate the default setting (CoT-guided cache-refactoring) against two ablated variants: one without CoT (using execution feedback only), and one with direct human expert guidance replacing CoT. As model size increases from 3B to 14B, CA$^2$P consistently achieves higher SR (26.57%, 44.50%, 53.75%) and GC (38.36%, 58.14%, 66.54%), while PSL (2.90$s$, 3.59$s$, 4.53$s$) also increases due to the larger model size. MU and HR remain relatively stable, indicating that CA$^2$P effectively balances performance and efficiency across model scales. For the ablation on cache-refactoring, using only execution feedback results in a noticeable drop in SR and GC, with slight reductions in PSL compared to the default setting. When replacing CoT with direct expert guidance, it achieves slightly higher SR and GC due to access to sufficient feedback, although the CoT-based approach remains highly competitive.

Table 4: Ablation on model choice

| Family | Method | CodeLLM | SR (↑) | PSL (↓) | HR (↑) |
|---|---|---|---|---|---|
| QWEN2.5 | CA²P | ✔ | 29.62±0.01 | 2.89±0.41 | 58.63±1.28 |
| | CA²P | ✘ | 26.38±0.23 | 3.12±0.43 | 47.12±3.94 |
| | CaP | ✔ | 23.63±0.16 | 7.94±0.47 | - |
| GEMMA | CA²P | ✔ | 28.83±0.05 | 2.84±0.86 | 52.46±3.28 |
| | CA²P | ✘ | 26.39±0.03 | 3.05±0.13 | 41.67±0.05 |
| | CaP | ✔ | 22.22±0.02 | 8.72±0.49 | - |
| LLAMA2 | CA²P | ✔ | 31.12±1.32 | 3.32±0.09 | 57.91±4.27 |
| | CA²P | ✘ | 25.33±1.07 | 3.46±0.26 | 52.85±1.29 |
| | CaP | ✔ | 17.83±2.46 | 11.48±0.23 | - |
| DEEPSEEK | CA²P | ✔ | 30.23±5.71 | 2.33±0.34 | 67.23±3.28 |
| | CA²P | ✘ | 23.04±3.21 | 2.64±0.86 | 61.34±3.42 |
| | CaP | ✔ | 18.98±0.41 | 13.07±1.13 | - |

Table 5: Ablation on CA²P. 'w/o.' denotes the removal of a component, and '→' indicates replacement with an alternative operation.

| Methods | SR (↑) | PSL (↓) | HR (↑) |
|---|---|---|---|
| CA²P | 45.91±4.37 | 3.24±0.65 | 73.65±2.15 |
| w/o. $\mathcal{I}$ | 38.64±1.55 | 4.12±0.12 | 66.78±3.12 |
| w/o. $\mathcal{C}$ | 34.37±0.52 | 3.33±0.23 | 67.56±3.51 |
| w/o. $\mathcal{H}$ | 34.64±1.98 | 4.25±1.27 | - |
| w/o. $\ell_{sema}$ | 35.02±1.76 | 3.74±0.29 | 48.61±1.96 |
| → Retrieval | 41.92±0.72 | 4.42±0.48 | 43.06±2.45 |
| → Regeneration | 35.77±1.76 | 6.89±0.20 | 71.39±2.83 |

**Ablation on model choice.** Table 4 reports the performance of CA²P applied with four different LLM families, contrasting CodeLLMs with general-purpose LLMs. The results show that CA²P equipped with CodeLLMs consistently outperforms its counterparts based on the LLMs, as well as CaP. On average, it achieves a $4.67\%$ higher SR, a $1.08\times$ reduction in PSL, and an $8.31\%$ higher HR compared to variants using the LLMs, while maintaining a more favorable trade-off than CaP.

**Ablation on CA²P.** Table 5 reports the contribution of each component of CA²P to overall performance. For the code cache structure, removing the Function-Interface tier $\mathcal{I}$ (w/o. $\mathcal{I}$), the Function-Code tier $\mathcal{C}$ (w/o. $\mathcal{C}$), or the entire code cache $\mathcal{H}$ (w/o. $\mathcal{H}$) leads to a substantial drop in SR, confirming that the hierarchical design is crucial for improving robustness and reducing latency.

# 5 CONCLUSION

We presented CA²P, a Cache-Augmented Code-as-Policies framework that improves CodeLLM-based robotic programming through function-level KV caching and cache-augmented code policy synthesis. CA²P stores generated code policies as function-level KV caches, enabling compositional programming through direct function reuse and supporting cache-refactoring via in-place editing. Extensive experiments, including real-world manipulation, demonstrate the robustness and efficiency of CA²P across diverse tasks, highlighting that cache-based synthesis enables agents to rapidly generate and adapt code policies, a key step toward scalable and general-purpose embodied intelligence.

**Limitation and future work.** Despite its strengths, CA²P currently maintains an isolated code cache for each agent, which may limit scalability in dynamic multi-agent cooperative settings. We plan to investigate code cache sharing across agents to enable multi-agent collaboration and to incorporate learning-based cache eviction and compression policies to further optimize cache utilization.

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

# A  RELATED WORK

**Large language models for embodied control.**    In embodied control, there is a growing trend of leveraging the reasoning capabilities of LLMs for task planning without additional finetuning (Huang et al., 2022b; Brohan et al., 2023; Song et al., 2023; Wang et al., 2023; Yao et al., 2023; Liang et al., 2024; Huang et al., 2023c; Zhou et al., 2023; Zhao et al., 2024; Huang et al., 2022a). Building on recent advances in the code-writing capabilities of LLMs (Chen et al., 2021; Nijkamp et al., 2022; Roziere et al., 2023; Hui et al., 2024; Guo et al., 2024; Zhu et al., 2024), the paradigm of LLM-based embodied control has shifted toward generating executable code as control policies (Liang et al., 2023; Huang et al., 2023b;a; Burns et al., 2024; Li et al., 2024; Mu et al., 2024; Vemprala et al., 2023; Singh et al., 2022). Beyond mapping instructions to predefined skills or action primitives, these frameworks prompt CodeLLMs to produce Python-like scripts that directly invoke perception and motor APIs, enabling embodied agents to perform motion-level control. In this work, we extend the CaP framework to improve robotic programming, overcoming the latency and inconsistency that arise from full regeneration of code policies in dynamic environments.

**Memory-based embodied agents.**    Recent work has explored equipping embodied agents with memory to support long-horizon reasoning, task adaptation, and generalization by reusing validated experiences (Xu et al., 2025; Shinn et al., 2024; Li et al., 2025b; Kang et al., 2023; Wang et al., 2024; Kagaya et al., 2024). Such approaches maintain structured buffers of past experiences, such as observations, trajectories, and latent representations, to facilitate more informed decision-making. In the context of CaP, only a few studies have explored leveraging memory to enhance code writing capabilities (Sarch et al., 2023; 2024; Tziafas & Kasaei, 2024). However, these systems mainly focus on task generalization and adaptation, with less emphasis on real-time responsiveness. Unlike prior memory-based approaches that primarily rely on text-level representations for generalization, $\text{CA}^2\text{P}$ introduces function-level KV caching tailored for function reuse, enabling rapid and reliable cache-augmented code policy synthesis.

**Key-value caching.**    Recent works (Chan et al., 2025; Jin et al., 2024; Lu et al., 2024), extending retrieval-augmented generation, incorporate KV caching methods that accelerate the generation process by reusing precomputed attention states at the document level. While effective in reducing computational redundancy, these approaches are not well suited for dynamic or continually changing contexts. To handle dynamic contexts, methods such as CacheBlend (Yao et al., 2025), EPIC (Hu et al., 2024), MPIC (Zhao et al., 2025), and PromptCache (Gim et al., 2024) propose adaptive and modular strategies that compose multiple KV cache segments without being restricted by fixed positional encodings. In parallel, methods such as FIM (Bavarian et al., 2022), PIE (He et al., 2024), and EFIM (Guo et al., 2025) introduce infilling techniques that enable fill-in-the-middle (FIM) by inserting new code lines into cached sequences. Inspired by these advances, $\text{CA}^2\text{P}$ adapts KV caching to embodied control, supporting compositional programming and refactoring to reduce the need for full regeneration and enable efficient, task-specific code policy synthesis.

# B EXPERIMENT SETTING

## B.1 RLBENCH

RLBench (James et al., 2020) is a large-scale benchmark and simulation environment for robotic manipulation, built around the Franka Emika Panda 7-DoF arm. Each task is defined as a goal-conditioned manipulation problem and can be executed in a photorealistic, interactive tabletop environment. As shown in Figure 6, the benchmark provides 100 hand-designed tasks ranging from simple primitives such as reaching and pushing to long-horizon activities such as opening an oven and placing a tray inside. Each task consists of one or more variations, and infinitely many episodes can be generated by randomizing object positions, colors, and shapes. Observations include RGB, depth, and segmentation masks from multiple cameras (stereo shoulder-mounted and wrist-mounted), along with proprioceptive states such as joint angles, velocities, torques, and end-effector pose. This diversity and realism make RLBench a suitable testbed for evaluating agents that require precise low-level control, visuomotor reasoning, and generalization across task variations.

**Environment.** Building on RLBench, we design three evaluation scenarios that reflect the unpredictability and uncertainty of open-domain environments. We evaluate on 20 benchmark tasks, each with variations, resulting in 40 task instances. (1) In *open-composition*, tasks are presented in streams of gradually increasing difficulty, defined by the number of high-level primitive calls and whether the agent must reason about object geometry (e.g., size or spatial layout). For example, an agent may progress from simple move actions, to pick or pick-and-place, and eventually to more complex operations such as multiple pick-and-place or precise insert. This setting evaluates whether skills from earlier tasks can be reused and combined to solve more complex configurations, reflecting the compositional demands of open-domain settings. (2) In *open-perturbation*, task descriptions are perturbed to mimic observation-level noise, such as referring to absent objects or omitting existing ones. For instance, the agent may encounter a description like *"There are two cups on the table"* when only one exists, or *"There is a button on the table"* when several are actually present. This scenario evaluates robustness under description-level uncertainty, requiring the agent to reconcile mismatches between the description and the actual scene while still pursuing the intended task goal. (3) In *open-evolution*, task streams vary unpredictably in both difficulty and observation quality. Unlike *open-composition*, where complexity grows gradually, and *open-perturbation*, where mismatches arise within tasks, this scenario mixes tasks of all difficulty levels while also introducing perturbations at random. An easy instruction may be followed by a difficult one with altered observations, or vice versa. The aim is to test whether agents remain stable and effective across irregular, noisy task sequences, mirroring the uneven and unstructured nature of open-domain environments.

Since RLBench is goal-conditioned by design, we checked both success metrics. In practice, the goal-conditioned success rate differed only marginally from the standard success rate across all of our runs. For this reason, we report only the standard success rate in the tables.

**Task.** RLBench tasks are executed through a library of high-level primitive APIs that we built on top of the simulator's low-level interfaces (see Table 6). For clarity, we group them into three functional categories. These include object-centric manipulation (e.g., pick, place, push), object-referenced motion (e.g., move or align to a quaternion), and robot-internal control (e.g., open or close the gripper). Each primitive follows a structured signature with object- or pose-specific arguments, plus optional parameters for fine-grained control (e.g., offsets or approach axes). This modular design promotes interpretable and reusable skill calls, supporting compositional and generalizable policy construction.

We categorize RLBench tasks by the complexity of the required skill sequence. Easy tasks involve at most two primitive calls, such as *"Reach target"* or *"Pick up cup"*. Medium tasks typically require three or more calls, for example *"Put rubbish in bin"* or *"Unplug charger"*. Hard tasks involve multi-object goals or spatial constraints, such as *"Put all groceries in cupboard"* or *"Take usb out of computer"*. This hierarchy shows how simpler skills can be combined to solve harder tasks, providing a principled way to evaluate $CA^2P$'s generalization to diverse objects and scenes.

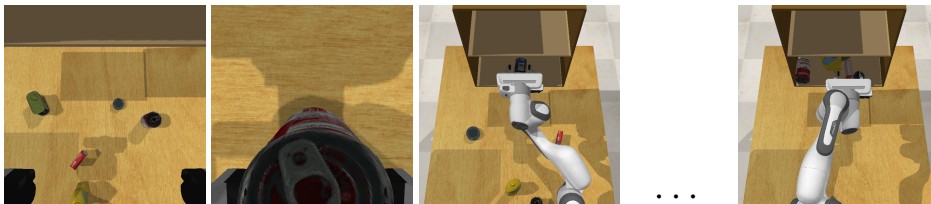

(a) Example of *"Put all groceries in cupboard"* with wrist view and overhead view

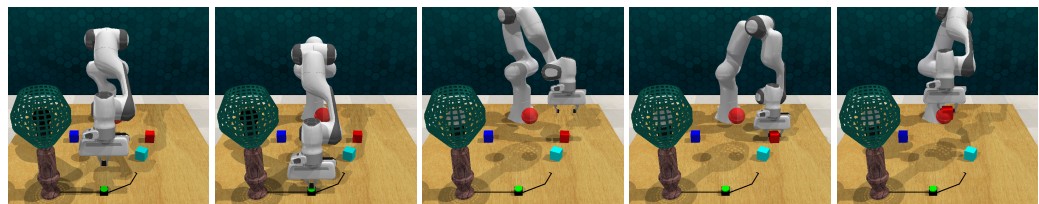

(b) Example of *"Lamp on and pick and lift the red block"* with front view

Figure 6: Environment examples set of RLBench

Table 6: Instructions and executable APIs in RLBench

| | Template | Example |
|---|---|---|
| Instruction | Move | *"Reach the red sphere."* |
| | Pick & Lift | *"Grasp the red cup and lift it."* |
| | Pick & Place | *"Pick up the crackers and place them in the cupboard."* |
| | Pick & Move | *"Take the charger out of the wall."* |
| | Press | *"Press the maroon button."* |
| | Push | *"Slide the block onto the target."* |
| API | Pick [Object] | run_action(pick, 'crackers', offset=[0.0, 0.0, 0.2], approach_axis='z') |
| | Place [Receptacle Object] | run_action(place, 'basket_ball_hoop', offset=[0.0, 0.0, 0.3], approach_dist=0.05) |
| | Move [Object] | run_action(move, 'block', offset=[0, -0.1, 0]) |
| | Push [Object] | run_action(push, 'target_button', offset=[0.0, 0.0, -0.03]) |
| | Align To Quaternion [Object] | run_action(align_to_quaternion, 'usb', align_dir='parallel') |
| | Open Gripper | run_action(open_gripper) |
| | Close Gripper | run_action(close_gripper) |

## B.2 ALFRED

ALFRED (Shridhar et al., 2020) is a large-scale benchmark for embodied AI that integrates vision-and-language navigation with manipulation-based rearrangement tasks. Each task is specified in natural language and requires agents to execute household activities in photorealistic 3D environments. As shown in Figure 7, the benchmark spans 120 indoor scenes (e.g., kitchens, living rooms), containing 58 manipulable object types (e.g., apple, phone) and 26 receptacle types (e.g., fridges, cabinets). From these components, 2685 unique task configurations are generated by combining one of seven instruction templates (e.g., pick-and-place, clean, heat) with scene and object variations. Examples of each template are provided in Figure 7. This diversity makes ALFRED a particularly suitable benchmark for evaluating agents on hierarchical reasoning, multi-step planning, and generalization across varied contexts.

**Environment.** To better emulate open-domain deployment, we design three long-horizon evaluation scenarios on top of ALFRED. (1) In *open-composition*, tasks are organized into a curriculum-like sequence where instruction types gradually increase in difficulty. The sequence begins with simple pick-and-place tasks that only require moving a single object to a receptacle, and then progresses to perception-oriented instructions such as examine. Subsequent stages introduce state-changing operations, including clean-and-place, heat-and-place, and cool-and-place, which require reasoning about object affordances and environment dynamics. Finally, the curriculum culminates in multi-object instructions such as pick-two-objects-and-place, which require hierarchical planning and the compositional reuse of previously synthesized policies. (2) In *open-perturbation*, tasks are subject to observation-level dynamics in which object states may change unpredictably during execution. For example, a cabinet that was initially open may become closed, or a light that was turned on

may switch off. Such perturbations require the agent to detect inconsistencies between expected and observed states, update its internal representation, and adapt its execution accordingly. This scenario evaluates robustness against environmental uncertainty while still preserving the original task goals. (3) In *open-evolution*, complexity is further increased by introducing changes not only in observations but also in goal conditions. For instance, the agent may be instructed to place a heated object on a plate, but midway through execution the target receptacle changes to a shelf. In this case, the agent must revise its plan, resynthesize or repair code policy in real time, and adapt its behavior to the updated objective. This scenario directly measures the agent's capacity for flexible replanning and generalization under evolving task specifications.

**Task.** ALFRED tasks are grounded in a library of action primitives (APIs) that enable interaction with objects and receptacles in the scene. As summarized in Table 7, these APIs cover both basic object manipulation (e.g., pickup, put, open, close) and state-altering operations (e.g., clean, heat, cool). Each API follows a structured signature with arguments specifying the target object and, when relevant receptacle or landmark, thereby allowing natural language instructions to be deterministically mapped to executable symbolic actions. Following the API design principles of prior works (Sarch et al., 2023; 2024), we adopt a similar modular organization to promote compositionality, extensibility, and interpretable policy synthesis.

In our evaluation scenarios, agents face a continual stream of tasks sampled in an open-ended fashion, reflecting the unpredictability of real-world environments. We evaluate $CA^2P$ on a comprehensive set of 812 tasks drawn from ALFRED's instruction templates and instantiated through the API set, and we group them into two categories: simple tasks, solvable with shallow logic (e.g., *"Move a plunger to the cabinet."*), and complex tasks, which require reasoning over multiple spatial and temporal dependencies or involve state changes (e.g., *"Place a cooked apple in the refrigerator."*). This task structure allows us to evaluate $CA^2P$'s ability to support scalable and reusable policy synthesis across diverse and dynamic task sequences.

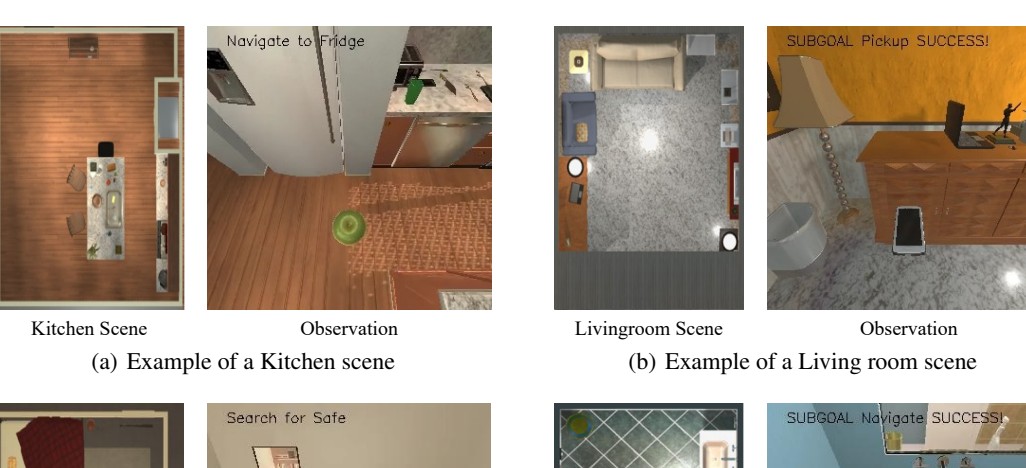

Kitchen Scene      Observation          Livingroom Scene      Observation

(a) Example of a Kitchen scene          (b) Example of a Living room scene

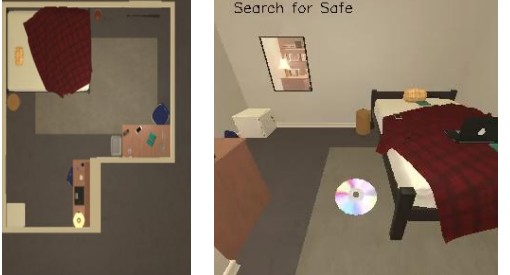

Bedroom Scene      Observation          Bathroom Scene      Observation

(c) Example of a Bedroom scene          (d) Example of a Bathroom scene

Figure 7: Examples of four scene types: kitchen, living room, bedroom, and bathroom

Table 7: Instructions and executable APIs in ALFRED

| | Template | Example |
|---|---|---|
| Instruction | Pick & Place | *"Move a plunger to the cabinet."* |
| | Stack & Place | *"Drop a frying pan with a knife in it into the sink."* |
| | Pick Two & Place | *"Put the two CDs on the desk closest to the window in the safe."* |
| | Clean & Place | *"Place a washed pan on the counter."* |
| | Heat & Place | *"Place a cooked apple in the refrigerator."* |
| | Cool & Place | *"Place a cold tomato slice on the counter."* |
| | Examine & in Light | *"Pick up the cell phone and look at it by the light of the lamp."* |
| API | OpenObject [Object] [Receptacle Object] | `target_lettuce = InteractionObject('Lettuce', landmark = 'Fridge')` `target_fridge.open()` |
| | CloseObject [Object] | `target_microwave = InteractionObject('Microwave')` `target_microwave.close()` |
| | ToggleObject [Object] | `target_lamp = InteractionObject('Lamp', landmark = 'Corner')` `target_lamp.toggle_on()` |
| | SliceObject [Object] [Receptacle Object] | `target_apple = InteractionObject(Apple, landmark = 'CounterTop')` `target_apple.slice()` |
| | GotoLocation [Object] [Receptacle Object] | `target_lettuce = InteractionObject('Lettuce', landmark = 'CounterTop')` `target_lettuce.go_to()` |
| | PickupObject [Object] [Receptacle Object] | `target_lettuce = InteractionObject('Lettuce', landmark = 'CounterTop')` `target_lettuce.pickup()` |
| | PutObject [Object] [Receptacle Object] | `target_lettuce = InteractionObject('Lettuce', landmark = 'CounterTop')` `target_countertop = InteractionObject('CounterTop')` `target_lettuce.place(target_countertop)` |
| | CoolObject [Object] [Receptacle Object] | `target_lettuce = InteractionObject('Lettuce')` `target_lettuce.cool()` |
| | HeatObject [Object] [Receptacle Object] | `target_sink = InteractionObject('Sink')` `target_sink.empty()` |
| | CleanObject [Object] [Receptacle Object] | `target_potato = InteractionObject('Potato', attributes = ['cooked'])` `target_potato.cook()` |

## B.3 TEACH

TEACh (Padmakumar et al., 2022) spans 109 unique scenes across all 30 kitchens and most of the 30 living rooms, bedrooms, and bathrooms in AI2-THOR, comprising 3215 successful human-human gameplay sessions with rich conversational data (~45k utterances averaging 13.67 per session) and long action trajectories (averaging 131.8 Follower actions per session). The dataset covers 12 task families (e.g., put all X on Y, make coffee, prepare breakfast) with 438 parameterized variants defined through a hierarchical Task Definition Language that specifies object state changes (e.g., sliced, toasted, clean) and spatial relations. From these components, TEACh defines three evaluation settings: Execution from Dialogue History (EDH) with 11,176 instances, Trajectory from Dialogue (TfD), and Two-Agent Task Completion (TATC), with seen/unseen splits to assess generalization across novel rooms. This diversity, combined with natural dialogue supervision featuring varied instruction granularity and real-time clarification, makes TEACh particularly suitable for studying language grounding, hierarchical reasoning, long-horizon control with dialogue-based correction, and generalization across interactive household contexts.

**Environment.** For our TEACh evaluation, we use three progressively challenging scenarios that test dialogue-grounded task execution in realistic settings. (1) In *open-composition*, we examine how agents handle increasing dialogue complexity across TEACh's task hierarchy. Simple tasks like *"Water the plant."* involve minimal back-and-forth: the Commander states the goal, and the Follower executes. By contrast, compositional tasks such as *"Prepare breakfast."* require more coordination. This scenario tests whether agents can maintain coherent dialogue over extended, multi-phase interactions. (2) In *open-perturbation*, we evaluate robustness under TEACh's natural human communication patterns. The dataset contains instructional errors (e.g., *"The mug is on the table."* when it is elsewhere), temporal misalignment (utterances arriving out of order), and mid-task corrections (e.g., *"Actually, use the other sink."*). Agents must detect conflicts between instructions and observations, initiate clarification dialogue, and recover from miscommunication, reflecting deployment conditions where instructions are imperfect. (3) In *open-evolution*, we test replanning capabilities under dynamic dialogue. The dataset includes cases where the Commander revises instructions mid-execution, for example changing the goal from *"Put each tissue box on a different table."* to *"Place all the tissue boxes on the same table."*. Agents must interpret such corrections, abandon partially executed plans, and formulate new plans through continued dialogue.

TEACh's action space supports these scenarios through a dual-channel design: physical actions (navigation, manipulation, state changes) executed by the Follower, and free-form text dialogue between the agents. The Commander's special actions (`ProgressCheck`, `SearchObject`) enable task monitoring and environmental queries that support collaboration. This asymmetric information structure, with task knowledge and execution distributed across agents, makes dialogue-based coordination essential in all three scenarios.

**Task.** TEACH tasks are grounded in a library of action primitives (APIs) that enable interaction with objects and receptacles in the scene. As summarized in Table 8, these APIs cover both basic object manipulation (e.g., pickup, place, open, close) and state-altering operations (e.g., clean, heat, slice, toggle). Each API follows a structured signature with arguments specifying the target object and, when relevant, the receptacle or landmark, thereby allowing natural language dialogues and instructions to be deterministically mapped to executable symbolic actions. Following the API design principles of prior works (Sarch et al., 2023; 2024), we adopt a similar modular organization to promote compositionality, extensibility, and interpretable policy synthesis.

In our evaluation scenarios, agents face a continual stream of tasks sampled in an open-ended fashion, reflecting the unpredictability of real-world collaborative environments. We evaluate $CA^2P$ on 621 tasks drawn from TEACH's dialogue-based instruction templates and instantiated through the API set, and we group them into two categories: simple tasks, solvable with shallow logic and minimal dialogue context (e.g., *"Put the tomato on the counter."*), and complex tasks, which require reasoning over extended dialogue histories, handling clarification requests, and managing multi-step dependencies with state changes (e.g., *"Make breakfast - toast the bread and serve it with cleaned lettuce on a plate."*). This task structure allows us to evaluate $CA^2P$'s ability to support scalable and reusable policy synthesis across diverse dialogue-driven interactions and dynamic task sequences.

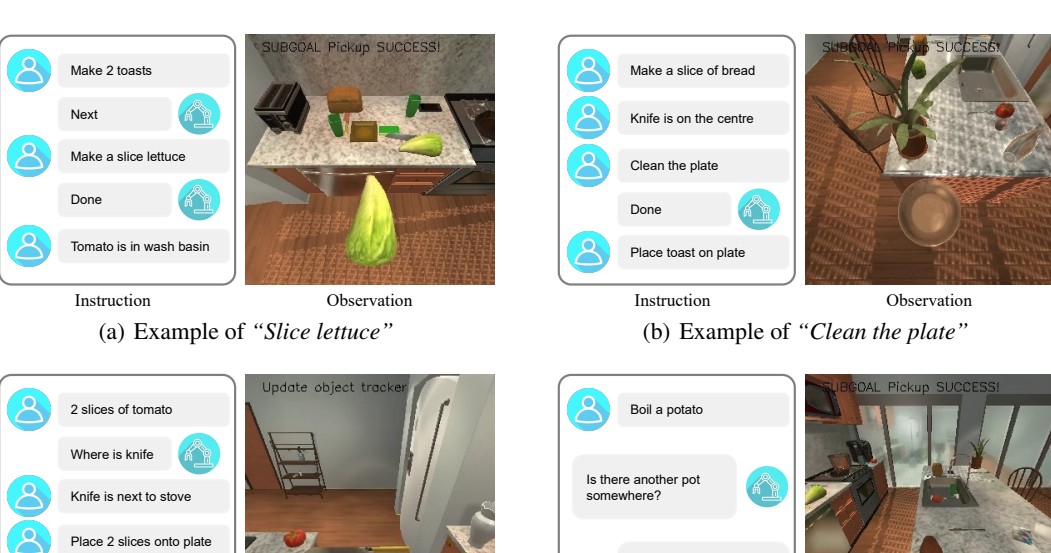

(a) Example of *"Slice lettuce"*          (b) Example of *"Clean the plate"*

(c) Example of *"Pick up Knife"*          (d) Example of *"Boil a potato"*

Figure 8: Examples of four task types: slice, clean, pick up, and boil

Table 8: Instructions and executable plans in TEACh

| | Type | Example |
|---|---|---|
| Instructions | Put all X on Y | <Commander>*"Put all tissue box on any side table."* |
| | Put all X in Y | <Commander>*"Hi. Today we are putting remote controls in a box."* *"The box is on the couch."* *"There is a remote control on the cabinet."* |
| | Water plant | <Commander>*"Water the plant."* |
| | Clean all X | <Driver>*"Hi, what do you need me to do?"* <Commander>*"Please clean all the plates."* |
| | Make X | <Commander>*"Please make a salad"* <Driver>*"Where do I find the thing I need?"* <Commander>*"the lettuce should be in the black bin"* |
| | Prepare X in Y | <Commander>*"We need to prepare coffee in a clean mug."* <Driver>*"Where is the mug please."* <Commander>*"In the sink."* |
| | Boil X on Y | <Commander>*"Boil a potato."* <Driver>*"Where is it?"* <Commander>*"On the chair."* |
| | Cook N slices of X Serve N slices of X | <Commander>*"Cook 5 slices of potato and serve on a plate."* <Commander>*"Make a 1 slice tomato"* |
| Plans | Pickup & Place | ```
target_newspaper = InteractionObject('Newspaper')
target_furniture = InteractionObject('Furniture')
target_newspaper.pickup()
target_newspaper.place(target_furniture)
``` |
| | Slice | ```
target_tomato = InteractionObject('Tomato', landmark = 'Sink')
target_tomato.go_to()
target_tomato.pickup()
target_tomato.slice()
``` |
| | Cook | ```
target_potato.pickup()
target_potato.go_to('Stove')
target_potato.boil()
``` |
| | Clean | ```
target_plate = InteractionObject('Plate', landmark = 'Sink')
target_plate.pickup()
target_plate.rinse()
``` |

## B.4 REAL-WORLD TEST

**Environment.** Our real-world setup consisted of a 7-DoF Franka Emika Research 3 robotic arm mounted on a tabletop workspace. For perception, we deployed two Intel RealSense D435 RGB-D cameras positioned on opposite sides of the table. The dual-camera configuration provided complementary viewpoints, and their depth streams were fused into a unified point cloud to reconstruct a high-resolution 3D map of the workspace. This spatial map served as the basis for accurate object localization and enabled reliable grounding of visual observations. The perception and control modules were integrated in real time, ensuring consistent operation during manipulation tasks.

**Object detection.** Task-relevant objects were placed on the tabletop within the robot's reachable workspace. To identify and localize them, RGB images captured by both cameras were processed using the Grounding DINO model, which generated category-aware bounding boxes. These 2D detections were then projected onto the fused 3D map, allowing us to recover precise object coordinates in the robot's reference frame. The combination of semantic cues (object category) and geometric grounding (3D position) enabled robust object detection and tracking across diverse physical configurations, which was crucial for executing manipulation skills.

**Task.** Real-world tasks are executed using a set of predefined primitive skill APIs. As summarized in Table 9, a total of 10 primitive skills are defined, including basic manipulation operations such as pick, place, move, and push. Each skill is associated with task-specific parameters; for example, pick allows adjustment of the gripper force, while pull controls the distance parameter. These continuous parameters enable fine-grained control in scenarios such as grasping cups of varying shapes or pulling

a drawer by a specified distance. Thus, the same primitive skill API can be flexibly adapted depending on the environment and task objectives. Real-world experiments were conducted in two environments, each consisting of multiple tasks with varying object configurations. For both environments, objects were sampled from a global object pool, and a random subset (typically 3-5 objects) was placed in randomized positions at the beginning of each trial. This randomization was performed across multiple trials (N=3 per task) to evaluate the robustness of our approach under diverse spatial configurations and object combinations.

The first environment is an office desk setup, where objects such as drawers, stationery, cups, and trash were placed on the desk. The tasks in this environment focus on organizing the workspace, including examples such as throwing paper into the trash bin or sorting stationery into a drawer. This environment was chosen to evaluate the compositionality of primitive skills and the reusability of cached code through repetitive organizing tasks within a constrained space. In particular, by repeatedly presenting tasks that are structurally similar but slightly varied, this setup assesses how $CA^2P$ can efficiently reuse skills and rapidly synthesize code policy.

The second environment is a cooking workstation preparation setup, which contains diverse objects such as a gas can, burner, and cooking utensils. The tasks in this environment focus on preparatory actions for cooking, such as removing objects placed on the burner to make it usable or organizing utensils to clear the workspace. This environment was chosen to evaluate whether the agent can detect new state changes or additional sub-tasks during execution and quickly update its policy to complete the task. Unlike the office desk setup, the tasks here cannot be solved by simply reusing cached code; instead, they are designed to require cache-refactoring to be executed. Consequently, this environment allows us to verify how effectively and efficiently the cache-based cache-refactoring mechanism of $CA^2P$ operates in real-world scenarios. Example images of both environments are shown in Figure 9.

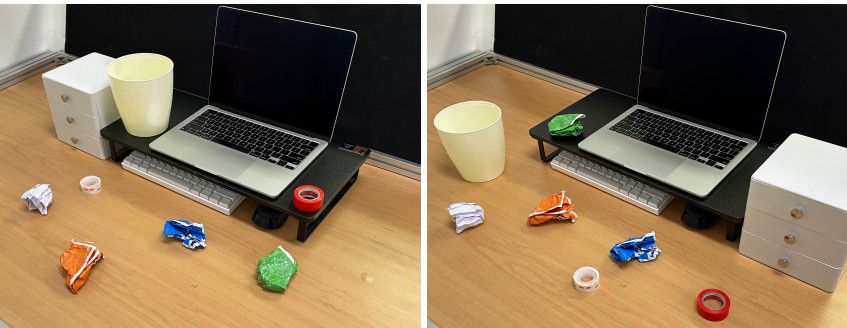

(a) Example of *Office Desk Rearrangement* environment

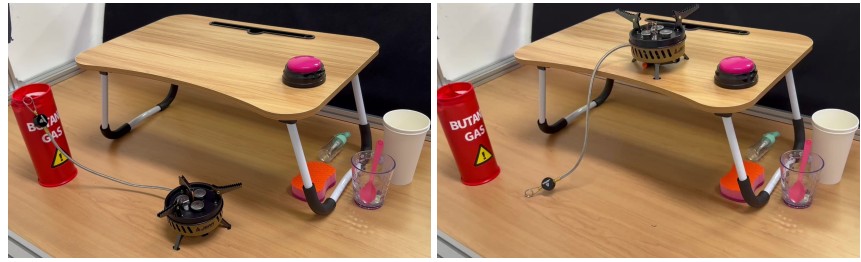

(b) Example of *Cooking Workstation Preparation* environment

Figure 9: Environment examples set of Real-world

Table 9: Instructions and executable APIs in Real-world

| | Template | Example |
|---|---|---|
| Instruction | Move | *"Move to the button."* |
| | Press | *"Press the emergency button."* |
| | Pick & Place | *"Throw the trash into the bin."* |
| | Pick & Pull | *"Open the drawer."* |
| | Pick & Push | *"Close the drawer."* |
| | Pick & Move | *"Take the tape and move it onto the drawer."* |
| API | Ready Pose | `go_to_ready_pose()` |
| | Set Target Pose [x,y,z,r,p,y] | `setTargetPose(...)` |
| | Move to Target | `execute_go()` |
| | Pick [gripper_force, axis] | `execute_pick(gripper_force=7, axis=2)` |
| | Place [axis] | `execute_place(axis=2)` |
| | Push [gripper_force, axis, distance] | `execute_push(gripper_force=5, axis=0, distance=0.08)` |
| | Pull [gripper_force, axis, distance] | `execute_pull(gripper_force=7, axis=1, distance=0.03)` |
| | Sweep [axis, distance] | `execute_sweep(axis=0, distance=0.04)` |
| | Rotate [gripper_force, axis] | `execute_rotate(gripper_force=6, axis=2)` |
| | Gripper [width, force] | `execute_gripper(0.005, 5)` |

## B.5 BASELINE

For comprehensive comparison, nine baseline methods are categorized into three groups:

- General CodeLLM-based programming methods leverage a CodeLLM to directly generate executable code policy from natural language instructions for solving embodied tasks.

  - CaP (Liang et al., 2023) introduces the Code-as-Policies paradigm, translating natural language instructions into executable robot control code.
  - SCoT (Li et al., 2025a) introduces structured chain-of-thought (CoT) prompting for reliable code generation, which we adapt to guide CodeLLM in producing code policy for embodied tasks.

- Memory-based approaches retrieve previously generated code or related textual information and supply it as additional context for policy synthesis.

  - HELPER (Sarch et al., 2023) equips embodied agents with an external memory of language–program pairs, enabling retrieval-augmented prompting for LLMs to parse open-domain instructions.
  - LRLL (Tziafas & Kasaei, 2024) is a lifelong learning framework that enables an LLM-based agent to dynamically grow and retrieve a skill library, allowing composable and generalizable policies for increasingly complex manipulation tasks.
  - PromptBook (Arenas et al., 2024) integrates examples, APIs, documentation, and CoT prompting to enable LLMs to generate code policy and acquire new low-level manipulation skills in a zero-shot manner.

- KV caching methods accelerate CodeLLM inference by reusing previously computed attention states throughout code policy generation.

  - CAG (Chan et al., 2025) proposes cache-augmented generation as an alternative to RAG by preloading relevant knowledge into the LLM's context and caching runtime parameters, thereby eliminating retrieval latency and errors while reducing system complexity.
  - RAGCache (Jin et al., 2024) introduces a multilevel dynamic caching system for RAG that stores and reuses intermediate knowledge states, reducing latency and computation costs while improving inference throughput.
  - PromptCache (Gim et al., 2024) accelerates LLM inference by caching and reusing attention states of recurring prompt segments, thereby achieving substantial latency reductions without modifying model parameters.
  - EPIC (Hu et al., 2024) introduces position-independent caching with the LegoLink algorithm to enable modular KV reuse beyond prefix matching, significantly improving LLM serving efficiency without sacrificing accuracy.

Unless otherwise noted, all baselines use the same CodeLLM configuration (max new tokens = 2048, temperature = 0.0, i.e., greedy decoding). When memory or cache modules are involved, we adopt the all-MiniLM-L6-v2 embedding model with cosine similarity for semantic retrieval.

**CaP.** We adopt CaP as a baseline framework by directly prompting a CodeLLM to generate executable code policy from natural language task instructions. In our implementation, the generated code is executed in the embodied environment without additional post-processing or fine-tuning, and each trial begins from a novel prompt without access to external memory or caching. As a result, long-context prompting and high token costs can become bottlenecks in long-horizon tasks, making this configuration representative of the raw performance of CaP in our evaluation. We refer to the publicly available implementation[1].

**SCoT.** We implement SCoT based on the code released by the authors (though the repository is no longer accessible), following their prompt design. The method uses a two-stage prompting format: first generating pseudo-code from task instructions, then producing executable code conditioned on it. This structured decomposition improves reliability compared to CaP. SCoT still follows CaP's design. It relies on long prompts for each new task without memory or reuse and is inefficient in long-horizon settings.

**HELPER.** We adopt HELPER as released, with no changes to hyperparameters or architectural components. The system retrieves the top-3 most relevant memory entries based on cosine similarity to the current task instruction. This retrieval-augmented prompting improves adaptability to diverse instructions. HELPER does not accumulate new experiences into memory and remains sensitive to noisy or irrelevant entries, relying on long-context code generation without efficient reuse. We refer to the publicly available implementation[2].

**LRLL.** We adopt LRLL as a baseline by implementing its core mechanisms for growing and retrieving a skill library with a CodeLLM for policy generation. Relevant skills are identified through semantic retrieval using cosine similarity (top-3) from the skill library based on the current task instruction. These skills are incorporated into the prompt context along with the task specification. The CodeLLM generates the final policy by composing the retrieved components. This design enables continual accumulation and reuse of skills through the library and provides more flexibility than approaches that rely only on transient retrieval. In practice, performance depends on how reliably the retrieved skills are composed. The experience memory also becomes more difficult to manage as it grows.

**PromptBook.** We implement PromptBook as a baseline by constructing prompts that combine API documentation, three to five usage examples per API, and chain-of-thought templates for task decomposition, with example retrieval based on semantic similarity. The assembled prompt guides the CodeLLM through reasoning steps before generating the final code policy. Prompts are typically on the order of about two thousand tokens depending on the retrieved examples. The design scales less efficiently because task-relevant information is provided within the prompt for each new task rather than being accumulated in a reusable library.

**CAG.** We adapt CAG, originally proposed for text-based QA, by integrating it with CaP so that the method produces executable code policy in CaP format for embodied control tasks. Successful code policy and associated skills are stored in the cache as preloaded context and reused for subsequent tasks instead of retrieving external text. The design reduces retrieval latency and errors through cached knowledge. The cache is fixed once populated and does not support dynamic updates, so adaptability to new tasks and environments is limited. We refer to the publicly available implementation[3].

**RAGCache.** We adapt RAGCache, originally proposed as a caching framework for RAG, by implementing it in the CaP setting so that it produces executable code policy for embodied control tasks. In our implementation, generated policies are stored in the cache together with their task descriptions. The cache is incrementally updated through prefix-based indexing to support retrieval of partial matches. On a cache hit, the retrieved policy is adapted to the current task using the CodeLLM. On a miss, a new policy is generated and added to the cache for future reuse. The design enables

---

[1]https://github.com/google-research/google-research/tree/master/code_as_policies

[2]https://github.com/Gabesarch/HELPER

[3]https://github.com/hhhuang/CAG

reuse of accumulated policies over time. Prefix-based retrieval can lead to low hit rates when the cache is still small.

**PromptCache.** We adapt PromptCache, originally introduced to accelerate inference on general text tasks, by reimplementing it in a CaP setting so that it produces executable code policy for embodied control. Our implementation caches the key–value states of recurring prompt segments, indexed by task prefixes, and reuses them for new tasks when sufficient similarity is detected. The design provides efficiency gains when prompt structures repeat. The cache is static and does not support dynamic updates, since it follows the original design around fixed prompt templates. Adaptability in diverse embodied tasks is limited. We refer to the publicly available implementation[4].

**EPIC.** We implement EPIC for embodied control by adapting its caching methodology to the CaP setting so that the system produces executable code policy. As the original implementation is not publicly available, we reproduced its approach based on the paper description. We added a preprocessing layer that transforms and aligns cached key–value states for task-specific adaptation before reuse. Cached representations are then fed to the CodeLLM to guide policy generation. The design enables flexible reuse of cached prompt information across tasks. The cache is static and does not support dynamic updates, which reduces adaptability in diverse embodied scenarios.

### B.6 METRIC

Performance evaluation is conducted using diverse metrics across four complementary dimensions:

- **Task accuracy** evaluates how reliably an agent achieves task goals.
  - Task success rate (SR) measures the proportion of tasks in which all required sub-goals are successfully completed, indicating overall task-level performance (Shridhar et al., 2020).
  - Goal-conditioned success rate (GC) computes the fraction of individual sub-goals achieved across all tasks, capturing the agent's partial progress even when full task completion is not attained (Shridhar et al., 2020).
- **Computational efficiency** quantifies the inference cost of policy synthesis.
  - Policy synthesis latency (PSL) measures the average time required by a CodeLLM-based method to produce executable code policy.
  - Time to First Token (TTFT) measures the initial decoding delay before the first output token is generated.
  - Number of generated tokens (NGT) counts the total tokens generated during code policy generation.
- **Behavioral consistency** assesses coherence and transferability across tasks.
  - Code similarity (CSIM) quantifies the structural overlap between code policy generated for logically equivalent tasks (Chon et al., 2024).
  - Forward transfer (FWT) measures the improvement in performance on future tasks resulting from updating the code cache with previously encountered tasks, thereby capturing positive knowledge transfer (Lopez-Paz & Ranzato, 2017).
  - Backward transfer (BWT) measures the effect of updating the code cache with subsequent tasks on performance for previously seen tasks, indicating whether prior knowledge is preserved or forgotten (Lopez-Paz & Ranzato, 2017).
- **Memory efficiency** quantifies the effective use of GPU memory resources during policy synthesis.
  - Hit rate (HR) computes the proportion of successful cache reuses during policy synthesis.
  - GPU memory utilization (MU) quantifies the proportion of peak allocated GPU memory with respect to the available capacity (Liu et al., 2025).
- Rank is computed as a weighted sum of SR and the inverse of min–max normalized PSL, yielding a single scalar indicator that balances effectiveness and efficiency.

Table 1 and Table 2 in Section 4.2 report the Rank metric, which combines task effectiveness and computational efficiency into a single scalar score. Specifically, we first compute the average PSL for each method and apply min-max normalization across all baselines. Since lower PSL values indicate better efficiency, we use the inverse of the normalized PSL to align its direction with SR. The final

---

[4]https://github.com/MachineLearningSystem/24MLSYS-prompt-cache

Rank is then calculated as a weighted sum of SR and the inverse-normalized PSL with $\alpha = \beta = 0.5$, as defined in Eq. (7).

$$\text{Rank} = \alpha \cdot \text{SR} + \beta \cdot \left(1 - \frac{\text{PSL} - \text{PSL}_{\min}}{\text{PSL}_{\max} - \text{PSL}_{\min}}\right) \tag{7}$$

## C  CA$^2$P: CACHE-AUGMENTED CODE-AS-POLICIES

### C.1  IMPLEMENTATION

Table 10: System configuration for the main framework server

| Component | Specification |
|---|---|
| Python | 3.9.18 |
| CUDA / cuDNN | CUDA 12.4 / cuDNN 9.1 |
| PyTorch | 2.5.1 |
| Transformers | 4.46.3 |
| GPU | NVIDIA RTX 4090 (24GB) |
| Random Seed | 3 runs with seeds 1, 2, 3 |

**System Architecture.**  The framework server coordinates prompt orchestration, code generation, cache access, and feedback-guided regeneration. The simulation server executes the generated code policy within the environment and returns structured feedback to the framework server. The two servers communicate asynchronously via REST APIs with JSON message passing. Control is framework-driven in RLBench and simulation-initiated in ALFRED and TEACh, reflecting differences in the benchmark interfaces. Algorithm 1 summarizes the control flow.

**Key Implementation Details.**  The following points summarize implementation aspects that are critical for faithful reproduction:

- **Prompt construction.** Prompts include task instructions and observations under a fixed header and canonical entry point. They also contain function documentation and example implementations that can be reused during replanning. We support multiple observation modalities, including natural-language instructions in RLBench, scene descriptions in ALFRED, dialogue in TEACh, and symbolic states in the real-world setup. Full task-specific templates are provided in the code release.

> **Prompt example**
>
> ```
> # MANDATORY: Use these EXACT objects and skills. NO placeholders!
> # Available objects: 'block', 'target', 'success'
> # Available skills: move, pick, place, push, open_gripper, close_gripper
> # IMPORTANT for button/switch tasks: close_gripper first, then push/press
> # For this specific task, start with: push, 'block')
> # Then continue with the remaining implementation.
> # If a line may cause errors or need revision,
> # insert '# ERROR_FLAG' immediately before it.
>
> # Starting code (continue from here):
> obs, reward, done = run_action(skill, object, offset(if needed))
> <task instruction text continues here...>
> ```

- **Clarification of locality score components.** The locality score comprises three components-$\ell_{\text{freq}}$, $\ell_{\text{asso}}$, and $\ell_{\text{sema}}$-each designed to capture a distinct aspect of behavioral regularities that arise in open-domain task streams. These terms are not intended to preserve recency alone; rather, they

reflect temporal repetition, compositional dependencies, and semantic distinctiveness. Formal definitions are provided below.

- $\ell_{\text{freq}}$ *(Usage frequency).* This term measures the relative frequency with which a function is invoked over recent tasks:

$$\ell_{freq}(f_k) = \frac{\text{count}(f_k)}{\max_j \text{count}(f_j)} \tag{8}$$

It captures repetitive or recurrent usage patterns that arise during continuous task execution.

- $\ell_{\text{asso}}$ *(Conditional association).* This component quantifies compositional relationships by measuring the conditional likelihood that $f_j$ is invoked following $f_k$ within an execution trace:

$$\ell_{asso}(f_j \mid f_k) = \frac{\text{cooccur}(f_k, f_j)}{\text{count}(f_k)} \tag{9}$$

It reflects order-dependent transitions and functional co-occurrence patterns that frequently appear in multi-step procedures.

- $\ell_{\text{sema}}$ *(Semantic novelty).* This term evaluates the semantic distinctiveness of a function by computing the perplexity of its implementation under the CodeLLM:

$$\ell_{sema}(f_k) = \frac{\text{PPL}(f_k)}{\max_j \text{PPL}(f_j)} \tag{10}$$

It promotes the retention of semantically unique or infrequent functions, supporting robustness to atypical or unpredictable task variations.

- **Compositional Programming.** During the initial generation, we inject KV states from *Function Interface tier* ($\mathcal{I}$) entries as additional context. *Function Code tier* entries ($\mathcal{C}$) are excluded at this stage to avoid redundant reasoning over full implementations. In subsequent generations, ($\mathcal{C}$) KV states are included, but only for those functions whose ($\mathcal{I}$) entries were actually used. The model decodes greedily and stops at predefined stop phrases or EOS. Stop phrases include markers such as '# code_end' or 'def'. The complete list is provided in the released code. Generated lines are appended after the canonical entry point and must be executable code. Non-executable content such as comments or placeholders is filtered by the stop rules.

- **Execution feedback.** During policy execution, structured feedback is recorded whenever the generated policy either fails to execute or does not satisfy task requirements. Feedback spans from low-level interpreter or API errors, to skill-level execution failures, to higher-level semantic violations and unmet goals. This design ensures that cache-refactoring is not limited to repairing syntactic errors, but also incorporates semantic corrections such as adjusting object usage or addressing unsatisfied subgoals. Representative feedback categories are illustrated in the code release.

- **Cache-refactoring.** Before execution, undefined function calls or missing parameters are flagged during generation. After generation, the flagged spans are corrected first, and then the program is executed. During execution, if a runtime exception occurs, the error is localized at the line level and the program is split into prefix, error, and suffix. We apply *fill-in-the-middle* guided by a short structured CoT trace, reusing preserved KV states for the prefix and suffix and regenerating only the middle span. The CoT trace is discarded before re-execution, and only the corrected code is run. Cache-refactoring after execution is limited to three retries. The same refinement mechanism is applied across all benchmarks with adaptations for each observation type.

---

**Example of cache-refactoring in real-world**

**Prompt:**

```
# Task: {instruction}
# IMPORTANT: Functions from robot_tasks are already imported.
Just call them directly, DO NOT redefine them.
# For other tasks without matching functions,
use basic skills (setTargetPose, execute_pick, execute_place)
```

```
# If a line may cause errors or need revision,
# insert '# ERROR_FLAG' immediately before it.

def robot_move(self):
        self.ps.go_to_ready_pose()
        # First handle the trash task using the imported function
```

**Generated code:**

```
...
push_button_task(self.ps, self.get_obj)
...
```

**Error feedback (as provided by API)**

```
# Task instruction:
Push the gas_manage_button to stop the gas leak.

# Generated robot_move implementation:
self.ps.go_to_ready_pose()
push_button_task(self.ps, self.get_obj)
self.ps.go_to_ready_pose()

# Known discrepancies:
Missing objects: button

# Think step-by-step about why the generated code fails and how to correct it.
Respond in strict JSON with keys:
  "reasoning": ordered list of short reasoning steps,
  "fixes": ordered list of concrete code edits or high-level actions,
  "verdict": brief summary of the main failure reason.
Let's reason step by step.
```

**Result of cache-refactoring:**

push_button_task(self.ps, self.get_obj, + button_obj="gas_manage_button")

- **Unified exception handling.** We thank the reviewer for raising the concern regarding real-world failure detection, and we have clarified this aspect in the Appendix. Although the internal mechanisms differ between simulation and real-world execution, we standardize all failure signals into a unified exception interface. Interpreter-level exceptions (e.g., `SyntaxError`, `NameError`) arise from Python execution and behave identically across environments, while environment-level exceptions (`RobotError`) capture grasp failures, workspace violations, and other skill-level issues. In simulation, these failures are reported directly through the native APIs. In real-world deployment with the Franka Emika Research 3, we detect failures using depth cameras, object detectors, and a VLM-based state check. For example, unsuccessful grasps or unmet semantic conditions (e.g., drawer not opened) trigger a `RobotError` when the observation does not match the expected postcondition. All exceptions-syntactic, execution-level, or perception-driven-are funneled through the same interface, enabling cache-refactoring to resolve them uniformly. Thus, the try-except block in Algorithm 1 is not simulation-specific; it reflects an environment-agnostic design that consistently supports both simulated and real-world execution.

## C.2 HYPERPARAMETER

**LLM Settings.** We use `Qwen2.5-Coder-14B` via Hugging Face with 8-bit quantization. For size ablations on the primary code model we evaluate `Qwen2.5-Coder` at {3B, 7B, 14B}. For cross-family ablations we fix the scale at 7B and compare the families Qwen2.5, Gemma, Llama 2, and DeepSeek; we additionally evaluate 7B general-purpose LLMs as drop-in code generators. All decoding and framework-level hyperparameters are held fixed as in Table 12.

Table 11: Interpreter-level and environment-level exceptions used in our framework

| Type | Category | Details |
|---|---|---|
| Interpreter-level | SyntaxError | Invalid Python syntax
Malformed code block
Non-code text inside code region |
| Interpreter-level | ClassNotFound | Missing Pygments lexer for code block
Fallback to generic text mode
Unsupported code type detected |
| Interpreter-level | FileNotFoundError | Missing object metadata file
Missing orientation mapping file
Missing noise parameter configuration |
| Environment-level | RobotError | Skill sequence execution failure
Path out of workspace
Object out of reach
Grasp or manipulation failure
Timeout during primitive execution
Invalid object type binding
Missing or undefined object handle
Missing constraints in object mapping
Infeasible approach pose
Orientation constraint violation
Safety stop or low-level controller failure |

Table 12: Hyperparameters (decoding and framework-level)

| **Model generation hyperparameters** | |
|---|---|
| max_new_tokens | 2048 |
| temperature | 0.0 (greedy) |
| top-$k$, top-$p$ | N/A (greedy; not used) |
| **Framework-level hyperparameters** | |
| Eviction threshold (perplexity-based) | $\tau = 15.0$ |
| Locality weights ($\alpha$: temporal, $\beta$: spatial, $\gamma$: semantic) | $\alpha = 0.4$, $\beta = 0.3$, $\gamma = 0.3$ |
| Execution trace length | 20 |
| Temporal decay rate | 0.01 |
| GPU memory threshold (%) / reserve (GB) | 0.85 / 2.0 |
| Min free GPU before generation (GB) | 1.5 |
| Top-$N$ blocks used | 2 |
| FIM repair: max_tokens | 128 |

# D  ADDITIONAL EXPERIMENTAL RESULT

## D.1  BENCHMARK EXPERIMENT DETAILS

Table 13 expands on Table 1 in Section 4.2 by reporting detailed results across all metrics for each open-domain scenario in the ALFRED benchmark. In Table 13(a), CA$^2$P achieves the highest SR, improving by 5.30% over LRLL. For GC, it surpasses LRLL by 3.72%. In terms of PSL, where lower is better, CA$^2$P reduces it by 2.08$\times$ compared to CAG. NGT is also significantly reduced by 58.18 tokens relative to RAGCache. All cache-based baselines and CA$^2$P begin with the same initial KV cache states derived from basic success code policies. CAG, EPIC, and PromptCache do not update the cache afterward, so no eviction occurs and the hit ratio remains 100%. RAGCache and

Table 13: Extended results from ALFRED benchmark evaluation on open-domain embodied tasks

(a) *Open-Composition* scenario

| Method | SR | GC | PSL | NGT | HR | MU |
|---|---|---|---|---|---|---|
| CaP | 46.48±1.87 | 66.03±2.72 | 16.21±1.03 | 186.43±5.30 | - | 24.87±0.38 |
| SCOT | 47.93±0.87 | 68.38±2.38 | 30.41±8.02 | 302.41±8.02 | - | 26.43±0.82 |
| HELPER | 53.83±0.39 | 76.03±3.23 | 16.82±1.20 | 175.83±6.23 | - | 25.39±1.02 |
| LRLL | 56.28±3.20 | 78.32±2.03 | 16.02±2.02 | 198.41±12.04 | - | 26.20±1.27 |
| PromptBook | 53.94±3.96 | 75.81±2.30 | 31.34±1.82 | 274.40±7.32 | - | 28.43±2.23 |
| CAG | 38.24±0.04 | 55.27±0.23 | 12.12±0.62 | 174.03±3.94 | 100.00±0.00 | 47.88±0.03 |
| RAGCache | 39.29±1.70 | 58.43±1.82 | 13.48±0.93 | 162.48±2.73 | 58.93±3.04 | 75.86±0.49 |
| EPIC | 43.38±1.02 | 62.01±2.20 | 14.02±1.04 | 172.46±5.71 | 100.00±0.00 | 53.03±1.02 |
| PromptCache | 40.37±0.37 | 59.06±3.29 | 12.82±0.86 | 173.51±5.91 | 100.00±0.00 | 50.47±2.22 |
| CA$^2$P | 61.58±1.86 | 82.04±2.22 | 5.82±0.57 | 104.30±5.32 | 75.46±1.21 | 86.08±0.67 |

(b) *Open-Perturbation* scenario

| Method | SR | GC | PSL | NGT | HR | MU |
|---|---|---|---|---|---|---|
| CaP | 42.23±0.12 | 60.83±0.32 | 16.15±0.23 | 184.53±0.42 | - | 24.23±1.02 |
| SCOT | 43.46±1.06 | 64.17±1.52 | 32.39±1.12 | 325.12±8.29 | - | 27.82±1.92 |
| HELPER | 52.49±0.24 | 73.82±0.52 | 15.94±0.69 | 189.62±3.02 | - | 26.39±1.93 |
| LRLL | 54.70±0.87 | 75.00±1.28 | 16.40±1.24 | 214.95±4.22 | - | 27.51±2.39 |
| PromptBook | 52.63±1.09 | 73.89±1.02 | 33.29±1.52 | 364.82±5.19 | - | 30.12±2.86 |
| CAG | 36.03±0.32 | 53.02±0.65 | 12.38±0.92 | 182.81±2.74 | 100.00±0.00 | 47.54±2.21 |
| RAGCache | 36.33±0.02 | 53.25±0.08 | 13.15±0.12 | 184.74±2.04 | 52.45±0.32 | 75.78±0.23 |
| EPIC | 37.08±1.02 | 56.66±1.22 | 13.74±0.69 | 178.29±5.72 | 100.00±0.00 | 54.83±0.32 |
| PromptCache | 36.92±2.22 | 55.12±1.95 | 12.93±1.48 | 163.35±8.92 | 100.00±0.00 | 51.25±1.08 |
| CA$^2$P | 57.23±0.23 | 78.77±0.66 | 6.32±0.51 | 128.50±2.20 | 71.04±1.00 | 82.45±0.81 |

(c) *Open-Evolution* scenario

| Method | SR | GC | PSL | NGT | HR | MU |
|---|---|---|---|---|---|---|
| CaP | 37.22±1.72 | 55.29±2.29 | 16.28±1.20 | 165.81±6.03 | - | 26.12±2.22 |
| SCOT | 39.00±0.98 | 57.77±1.36 | 32.04±2.10 | 334.68±7.03 | - | 27.83±2.92 |
| HELPER | 47.75±1.08 | 68.92±1.52 | 16.42±0.92 | 214.32±4.55 | - | 27.92±0.76 |
| LRLL | 51.60±0.30 | 70.81±1.00 | 16.63±1.03 | 214.82±2.11 | - | 28.23±0.76 |
| PromptBook | 50.49±0.51 | 70.54±0.78 | 33.49±2.07 | 368.90±7.90 | - | 31.83±0.96 |
| CAG | 32.18±1.12 | 47.75±1.51 | 12.39±0.92 | 174.42±3.30 | 100.00±0.00 | 50.95±0.93 |
| RAGCache | 33.83±1.53 | 48.44±2.02 | 13.02±1.15 | 188.12±6.32 | 48.67±3.20 | 73.54±1.82 |
| EPIC | 34.23±0.23 | 51.48±1.52 | 13.49±1.14 | 177.23±5.81 | 100.00±0.00 | 51.32±1.22 |
| PromptCache | 34.14±0.82 | 50.22±1.62 | 13.18±1.15 | 173.58±4.26 | 100.00±0.00 | 47.83±1.92 |
| CA$^2$P | 55.89±0.85 | 75.46±1.71 | 6.29±1.25 | 204.34±3.21 | 70.26±1.82 | 78.37±1.21 |

CA$^2$P keep inserting new entries after initialization, which reduces the hit ratio. Nevertheless, CA$^2$P excels in MU, achieving 10.22% higher than RAGCache. In Table 13(b), CA$^2$P achieves the highest SR, improving by 2.53% over LRLL. For GC, it surpasses LRLL by 3.77%. In terms of PSL, where lower is better, CA$^2$P reduces it by 1.96× compared to CAG. NGT is also significantly reduced by 34.85 tokens relative to PromptCache. Furthermore, CA$^2$P excels in MU, achieving 6.67% higher than RAGCache. In Table 13(c), CA$^2$P achieves the highest SR, improving by 4.29% over LRLL. For GC, it surpasses LRLL by 4.65%. In terms of PSL, where lower is better, CA$^2$P reduces it by 1.97× compared to CAG. NGT increases by 38.53 tokens compared to CaP. Furthermore, CA$^2$P excels in MU, achieving 4.83% higher than RAGCache.

Table 14 expands on Table 1 in Section 4.2 by reporting detailed results across all metrics for each open-domain scenario in the TEACh benchmark. In Table 14(a), CA$^2$P achieves the highest SR, improving by 9.60% over LRLL. For GC, it surpasses LRLL by 8.00%. In terms of PSL, where lower is better, CA$^2$P reduces it by 2.72× compared to CAG. NGT is also significantly reduced by 79.60 tokens relative to HELPER. Furthermore, CA$^2$P excels in MU, achieving 10.95% higher than RAGCache. In Table 14(b), CA$^2$P achieves the highest SR, improving by 9.48% over LRLL. For GC, it surpasses LRLL by 8.20%. In terms of PSL, where lower is better, CA$^2$P reduces it by 2.50×

Table 14: Extended results from TEACh benchmark evaluation on open-domain embodied tasks

(a) *Open-Composition* scenario

| Method | SR | GC | PSL | NGT | HR | MU |
|---|---|---|---|---|---|---|
| CaP | 43.92±0.81 | 65.08±1.11 | 17.29±0.73 | 208.39±5.67 | - | 24.84±1.07 |
| SCOT | 44.03±1.47 | 65.73±1.52 | 33.00±0.91 | 316.82±8.09 | - | 27.10±2.89 |
| HELPER | 47.14±0.63 | 69.45±1.12 | 17.31±1.08 | 201.81±3.28 | - | 26.28±1.05 |
| LRLL | 48.99±1.73 | 71.20±2.15 | 17.82±0.82 | 221.39±2.10 | - | 26.71±1.22 |
| PromptBook | 47.08±2.24 | 70.85±2.86 | 36.92±1.47 | 355.11±12.92 | - | 30.29±2.71 |
| CAG | 35.03±2.01 | 60.39±2.41 | 14.23±1.28 | 232.59±4.31 | 100.00±0.00 | 47.66±3.04 |
| RAGCache | 35.23±0.05 | 57.03±0.12 | 14.97±0.15 | 252.44±1.08 | 48.48±0.32 | 77.28±0.84 |
| EPIC | 36.39±0.64 | 59.98±3.07 | 15.43±0.44 | 226.95±0.91 | 100.00±0.00 | 50.83±0.61 |
| PromptCache | 36.15±0.86 | 59.28±0.92 | 14.88±0.29 | 217.50±1.30 | 100.00±0.00 | 52.71±0.83 |
| $CA^2P$ | 58.59±0.39 | 79.20±0.64 | 5.23±0.09 | 122.21±5.73 | 76.24±0.39 | 88.23±0.48 |

(b) *Open-Perturbation* scenario

| Method | SR | GC | PSL | NGT | HR | MU |
|---|---|---|---|---|---|---|
| CaP | 42.21±0.28 | 63.83±0.63 | 17.23±0.39 | 212.24±2.75 | - | 25.09±0.38 |
| SCOT | 42.49±1.02 | 65.31±1.98 | 33.94±1.92 | 292.84±5.09 | - | 28.39±1.08 |
| HELPER | 44.05±0.29 | 67.19±0.83 | 17.39±1.02 | 224.00±0.08 | - | 26.93±0.62 |
| LRLL | 46.78±0.91 | 68.29±0.92 | 17.93±1.22 | 226.54±1.02 | - | 27.36±1.40 |
| PromptBook | 44.85±1.00 | 66.98±1.47 | 35.04±1.24 | 363.71±5.30 | - | 30.29±1.22 |
| CAG | 33.89±0.92 | 57.00±1.02 | 14.02±0.24 | 214.29±2.40 | 100.00±0.00 | 49.01±2.09 |
| RAGCache | 34.08±0.04 | 57.56±0.98 | 15.51±0.42 | 222.30±2.20 | 49.28±0.43 | 76.89±0.00 |
| EPIC | 34.39±0.51 | 61.03±0.58 | 15.66±0.48 | 212.71±1.12 | 100.00±0.00 | 51.38±0.35 |
| PromptCache | 33.12±0.12 | 59.42±0.39 | 14.94±1.31 | 224.53±2.04 | 100.00±0.00 | 45.84±0.00 |
| $CA^2P$ | 56.26±1.09 | 76.49±1.32 | 5.61±1.83 | 127.11±6.21 | 74.02±1.18 | 89.03±1.12 |

(c) *Open-Evolution* scenario

| Method | SR | GC | PSL | NGT | HR | MU |
|---|---|---|---|---|---|---|
| CaP | 38.07±0.12 | 58.97±0.33 | 17.03±1.20 | 239.60±5.20 | - | 24.01±1.08 |
| SCOT | 37.91±0.41 | 56.49±0.39 | 35.03±1.58 | 312.49±5.23 | - | 30.41±1.29 |
| HELPER | 40.00±0.38 | 61.47±1.05 | 17.48±0.71 | 211.60±3.04 | - | 32.70±0.24 |
| LRLL | 41.85±0.02 | 61.84±0.12 | 18.03±0.39 | 214.48±0.87 | - | 32.03±0.08 |
| PromptBook | 41.07±0.32 | 62.78±0.48 | 36.82±0.49 | 378.14±8.39 | - | 36.16±0.89 |
| CAG | 30.58±0.33 | 52.93±0.38 | 14.82±0.11 | 234.12±0.91 | 100.00±0.00 | 50.59±0.20 |
| RAGCache | 29.45±1.23 | 53.41±1.87 | 16.07±0.29 | 218.00±0.83 | 43.41±0.33 | 81.18±0.64 |
| EPIC | 33.92±1.52 | 58.75±1.74 | 17.18±0.54 | 226.10±2.22 | 100.00±0.00 | 52.83±1.04 |
| PromptCache | 31.07±1.49 | 55.02±0.49 | 16.32±0.24 | 217.6±0.92 | 100.00±0.00 | 51.71±0.39 |
| $CA^2P$ | 52.93±0.62 | 75.39±0.87 | 7.03±0.82 | 128.20±5.09 | 70.92±1.02 | 81.82±0.39 |

compared to CAG. NGT is also significantly reduced by 85.13 tokens relative to CaP. Furthermore, $CA^2P$ excels in MU, achieving 12.14% higher than RAGCache. In Table 14(c), $CA^2P$ achieves the highest SR, improving by 11.08% over LRLL. For GC, it surpasses LRLL by 13.55%. In terms of PSL, where lower is better, $CA^2P$ reduces it by 2.11× compared to CAG. NGT is also significantly reduced by 83.40 tokens relative to HELPER. Furthermore, $CA^2P$ excels in MU, achieving 0.64% higher than RAGCache.

Table 15: Extended results from RLBench benchmark evaluation on open-domain embodied tasks

(a) *Open-Composition* scenario

| Method | SR | PSL | NGT | HR | MU |
|---|---|---|---|---|---|
| CaP | 31.23±3.32 | 11.39±0.53 | 123.43±3.55 | - | 23.08±0.88 |
| SCOT | 32.32±2.22 | 22.03±2.03 | 246.32±12.43 | - | 27.48±2.33 |
| HELPER | 35.42±1.32 | 11.48±0.23 | 154.18±1.45 | - | 26.48±1.93 |
| LRLL | 36.11±0.00 | 10.83±0.82 | 120.00±2.15 | - | 26.43±0.00 |
| PromptBook | 35.93±1.12 | 23.49±0.63 | 206.48±3.23 | - | 28.23±0.73 |
| CAG | 26.37±2.03 | 8.53±0.83 | 157.52±10.30 | 100.00±0.00 | 44.95±4.50 |
| RAGCache | 27.26±1.08 | 9.09±0.63 | 149.35±5.02 | 59.02±1.03 | 75.34±4.83 |
| EPIC | 29.24±2.88 | 9.48±1.20 | 163.81±5.32 | 100.00±0.00 | 48.39±2.22 |
| PromptCache | 28.33±1.73 | 8.99±0.39 | 272.90±1.67 | 100.00±0.00 | 47.33±1.01 |
| $\text{CA}^2\text{P}$ | 45.91±4.37 | 3.24±0.65 | 105.62±3.09 | 73.65±2.15 | 83.63±3.12 |

(b) *Open-Perturbation* scenario

| Method | SR | PSL | NGT | HR | MU |
|---|---|---|---|---|---|
| CaP | 30.46±0.02 | 11.23±0.93 | 132.84±10.82 | - | 23.28±2.83 |
| SCOT | 30.32±0.28 | 21.37±1.84 | 189.23±15.64 | - | 29.32±3.85 |
| HELPER | 34.84±0.03 | 11.38±0.06 | 159.08±1.45 | - | 28.34±1.75 |
| LRLL | 33.82±0.02 | 12.01±0.11 | 122.81±0.78 | - | 26.47±2.30 |
| PromptBook | 33.03±0.67 | 24.03±3.89 | 192.93±12.68 | - | 28.32±0.93 |
| CAG | 24.89±1.21 | 8.39±0.37 | 221.47±3.84 | 100.00±0.00 | 45.49±0.82 |
| RAGCache | 27.03±1.04 | 9.28±0.66 | 238.79±5.38 | 57.38±1.08 | 78.93±2.02 |
| EPIC | 29.04±0.83 | 9.48±0.32 | 208.45±3.04 | 100.00±0.00 | 48.29±2.23 |
| PromptCache | 27.41±0.81 | 8.97±0.35 | 250.40±5.83 | 100.00±0.00 | 47.55±1.42 |
| $\text{CA}^2\text{P}$ | 42.82±2.64 | 3.47±0.16 | 106.00±5.65 | 70.73±2.73 | 82.34±1.04 |

(c) *Open-Evolution* scenario

| Method | SR | PSL | NGT | HR | MU |
|---|---|---|---|---|---|
| CaP | 26.84±1.20 | 11.23±0.24 | 152.62±5.03 | - | 24.62±1.03 |
| SCOT | 27.11±0.37 | 22.83±1.42 | 189.36±5.23 | - | 28.54±1.40 |
| HELPER | 31.12±1.08 | 12.16±1.42 | 174.00±3.33 | - | 28.81±0.02 |
| LRLL | 32.45±0.94 | 12.17±2.01 | 198.83±0.53 | - | 28.58±2.32 |
| PromptBook | 31.65±3.32 | 25.93±2.09 | 231.96±8.39 | - | 31.34±3.83 |
| CAG | 22.82±0.73 | 8.08±1.91 | 178.81±5.39 | 100.00±0.00 | 44.39±2.93 |
| RAGCache | 23.79±1.20 | 9.02±0.18 | 183.11±4.23 | 53.49±1.27 | 82.79±0.34 |
| EPIC | 25.85±1.25 | 9.28±0.47 | 182.31±6.34 | 100.00±0.00 | 48.81±0.53 |
| PromptCache | 25.03±0.85 | 9.12±0.65 | 193.77±3.78 | 100.00±0.00 | 47.93±0.75 |
| $\text{CA}^2\text{P}$ | 33.98±1.27 | 4.39±0.43 | 101.02±3.02 | 70.29±3.21 | 82.27±6.27 |

Table 15 expands on Table 1 in Section 4.2 by reporting detailed results across all metrics for each open-domain scenario in the RLBench benchmark. In Table 15(a), $\text{CA}^2\text{P}$ achieves the highest SR, improving by 9.80% over LRLL. In terms of PSL, where lower is better, $\text{CA}^2\text{P}$ reduces it by 2.63× compared to CAG. NGT is also significantly reduced by 14.38 tokens relative to LRLL. Furthermore, $\text{CA}^2\text{P}$ excels in MU, achieving 8.29% higher than RAGCache. In Table 15(b), $\text{CA}^2\text{P}$ achieves the highest SR, improving by 9.00% over LRLL. In terms of PSL, where lower is better, $\text{CA}^2\text{P}$ reduces it by 2.42× compared to CAG. NGT is also significantly reduced by 16.81 tokens relative to LRLL. Furthermore, $\text{CA}^2\text{P}$ excels in MU, achieving 3.41% higher than RAGCache. In Table 15(c), $\text{CA}^2\text{P}$ achieves the highest SR, improving by 1.53% over LRLL. In terms of PSL, where lower is better, $\text{CA}^2\text{P}$ reduces it by 1.84× compared to CAG. NGT is also significantly reduced by 51.60 tokens relative to CaP. Furthermore, $\text{CA}^2\text{P}$ reports slightly lower MU compared to RAGCache.

## D.2 REAL-WORLD EXPERIMENT DETAILS

### Table 16: Extended results from Real-world robotic manipulation

#### (a) *Office Desk Rearrangement*

| Method | SR | GC | PSL | NGT | HR | MU |
|--------|-----|-----|-----|-----|-----|-----|
| CaP | 33.33±0.00 | 44.44±0.00 | 11.94±1.17 | 148.02±3.30 | - | 25.91±1.93 |
| LRLL | 55.56±11.11 | 60.74±9.25 | 12.31±0.74 | 146.39±5.83 | - | 29.04±0.84 |
| RAGCache | 44.44±22.22 | 53.09±19.21 | 9.08±0.74 | 113.33±3.86 | 87.45±2.39 | 77.04±3.02 |
| CA²P | 77.78±11.11 | 81.48±13.58 | 3.80±0.31 | 59.67±5.25 | 92.60±3.23 | 87.30±3.20 |

#### (b) *Cooking Workstation Preparation*

| Method | SR | GC | PSL | NGT | HR | MU |
|--------|-----|-----|-----|-----|-----|-----|
| CaP | 51.85±6.42 | 54.32±2.14 | 13.07±1.09 | 154.02±5.30 | - | 26.31±1.83 |
| LRLL | 55.56±0.00 | 55.56±0.00 | 12.65±1.14 | 148.05±2.04 | - | 29.82±1.28 |
| RAGCache | 37.04±6.42 | 37.04±6.42 | 9.63±1.01 | 120.49±6.39 | 62.05±1.03 | 67.93±3.72 |
| CA²P | 81.48±12.83 | 82.81±10.52 | 2.85±0.54 | 73.27±3.29 | 72.72±3.02 | 75.83±3.20 |

Table 16 expands on Table 2 in Section 4.2 by reporting detailed results across all metrics for each open-domain scenario in real-world deployment. In Table 16(a), CA²P achieves the highest SR, improving by 22.22% over LRLL. In terms of PSL, where lower is better, CA²P reduces it by 2.39× compared to RAGCache. NGT is also significantly reduced by 53.66 tokens relative to RAGCache. Furthermore, CA²P excels in MU, achieving 10.26% higher than RAGCache. In Table 16(b), CA²P achieves the highest SR, improving by $25, 92\%$ over LRLL. In terms of PSL, where lower is better, CA²P reduces it by 3.38× compared to RAGCache. NGT is also significantly reduced by 47.22 tokens relative to RAGCache. Furthermore, CA²P excels in MU, achieving 7.90% higher than RAGCache.

### D.2.1 OFFICE DESK REARRANGEMENT

The first real-world environment is an office desk setup designed around a long-horizon task of cleaning and organizing the workspace. This section presents a detailed description of the environment illustrated in Figure 3. The task is decomposed into three sequential subtasks. In the first subtask, the agent picks up two trash items and places them into the trash bin. The second subtask requires sorting stationery into the top drawer, demonstrating the agent's ability to handle container interactions. Finally, in the third subtask, the agent disposes of the remaining trash into the bin and organizes the leftover stationery into the middle drawer. This sequence evaluates the compositionality of primitive skills such as pick, place, and pull, while also assessing how cached code can be efficiently reused across structurally similar operations. By combining repetitive but slightly varied object interactions, this environment highlights CA²P's capability to synthesize consistent code policy for long-horizon desk-cleaning tasks. Figure 10 provides qualitative images of the actual experiment setup and the execution sequence.

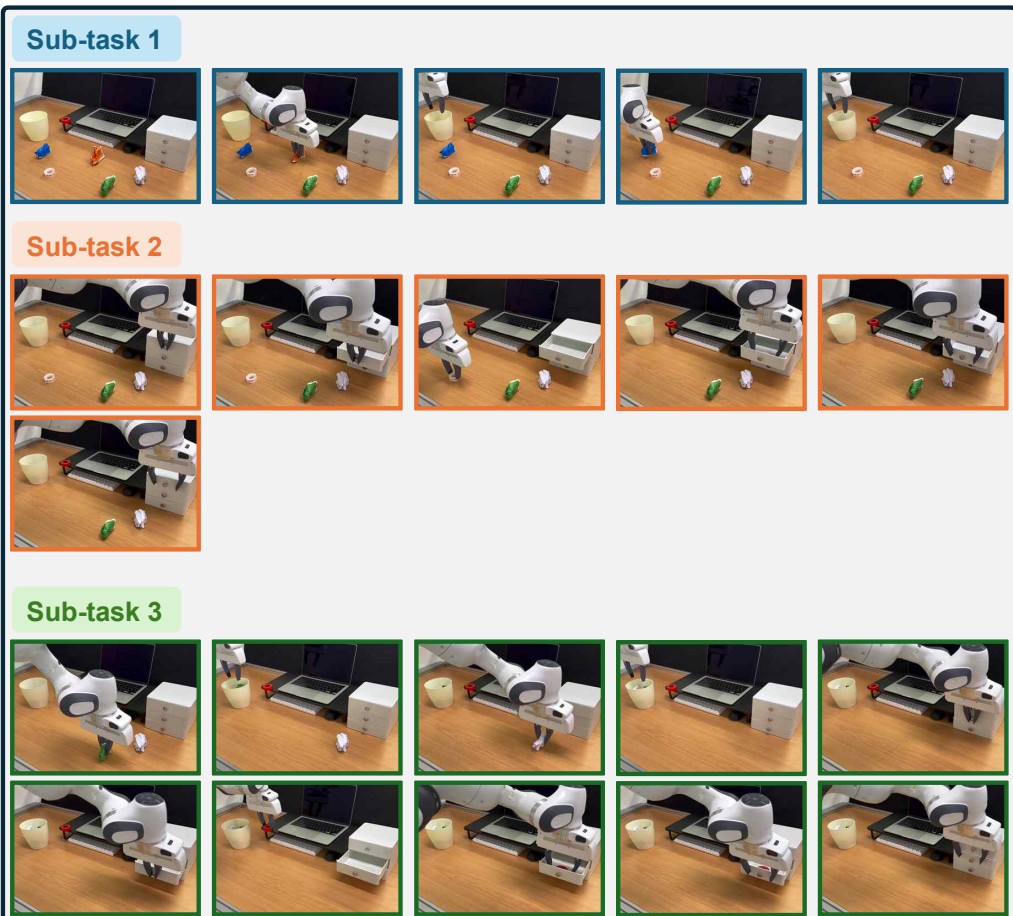

Figure 10: Real-world *Office Desk Rearrangement*. The full task sequence is decomposed into three subtasks: (1) picking up two trash items and throwing them into the bin, (2) organizing stationery into the top drawer, and (3) disposing of remaining trash and placing the leftover stationery into the middle drawer.

### D.2.2 COOKING WORKSTATION PREPARATION

The second real-world environment is a cooking workstation setup. This section provides the detailed description of the environment shown in Figure 1. The task is decomposed into three subtasks. In the first subtask, the agent lifts a portable burner from the floor and places it onto the sink (executed on the desk in our real setup due to space constraints). During this process, the gas hose connecting the burner and the gas canister becomes detached. The second subtask requires the agent to press the emergency gas shutoff switch to stop further leakage. Since this step should be performed quickly to reduce the impact of the leak, the latency of code policy synthesis is critical. Our framework was able to generate and execute the necessary code with low latency, resulting in a faster response compared to baselines. In the final subtask, the agent reconnects the detached hose to the gas canister and toggles the emergency switch again to restore the gas flow, leaving the burner ready for use. This environment demonstrates how $CA^2P$'s efficient code reuse and rapid synthesis enable timely adaptation to unexpected state changes, which in turn contributes to comparatively safer outcomes in practice. Figure 11 illustrates the entire execution sequence.

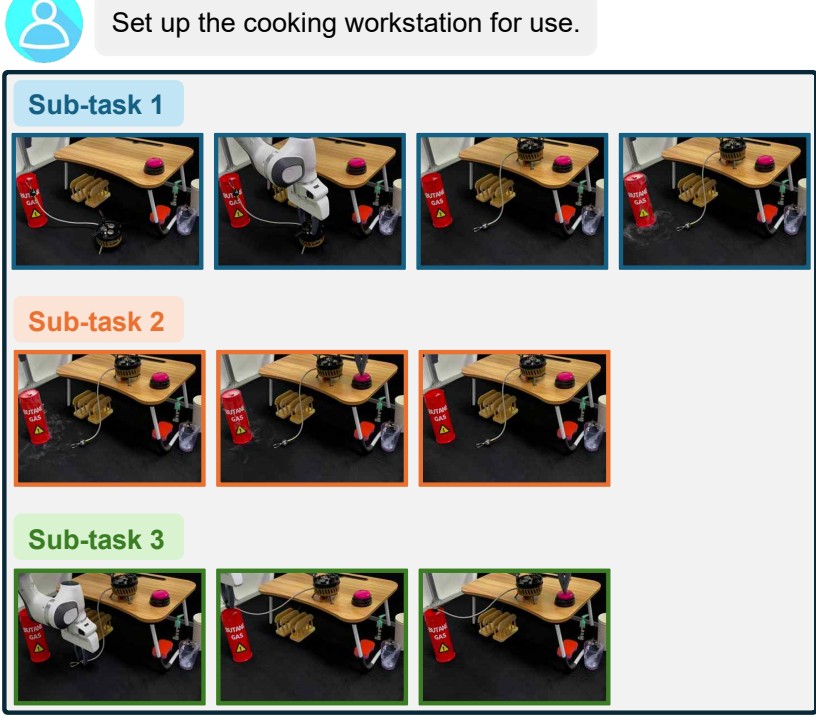

Figure 11: Real-world *Cooking Workstation Preparation*. The sequence involves three subtasks: (1) placing the burner onto the sink (desk in real setup), during which the gas hose disconnects; (2) pressing the emergency gas shutoff switch to quickly stop the leak; and (3) reconnecting the hose and toggling the switch to restore gas flow.

### D.3 ANALYSIS ON CODE CACHE WARM-UP

Figure 12 presents the complete version of Figure 4 in Section 4.3, showing the bootstrapping performance of the code cache over a stream of 40 open-domain tasks. This experiment evaluates how the function-level KV caching performs when initialized with an empty cache and progressively populated through task execution. In Figure 12(a), $CA^2P$ consistently improves and maintains a higher SR, while Figure 12(b) shows a rapid reduction in PSL compared to other baselines. It also maintains a higher HR than RAGCache, as shown in Figure 12(c), and achieves more efficient MU, as shown in Figure 12(d), demonstrating the effectiveness of our code cache management strategy. These results indicate that $CA^2P$ enables stable and efficient operation throughout the warm-up process, supporting consistent task performance via reliable bootstrapping within available resources.

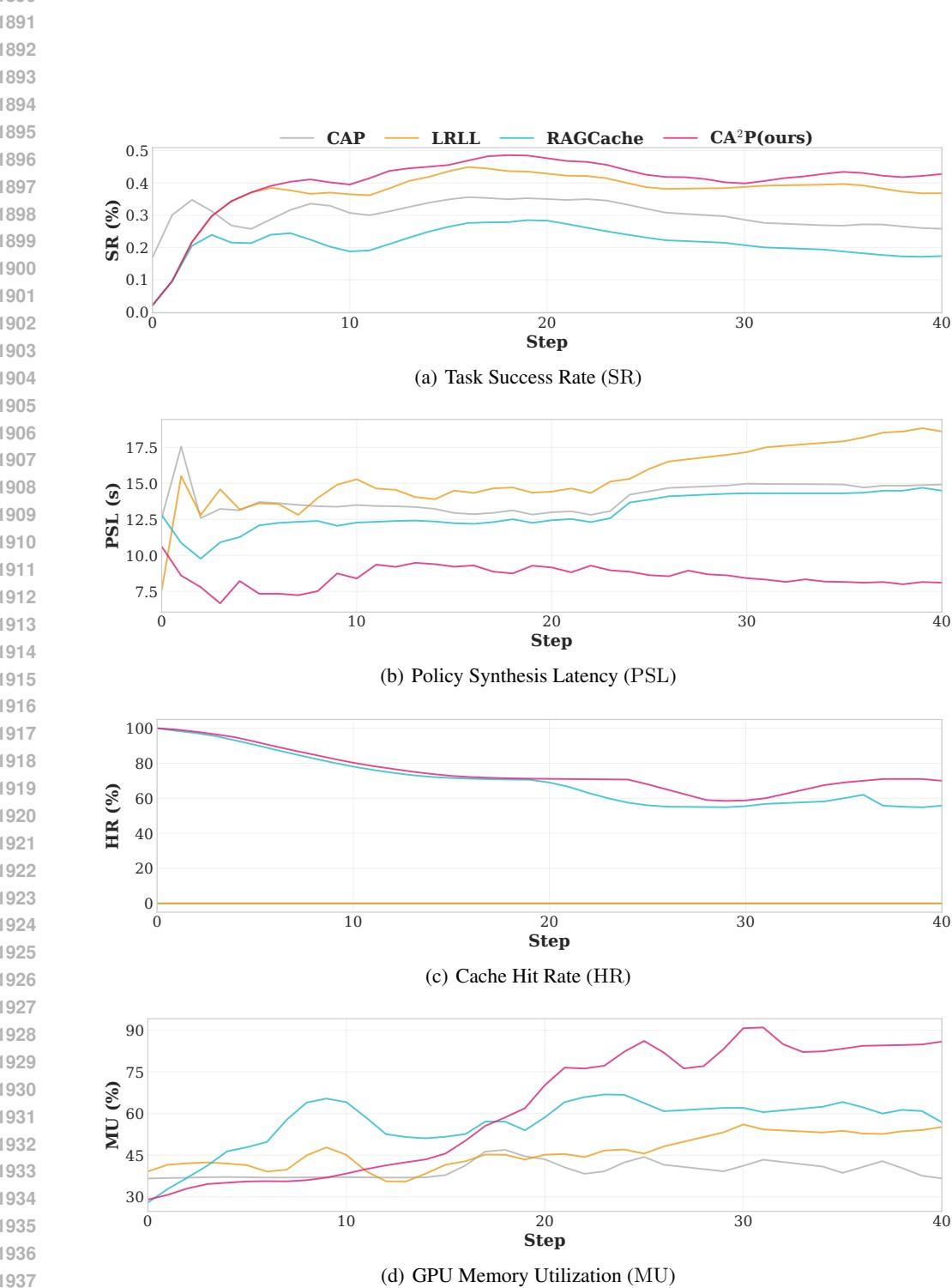

(a) Task Success Rate (SR)

(b) Policy Synthesis Latency (PSL)

(c) Cache Hit Rate (HR)

(d) GPU Memory Utilization (MU)

Figure 12: Analysis on code cache warm-up, with SR, PSL, HR, and MU over 40 tasks

## D.4  ANALYSIS ON BEHAVIOR CONSISTENCY

Table 17: Continual adaptation performance on an open-domain scenario

| **RLBench** | *Initial* (after task IDs 1–9) | | | | | | |
|---|---|---|---|---|---|---|---|
| Method | SR | FWT | BWT | PSL | NGT | HR | MU |
| CaP | 33.33±0.00 | 0.00±0.00 | 0.00±0.00 | 9.34±0.06 | 112.67±0.00 | - | 24.86±0.00 |
| LRLL | 33.33±0.00 | -6.25±0.00 | 5.56±0.00 | 18.42±0.42 | 223.44±0.00 | - | 25.09±0.00 |
| RAGCache | 33.33±0.00 | 0.00±0.00 | 0.00±0.00 | 9.76±0.18 | 115.33±0.00 | 22.00±0.00 | 77.56±0.00 |
| CA$^2$P | **44.44±0.00** | 2.08±0.00 | **0.00±0.00** | 4.19±0.03 | 222.50±0.71 | **33.33±0.00** | **74.31±0.00** |

| **RLBench** | *Middle* (after task IDs 10–18) | | | | | | |
|---|---|---|---|---|---|---|---|
| Method | SR | FWT | BWT | PSL | NGT | HR | MU |
| CaP | 27.78±0.00 | 0.00±0.00 | 0.00±0.00 | 9.40±0.12 | 112.78±0.00 | - | 16.58±0.00 |
| LRLL | 33.33±0.00 | 0.0±0.0 | 5.56±0.00 | 17.92±0.20 | 217.11±0.00 | - | 25.48±0.00 |
| RAGCache | 33.33±0.00 | 0.00±0.00 | 0.00±0.00 | 9.35±0.17 | 111.78±0.00 | 11.00±0.00 | 77.47±0.00 |
| CA$^2$P | **38.89±0.00** | 4.76±0.00 | **0.00±0.00** | 3.36±0.13 | 194.50±79.90 | **41.67±3.92** | **74.26±2.30** |

| **RLBench** | *Final* (after task IDs 19–25) | | | | | | |
|---|---|---|---|---|---|---|---|
| Method | SR | FWT | BWT | PSL | NGT | HR | MU |
| CaP | 37.33±1.88 | - | 0.00±0.00 | 9.40±0.47 | 112.95±4.01 | - | 47.41±8.38 |
| LRLL | 40.00±0.00 | - | 4.00±0.00 | 23.34±0.33 | 274.22±0.00 | - | 43.14±0.00 |
| RAGCache | 36.44±0.00 | - | −2.00±2.83 | 8.79±0.08 | 105.22±0.00 | 50.00±0.00 | 77.60±0.00 |
| CA$^2$P | **46.00±2.83** | - | **2.0±2.83** | 6.61±1.78 | 188.50±60.10 | **61.91±6.74** | **80.38±5.69** |

Table 17 shows snapshot evaluations of the code cache at the *Initial*, *Middle*, and *Final* phases during a continual task stream of 25 tasks, providing the detailed results with all metrics corresponding to Table 3 in Section 4.3. At each phase, the entire task set is re-evaluated while preserving the current state of the code cache to assess the behavioral consistency of the accumulated code, reporting CSIM, FWT, BWT, and SR. The results show that CA$^2$P not only improves SR but also maintains consistent code structure across tasks; it achieves positive FWT across phases without forgetting (non-negative BWT). This demonstrates that CA$^2$P supports consistent behavior by transferring function-level knowledge in open-domain tasks.

## D.5  ABLATION ON MODEL SIZE AND CACHE-REFACTORING

Table 18: Ablation on model size and cache-refactoring. Evaluation of CodeLLMs with different scales (3B, 7B, 14B), contrasting cache-refactoring against ablated settings: cache-refactoring without CoT (execution feedback only) and refactoring with direct human expert guidance.

| Setting | Model Size | SR | GC | PSL | NGT | HR | MU |
|---|---|---|---|---|---|---|---|
| W/O GUIDANCE | 3B | 24.67 | 35.54 | 2.71 | 121.14 | 66.27 | 49.71 |
| | 7B | 40.12 | 50.74 | 3.37 | 113.03 | 71.12 | 67.07 |
| | 14B | 45.80 | 60.63 | 4.48 | 103.00 | 74.50 | 82.63 |
| COT GUIDANCE | 3B | 26.57 | 38.36 | 2.90 | 115.66 | 66.84 | 53.43 |
| | 7B | 44.50 | 58.14 | 3.59 | 90.69 | 72.45 | 68.90 |
| | 14B | 53.75 | 66.54 | 4.53 | 104.96 | 74.56 | 84.86 |
| EXPERT GUIDANCE | 3B | 28.36 | 39.74 | 3.19 | 127.01 | 68.28 | 52.34 |
| | 7B | 46.69 | 59.15 | 3.89 | 123.15 | 71.96 | 68.97 |
| | 14B | 55.09 | 67.92 | 4.74 | 112.01 | 74.65 | 84.64 |

Table 18 provides the numerical values of the data plotted in Figure 5. It compares CA$^2$P across different CodeLLM sizes and cache-refactoring configurations. We evaluate the default setting (CoT-guided cache-refactoring) against two ablated variants: without CoT and with expert guidance replaced. As model size increases from 3B to 14B, CA$^2$P consistently achieves higher SR (26.57%, 44.50%, 53.75%) and GC (38.36%, 58.14%, 66.54%), while PSL (2.90$s$, 3.59$s$, 4.53$s$) also increases

due to the larger model size. MU and HR remain relatively stable, indicating that $CA^2P$ effectively balances performance and efficiency across model scales. For the ablation on cache-refactoring, using only execution feedback results in a noticeable drop in SR and GC, with slight reductions in PSL and NGT compared to the default setting. This highlights the efficiency of CoT-guided cache-refactoring. When replacing CoT with direct expert guidance, it achieves slightly higher SR and GC due to access to sufficient feedback, although the CoT-based approach remains highly competitive.

### D.6  ABLATION ON MODEL CHOICE

Table 19: Ablation on model choice

| Family | Method | CodeLLM | SR | PSL | NGT | HR | MU |
|---|---|---|---|---|---|---|---|
| QWEN2.5 | $CA^2P$ | ✔ | 29.62±0.01 | 2.89±0.41 | 106.67±1.70 | 58.63±1.28 | 70.40±2.30 |
| | $CA^2P$ | ✘ | 26.38±0.23 | 3.12±0.43 | 158.67±1.25 | 47.12±3.94 | 68.03±2.97 |
| | CaP | ✔ | 23.63±0.16 | 7.94±0.47 | 110.32±3.40 | - | 12.93±0.49 |
| GEMMA | $CA^2P$ | ✔ | 28.83±0.05 | 2.84±0.86 | 125.67±11.09 | 52.46±3.28 | 68.32±0.32 |
| | $CA^2P$ | ✘ | 26.39±0.03 | 3.05±0.13 | 144.23±0.03 | 41.67±0.05 | 65.94±0.12 |
| | CaP | ✔ | 22.22±0.02 | 8.72±0.49 | 138.49±3.53 | - | 13.76±1.67 |
| LLAMA2 | $CA^2P$ | ✔ | 31.12±1.32 | 3.32±0.09 | 151.67±3.77 | 57.91±4.27 | 67.99±0.92 |
| | $CA^2P$ | ✘ | 25.33±1.07 | 3.46±0.26 | 168.03±4.01 | 52.85±1.29 | 62.03±0.88 |
| | CaP | ✔ | 17.83±2.46 | 11.48±0.23 | 156.40±3.49 | - | 11.20±0.30 |
| DEEPSEEK | $CA^2P$ | ✔ | 30.23±5.71 | 2.33±0.34 | 130.00±17.34 | 67.23±3.28 | 71.09±2.03 |
| | $CA^2P$ | ✘ | 23.04±3.21 | 2.64±0.86 | 134.00±1.73 | 61.34±3.42 | 68.03±2.83 |
| | CaP | ✔ | 18.98±0.41 | 13.07±1.13 | 146.00±3.88 | - | 13.28±0.24 |

Table 19 extends Table 4 in Section 4.3 by reporting additional results, and presents the performance of $CA^2P$ applied with four different LLM families, contrasting CodeLLMs with general-purpose LLMs. The results show that $CA^2P$ equipped with CodeLLMs consistently outperforms its counterparts based on the LLMs, as well as CaP. On average, it achieves a $4.67\%$ higher SR, a $1.08\times$ reduction in PSL, and an $8.31\%$ higher HR compared to variants using the LLMs, while maintaining a more favorable trade-off than CaP. These findings demonstrate that $CA^2P$ is not restricted to a specific LLM architecture and remains broadly compatible with diverse CodeLLMs.

### D.7  ABLATION ON $CA^2P$

Table 20: Ablation on $CA^2P$. 'w/o.' denotes the removal of a component, and '→' indicates replacement with an alternative operation

| Method | SR | PSL | NGT | HR | MU |
|---|---|---|---|---|---|
| $CA^2P$ | 45.91±4.37 | 3.24±0.65 | 105.62±3.09 | 73.65±2.15 | 83.63±3.12 |
| w/o. $\mathcal{I}$ | 38.64±1.55 | 4.12±0.12 | 114.50±0.70 | 66.78±3.12 | 76.04±2.38 |
| w/o. $\mathcal{C}$ | 34.37±0.52 | 3.33±0.23 | 88.67±2.51 | 67.56±3.51 | 72.93±1.04 |
| w/o. $\mathcal{H}$ | 34.64±1.98 | 4.25±1.27 | 113.00±7.81 | - | 25.83±2.12 |
| w/o. $\ell_{\text{sema}}$ | 35.02±1.76 | 3.74±0.29 | 116.20±2.36 | 48.61±1.96 | 74.74±2.09 |
| → Retrieval | 41.92±0.72 | 4.42±0.48 | 117.39±3.67 | 43.06±2.45 | 46.72±2.49 |
| → Regenerate | 35.77±1.76 | 6.89±0.20 | 161.00±4.24 | 71.39±2.83 | 77.23±1.82 |

Table 20 presents additional experimental results that complement Table 5 in Section 4.3, and analyzes the contribution of each component of $CA^2P$ to overall performance. For the code cache structure, removing the Function-Interface tier $\mathcal{I}$ (denoted as w/o. $\mathcal{I}$), the Function-Code tier $\mathcal{C}$ (w/o. $\mathcal{C}$), or the entire cache $\mathcal{H}$ (w/o. $\mathcal{H}$) leads to a substantial drop in SR, confirming that the hierarchical design is crucial for reducing reasoning overhead and enabling reliable policy synthesis. In particular, removing $\mathcal{C}$ (w/o. $\mathcal{C}$) disables direct code reuse, leading to invalid policy edits that yield the lowest SR despite the shortest PSL. Removing the semantic term in the locality score (w/o. $\ell_{\text{sema}}$) reduces long-horizon stability, indicating that semantic diversity in the code cache is critical

for open-domain tasks. Replacing perplexity with a retriever ($\rightarrow$ Retrieval) yields comparable SR, but perplexity is preferable as it requires no additional encoder and achieves more efficient PSL and HR. Replacing cache-refactoring with full regeneration ($\rightarrow$ Regeneration) increases PSL and decreases SR, highlighting the efficiency and reliability of cache-refactoring.

## D.8   ABLATION ON THE CONTRIBUTION OF EACH LOCALITY COMPONENT

Table 21: Ablation of locality across heterogeneous task regimes

| **RLBench** | *Repetitive* | | | *Order-dependent* | | | *Semantic diverse* | | |
|---|---|---|---|---|---|---|---|---|---|
| Methods | SR ($\uparrow$) | PSL ($\downarrow$) | HR ($\uparrow$) | SR ($\uparrow$) | PSL ($\downarrow$) | HR ($\uparrow$) | SR ($\uparrow$) | PSL ($\downarrow$) | HR ($\uparrow$) |
| $\alpha = 1$ | 36.43 | 3.34 | 67.04 | 34.34 | 3.32 | 61.04 | 35.12 | 3.40 | 60.23 |
| $\beta = 1$ | 33.32 | 3.43 | 56.04 | 37.20 | 3.33 | 69.32 | 33.92 | 3.92 | 54.95 |
| $\gamma = 1$ | 34.42 | 3.30 | 61.26 | 33.33 | 3.14 | 56.58 | 38.12 | 3.60 | 68.29 |

Table 21 presents an extended ablation designed to isolate the contribution of each locality component. For each setting, one component is assigned a weight of 1 while the remaining components are set to 0 (e.g., $(\alpha, \beta, \gamma) = (1, 0, 0)$). The evaluation is performed on open-domain task streams constructed to emphasize distinct structural properties, and three such task-stream types are considered in this analysis. **Repetitive task streams** consist of tasks in which similar operations appear repeatedly across episodes. In such settings, the dominant signal is the recurrence of individual functions, and prioritizing entries by invocation frequency is sufficient; accordingly, $\ell_{\text{freq}}$ alone captures the structure of these streams and yields the strongest performance. In contrast, **order-dependent task streams** require functions to be executed in specific sequences, and correctness depends on consistent ordering relations (e.g., inspect before transform). The key structure lies in conditional co-occurrence rather than raw frequency, making $\ell_{\text{asso}}$ the most effective component for maintaining useful function transitions. Finally, **semantically diverse task streams** include tasks with large functional variation or uncommon operations, where rare but semantically distinct functions become important. Preserving diversity in the cache is therefore critical, and $\ell_{\text{sema}}$ alone best supports this requirement by prioritizing functions with high semantic uniqueness.

## D.9   ABLATION OF PRE-POPULATION CONTENTS AND CACHE-REFACTORING

Table 22: Ablation of pre-population content influence under KV cache enabled vs. KV cache disabled settings

| **RLBench** | *Program-level* | | | | *Function-level* | | | |
|---|---|---|---|---|---|---|---|---|
| Methods | SR ($\uparrow$) | GC ($\uparrow$) | PSL ($\downarrow$) | HR ($\uparrow$) | SR ($\uparrow$) | GC ($\uparrow$) | PSL ($\downarrow$) | HR ($\uparrow$) |
| Non KV Cache | 44.44 | 53.33 | 8.04 | 74.50 | 77.78 | 77.78 | 6.21 | 87.25 |
| KV Cache | 66.67 | 71.11 | 4.52 | 74.50 | 88.89 | 88.89 | 3.80 | 87.25 |

The experiment is designed to show that the framework enables cache-refactoring of previously generated functions and supports low-latency synthesis without full regeneration. The purpose is not to indicate reliance on pre-population, but to demonstrate that even when the cache originates from a different domain, the system can efficiently modify and compose cached functions to achieve reliable execution in real-world conditions. To further analyze this capability, we conduct a controlled comparison between two types of stored code: (1) **program-level** code memory composed of monolithic code blocks, and (2) **function-level** code memory composed of decomposed, reusable functional units. For each type, we evaluate two variants-one that maintains the stored code in KV cache form and one that does not. All cached content originates from the simulation domain, and no real-world–specific information is preloaded. Across all four settings, the function-level memory with KV cache maintenance consistently shows higher task success, more stable execution behavior, and lower synthesis latency. These results indicate that fine-grained function-level KV caching, together with cache-refactoring, is crucial for fast and reliable adaptation in open-domain.

### D.10 Ablation on model quantization.

Table 23: Ablation of model quantization

| **RLBench** | *Open-Composition* | | *Open-Perturbation* | | *Open-Evolution* | |
|---|---|---|---|---|---|---|
| Methods | SR ($\uparrow$) | PSL ($\downarrow$) | SR ($\uparrow$) | PSL ($\downarrow$) | SR ($\uparrow$) | PSL ($\downarrow$) |
| 4bit quantization | 40.32 | 2.53 | 36.05 | 2.58 | 30.33 | 3.21 |
| 8bit quantization | **45.91** | **3.24** | **42.82** | **3.47** | **33.98** | **4.39** |

Model quantization (Wolf et al., 2019) is largely complementary to our framework and can be applied alongside it to further improve efficiency. We additionally provide experimental results combining our method with quantization in the analysis below. These results further highlight that our framework pursues a different objective from conventional LLM serving while remaining fully compatible with it.

