# OpenReview forum: "Ca$^2$P: Cache-Augmented Code-as-Policies for Open-Domain Embodied Tasks"
_ICLR.cc/2026/Conference — Submitted to ICLR 2026_

### Official Review · Reviewer_cxo5 · 2025-10-31

**Soundness:** 3
**Presentation:** 4
**Contribution:** 1
**Rating:** 2
**Confidence:** 4

**Summary:**

This paper proposes a caching mechanism for the code as policies base. At a high-level, a coding LLM is used to write composable functions for executing manipulation tasks given a set of perception and control APIs (the idea previously in code as policies). In this work, the authors propose to augment it with an explicit cache across domains, which store separately the function description and the function implementation. At deployment time, functions are accumulated from successful runs and reused later in other tasks by referencing the stored function description.

**Strengths:**

- The paper is written with clarity in general and easy to follow.
- The empirical results are extensive across several evaluation domains.

**Weaknesses:**

- The contribution of this work appears to solely focus on the cache management for code generation (though for applications in embodied tasks), which has been very well understood and explored in both research and commercial systems such as coding agents. As a result, it is questionable whether the contribution is meaningful enough for ICLR community. Importantly, the major benefit in the context of embodied tasks appears to be mainly efficiency gain for LLM calls (which typically happens before any robot execution). Although it may be argued that this is important for certain tasks (such as that in figure 1), the scope remains limited and there are many ways to improve efficiency that has been widely deployed in existing tool boxes (such as token caching, model quantization).
- The open-sourced implementation of code as policies already contains a caching mechanism, albeit within the per-episode execution. It would enhance the paper if it can made further clearer the differences in the proposed caching mechanism to the existing one. Notably, what advantages does it offer?

**Questions:**

See weaknesses section above.

---

> ### Author Response · Authors · 2025-11-18
> **Response to Weakness 1**
>
> Dear Reviewer cxo5,
>
> We appreciate the reviewer’s insightful comments. Before addressing the specific points raised, we would like to clarify the terminology to ensure we are aligned. Throughout our paper, we consistently use the terms KV cache, KV states, attention computation, and attention states, and provide appropriate citations in both the main text and the Appendix, beginning from the Abstract and Introduction where the concept is first introduced.
>
> If the reviewer had a broader class of caching mechanisms in mind, we would be glad to clarify and discuss this further during the discussion period. If so, we would greatly appreciate any pointers or citations to the specific caching mechanisms the reviewer is referring to. For now, our response proceeds under the understanding that the “caching’’ mentioned in the reviewer's comment aligns with KV caching, which is the mechanism explicitly defined in prior works [1,2,3,4,5,6] and employed in our work.
>
> ---
>
> > **Weakness 1**: The contribution of this work appears to solely focus on the cache management for code generation (though for applications in embodied tasks), which has been very well understood and explored in both research and commercial systems such as coding agents. As a result, it is questionable whether the contribution is meaningful enough for ICLR community. Importantly, the major benefit in the context of embodied tasks appears to be mainly efficiency gain for LLM calls (which typically happens before any robot execution). Although it may be argued that this is important for certain tasks (such as that in figure 1), the scope remains limited and there are many ways to improve efficiency that has been widely deployed in existing tool boxes (such as token caching, model quantization).
>
> We would like to clarify that although our work leverages cache management techniques within the context of code generation, improving the efficiency of LLM calls is not the primary objective. Instead, our goal is to develop a framework that enables code agents-specifically Code-as-Policies(CaP) [7]-to utilize memory-based mechanisms in an efficient and effective manner, thereby allowing them to operate reliably in open-domain embodied tasks.
>
> Open-domain embodied tasks are characterized by continual and unpredictable variations in both instructions and environmental states. These conditions require agents not only to reuse previously successful solutions but also to rapidly adapt them to new situations. Recent work on embodied control further investigates time-sensitive scenarios where decoupled progression between the agent’s actions and the environment’s dynamics leads to stale observations and outdated context when inference latency accumulates [8,9,10]. These studies emphasize that agents must provide fast reflexive responses while also performing reflective refinements to remain reliable under rapidly changing conditions. To support such behavior within the CaP paradigm, the memory must store executable modules rather than raw text, be organized in a library-like structure for compositional reuse, and remain both immediately accessible and modifiable during ongoing task execution.
>
> To meet these memory requirements for embodied agents, we repurpose the existing KV caching mechanism to support function-level code reuse, which serves as the foundation of our cache-augmented code policy synthesis. This design allows the CodeLLM-based agent to respond effectively to unpredictable situations, as cached functions remain both immediately executable and easily editable throughout task execution.
>
> Our function-level KV caches preserve the modular and executable nature of function code, making them well suited for embodied settings. At the same time, they enable rapid adaptation by allowing the CodeLLM to edit decomposed sub-policies directly within their KV states, thereby supporting task- and environment-specific modifications without requiring full regeneration.
>
> This approach is fundamentally different from existing KV cache management techniques for code generation as well as from conventional LLM serving methods. Below, we further clarify these differences through both conceptual analysis and experimental evidence.

---

> ### Author Response · Authors · 2025-11-18
> **Response to Weakness 1**
>
> **Comparison with existing KV cache management approaches.** Existing KV cache management approaches for code generation, such as PromptCache [11], primarily focus on reusing KV states to reduce attention computation during inference, which is particularly effective before task execution. While these methods accelerate decoding, they do not preserve executable and reusable program structures, nor do they support modifying cached functions during execution. PromptCache introduces a modular caching mechanism for code generation, but it is not designed to operate under continually changing tasks such as open-domain embodied tasks. Moreover, its compositional capabilities rely on user-specified structures rather than enabling scalable and autonomous agentic control. As shown in Table 1 of the main text, we include PromptCache as a baseline to empirically demonstrate these differences.
>
> | **ALFRED** | *Open-Composition* ||| *Open-Perturbation* ||| *Open-Evolution* |||
> |-----------|----------------------|------|------|----------------------|------|------|----------------------|------|------|
> | **Methods** | **SR (↑)** | **PSL (↓)** | **Rank (↓)** | **SR (↑)** | **PSL (↓)** | **Rank (↓)** | **SR (↑)** | **PSL (↓)** | **Rank (↓)** |
> | PromptCache | 40.37±0.37 | 12.82±0.86 | 4 (0.56) | 36.92±2.22 | 12.93±1.48 | 5 (0.56) | 34.14±0.82 | 13.18±1.15 | 6 (0.54) |
> | **Ca$^{2}$P (ours)** | **61.58±1.86** | **5.82±0.57** | **1 (0.81)** | **57.23±0.23** | **6.32±0.51** | **1 (0.79)** | **55.89±0.85** | **6.29±1.25** | **1 (0.78)** |
>
> | **TEACh** | *Open-Composition* ||| *Open-Perturbation* ||| *Open-Evolution* |||
> |-----------|----------------------|------|------|----------------------|------|------|----------------------|------|------|
> | **Methods** | **SR (↑)** | **PSL (↓)** | **Rank (↓)** | **SR (↑)** | **PSL (↓)** | **Rank (↓)** | **SR (↑)** | **PSL (↓)** | **Rank (↓)** |
> | PromptCache | 36.15±0.86 | 14.88±0.29 | 6 (0.53) | 33.12±0.12 | 14.94±1.31 | 6 (0.51) | 31.07±1.49 | 16.32±0.24 | 6 (0.50) |
> | **Ca$^{2}$P (ours)** | **58.59±0.39** | **5.23±0.09** | **1 (0.79)** | **56.26±1.09** | **5.61±1.83** | **1 (0.78)** | **52.93±0.62** | **7.03±0.82** | **1 (0.76)** |
>
> | **RLBench** | *Open-Composition* ||| *Open-Perturbation* ||| *Open-Evolution* |||
> |-----------|----------------------|------|------|----------------------|------|------|----------------------|------|------|
> | **Methods** | **SR (↑)** | **PSL (↓)** | **Rank (↓)** | **SR (↑)** | **PSL (↓)** | **Rank (↓)** | **SR (↑)** | **PSL (↓)** | **Rank (↓)** |
> | PromptCache | 28.33±1.73 | 8.99±0.39 | 3 (0.50) | 27.41±0.81 | 8.97±0.35 | 3 (0.50) | 25.03±0.85 | 9.12±0.65 | 4 (0.52) |
> | **Ca$^{2}$P (ours)** | **45.91±4.37** | **3.24±0.65** | **1 (0.73)** | **42.82±2.64** | **3.47±0.16** | **1 (0.71)** | **33.98±1.27** | **4.39±0.43** | **1 (0.67)** |

---

> ### Author Response · Authors · 2025-11-18
> **Response to Weakness 1**
>
> **Comparison with existing KV caching for LLM serving.** Similarly, KV caching methods developed for LLM serving (e.g., prefix caching, modular KV reuse, position-independent caching) are designed to increase throughput for static text generation and typically assume that cached states correspond to fixed prompts or documents [12]. These approaches cannot decompose cached contexts into composable functional units, nor can they support in-place editing within KV states, both of which are essential for continual adaptation in open-domain embodied tasks. We therefore include RAGCache [13] and CAG [14] as baselines in Table 1 of the main text to empirically highlight these differences.
>
> | **ALFRED** | *Open-Composition* ||| *Open-Perturbation* ||| *Open-Evolution* |||
> |-----------|----------------------|------|------|----------------------|------|------|----------------------|------|------|
> | **Methods** | **SR (↑)** | **PSL (↓)** | **Rank (↓)** | **SR (↑)** | **PSL (↓)** | **Rank (↓)** | **SR (↑)** | **PSL (↓)** | **Rank (↓)** |
> | CAG | 38.24±0.04 | 12.12±0.62 | 3 (0.57) | 36.03±0.32 | 12.38±0.92 | 4 (0.57) | 32.18±1.12 | 12.39±0.92 | 4 (0.55) |
> | RAGCache | 39.29±1.70 | 13.48±0.93 | 7 (0.55) | 36.33±0.02 | 13.15±0.12 | 6 (0.56) | 33.83±1.53 | 13.02±1.15 | 5 (0.55) |
> | **Ca$^{2}$P (ours)** | **61.58±1.86** | **5.82±0.57** | **1 (0.81)** | **57.23±0.23** | **6.32±0.51** | **1 (0.79)** | **55.89±0.85** | **6.29±1.25** | **1 (0.78)** |
>
> | **TEACh** | *Open-Composition* ||| *Open-Perturbation* ||| *Open-Evolution* |||
> |-----------|----------------------|------|------|----------------------|------|------|----------------------|------|------|
> | **Methods** | **SR (↑)** | **PSL (↓)** | **Rank (↓)** | **SR (↑)** | **PSL (↓)** | **Rank (↓)** | **SR (↑)** | **PSL (↓)** | **Rank (↓)** |
> | CAG | 35.03±2.01 | 14.23±1.28 | 4 (0.53) | 33.89±0.92 | 14.02±0.24 | 2 (0.53) | 30.58±0.33 | 14.82±0.11 | 5 (0.52) |
> | RAGCache | 35.23±0.05 | 14.97±0.15 | 7 (0.52) | 34.08±0.04 | 15.51±0.42 | 7 (0.50) | 29.45±1.23 | 16.07±0.29 | 8 (0.50) |
> | **Ca$^{2}$P (ours)** | **58.59±0.39** | **5.23±0.09** | **1 (0.79)** | **56.26±1.09** | **5.61±1.83** | **1 (0.78)** | **52.93±0.62** | **7.03±0.82** | **1 (0.76)** |
>
> | **RLBench** | *Open-Composition* ||| *Open-Perturbation* ||| *Open-Evolution* |||
> |-----------|----------------------|------|------|----------------------|------|------|----------------------|------|------|
> | **Methods** | **SR (↑)** | **PSL (↓)** | **Rank (↓)** | **SR (↑)** | **PSL (↓)** | **Rank (↓)** | **SR (↑)** | **PSL (↓)** | **Rank (↓)** |
> | CAG | 26.37±2.03 | 8.53±0.83 | 2 (0.50) | 24.89±1.21 | 8.39±0.37 | 2 (0.50) | 22.82±0.73 | 8.08±1.91 | 2 (0.53) |
> | RAGCache | 27.26±1.08 | 9.09±0.63 | 6 (0.49) | 27.03±1.04 | 9.28±0.66 | 5 (0.49) | 23.79±1.20 | 9.02±0.18 | 5 (0.51) |
> | PromptCache | 28.33±1.73 | 8.99±0.39 | 3 (0.50) | 27.41±0.81 | 8.97±0.35 | 3 (0.50) | 25.03±0.85 | 9.12±0.65 | 4 (0.52) |
> | **Ca$^{2}$P (ours)** | **45.91±4.37** | **3.24±0.65** | **1 (0.73)** | **42.82±2.64** | **3.47±0.16** | **1 (0.71)** | **33.98±1.27** | **4.39±0.43** | **1 (0.67)** |

---

> ### Author Response · Authors · 2025-11-18
> **Response to Weakness 1**
>
> **Compatibility with model quantization.** Model quantization [15] is largely complementary to Ca$^{2}$P and can be applied alongside it to further improve efficiency. We additionally provide experimental results combining our method with quantization in the analysis below. These results further highlight that Ca$^{2}$P pursues a different objective from conventional LLM serving while remaining fully compatible with it, and we will include this experiment in Appendix for completeness.
>
> | **RLBench** | *Open-Composition* || *Open-Perturbation* || *Open-Evolution* ||
> |-----------|----------------------|------|----------------------|------|----------------------|------|
> | **Methods** | **SR (↑)** | **PSL (↓)** | **SR (↑)** | **PSL (↓)** | **SR (↑)** | **PSL (↓)** |
> | 4bit quantization | 40.32 | 2.53 | 36.05 | 2.58 | 30.33 | 3.21 |
> | **8bit quantization** | **45.91** | **3.24** | **42.82** | **3.47** | **33.98** | **4.39** |
>
>
> Ca$^{2}$P improves the efficiency and responsiveness of code policy synthesis by leveraging function-level KV caches, which support targeted in-place modifications within KV states without requiring full regeneration.
> These capabilities allow embodied agents to adapt efficiently to unpredictable and dynamic task conditions, which prior cache management and LLM serving techniques are not designed to handle.
>
> [1] Zhang et al., *CaM: Cache Merging for Memory-efficient LLMs Inference.* ICML 2024.
>
> [2] Bavarian et al., *Efficient Training of Language Models to Fill in the Middle.* arXiv 2022.
>
> [3] Lu et al., *TurboRAG: Accelerating Retrieval-Augmented Generation with Precomputed KV Caches for Chunked Text.* EMNLP 2025.
>
> [4] He et al., *Let the Code LLM Edit Itself When You Edit the Code.* ICLR 2025.
>
> [5] Guo et al., *EFIM: Efficient Serving of LLMs for Infilling Tasks with Improved KV Cache Reuse.* arXiv 2025.
>
> [6] Zhao et al., *MPIC: Position-independent Multimodal Context Caching System for Efficient MLLM Serving.* arXiv 2025.
>
> [7] Liang et al., *Code as Policies: Language Model Programs for Embodied Control.* ICRA 2023.
>
> [8] Huang et al. *Dadu-Corki: Algorithm-Architecture Co-Design for Embodied AI-powered Robotic Manipulation.* ISCA 2025.
>
> [9] Zheng et al. *LLM-Enhanced Rapid-Reflex Async-Reflect Embodied Agent for Real-Time Decision-Making in Dynamically Changing Environments.* arXiv 2025.
>
> [10] Zhang et al. *Boosting Embodied AI Agents through Perception-Generation Disaggregation and Asynchronous Pipeline Execution.* arXiv 2025.
>
> [11] Gim et al., *Prompt Cache: Modular Attention Reuse for Low-latency Inference.* MLSys 2024.
>
> [12] Yao et al., *CacheBlend: Fast Large Language Model Serving for RAG with Cached Knowledge Fusion.* EuroSys 2025.
>
> [13] Jin et al., *RAGCache: Efficient Knowledge Caching for Retrieval-augmented Generation.* Transactions on Computer Systems 2024.
>
> [14] Chan et al., *Don't Do RAG: When Cache-Augmented Generation Is All You Need for Knowledge Tasks.* WWW 2025.
>
> [15] Wolf et al., *HuggingFace's Transformers: State-of-the-art Natural Language Processing.* arXiv 2019.

---

> ### Author Response · Authors · 2025-11-18
> **Response to Weakness 2**
>
> > **Weakness 2**: The open-sourced implementation of code as policies already contains a caching mechanism, albeit within the per-episode execution. It would enhance the paper if it can made further clearer the differences in the proposed caching mechanism to the existing one. Notably, what advantages does it offer?
>
> As the reviewer insightfully pointed out, the KV caching mechanism in the open-sourced Code-as-Policies implementation is limited to per-episode reuse, where KV states are retained only within a single code-generation session. This mechanism helps avoid redundant attention computation for the current episode and is therefore useful primarily before execution. However, this KV caching mechanism does not persist across tasks, which is a critical limitation in open-domain settings where future environments and instructions are unpredictable. It does not store executable functional units, and it cannot revise sub-policies within an ongoing code policy without regenerating the entire program.
>
> In contrast, our function-level KV caching mechanism is designed to meet the memory requirements of open-domain embodied agents operating over long task streams, where the task sequence is not predetermined and can change unpredictably. We introduce function-level KV caches that (i) persist validated code functions across tasks, (ii) store them as modular and executable units in the form of KV states, and (iii) support in-place modification through cache-refactoring directly within those KV states. These capabilities allow the agent to reuse previously successful sub-policies and adapt them rapidly to new or changing conditions without full regeneration. To highlight this distinction, we include baselines in Table 1 of the main text, such as Code-as-Policies (CaP) [1] and Structured Chain-of-Thought (SCoT) [2], which rely solely on per-episode KV caching.
>
> | **ALFRED** | *Open-Composition* ||| *Open-Perturbation* ||| *Open-Evolution* |||
> |-----------|----------------------|------|------|----------------------|------|------|----------------------|------|------|
> | **Methods** | **SR (↑)** | **PSL (↓)** | **Rank (↓)** | **SR (↑)** | **PSL (↓)** | **Rank (↓)** | **SR (↑)** | **PSL (↓)** | **Rank (↓)** |
> | CaP | 46.48±1.87 | 16.21±1.03 | 8 (0.53) | 42.23±0.12 | 16.15±0.23 | 8 (0.53) | 37.22±1.72 | 16.28±1.20 | 8 (0.50) |
> | SCoT | 47.93±0.87 | 30.02±3.38 | 10 (0.27) | 43.46±1.06 | 32.39±1.12 | 10 (0.23) | 39.00±0.98 | 32.04±2.10 | 10 (0.22) |
> | **Ca$^{2}$P (ours)** | **61.58±1.86** | **5.82±0.57** | **1 (0.81)** | **57.23±0.23** | **6.32±0.51** | **1 (0.79)** | **55.89±0.85** | **6.29±1.25** | **1 (0.78)** |
>
> | **TEACh** | *Open-Composition* ||| *Open-Perturbation* ||| *Open-Evolution* |||
> |-----------|----------------------|------|------|----------------------|------|------|----------------------|------|------|
> | **Methods** | **SR (↑)** | **PSL (↓)** | **Rank (↓)** | **SR (↑)** | **PSL (↓)** | **Rank (↓)** | **SR (↑)** | **PSL (↓)** | **Rank (↓)** |
> | CaP | 43.92±0.81 | 17.29±0.73 | 5 (0.53) | 42.21±0.28 | 17.23±0.39 | 5 (0.51) | 38.07±0.12 | 17.03±1.20 | 4 (0.52) |
> | SCoT | 44.03±1.47 | 33.00±0.91 | 9 (0.28) | 42.49±1.02 | 33.94±1.92 | 9 (0.23) | 37.91±0.41 | 35.03±1.58 | 9 (0.22) |
> | **Ca$^{2}$P (ours)** | **58.59±0.39** | **5.23±0.09** | **1 (0.79)** | **56.26±1.09** | **5.61±1.83** | **1 (0.78)** | **52.93±0.62** | **7.03±0.82** | **1 (0.76)** |
>
> | **RLBench** | *Open-Composition* ||| *Open-Perturbation* ||| *Open-Evolution* |||
> |-----------|----------------------|------|------|----------------------|------|------|----------------------|------|------|
> | **Methods** | **SR (↑)** | **PSL (↓)** | **Rank (↓)** | **SR (↑)** | **PSL (↓)** | **Rank (↓)** | **SR (↑)** | **PSL (↓)** | **Rank (↓)** |
> | CaP | 31.23±3.32 | 11.39±0.53 | 8 (0.45) | 30.46±0.02 | 11.23±0.93 | 7 (0.46) | 26.84±1.20 | 11.23±0.24 | 7 (0.48) |
> | SCoT | 32.32±2.22 | 22.03±2.03 | 9 (0.20) | 30.32±0.28 | 21.37±1.84 | 9 (0.22) | 27.11±0.37 | 22.83±1.42 | 9 (0.21) |
> | **Ca$^{2}$P (ours)** | **45.91±4.37** | **3.24±0.65** | **1 (0.73)** | **42.82±2.64** | **3.47±0.16** | **1 (0.71)** | **33.98±1.27** | **4.39±0.43** | **1 (0.67)** |

---

> ### Author Response · Authors · 2025-11-18
> **Response to Weakness 2**
>
> In addition, as summarized in Appendix A of our paper, prior works that extend Code-as-Policies with memory also aim to share code across tasks, but they typically store code in textual form, which is neither composable nor modifiable within KV states, and they use it only as additional context for generation. These approaches do not provide immediately executable function-level representations, nor do they support reusing or editing code within KV states. As a consequence, they continue to rely on per-episode KV caching and cannot achieve the level of efficiency or responsiveness needed for open-domain tasks, where targeted in-place modification without full regeneration offers clear advantages. For this reason, we include prior memory-based code-agent approaches, such as HELPER [3], LRLL [4], and PromptBook [5], as baselines in Table 1 of the main text.
>
> | **ALFRED** | *Open-Composition* ||| *Open-Perturbation* ||| *Open-Evolution* |||
> |-----------|----------------------|------|------|----------------------|------|------|----------------------|------|------|
> | **Methods** | **SR (↑)** | **PSL (↓)** | **Rank (↓)** | **SR (↑)** | **PSL (↓)** | **Rank (↓)** | **SR (↑)** | **PSL (↓)** | **Rank (↓)** |
> | HELPER | 53.83±0.39 | 16.82±1.20 | 6 (0.55) | 52.49±0.24 | 15.94±0.69 | 3 (0.58) | 47.75±1.08 | 16.42±0.92 | 3 (0.55) |
> | LRLL | 56.28±3.20 | 16.02±2.02 | 2 (0.58) | 54.70±0.87 | 16.40±1.24 | 2 (0.59) | 51.60±0.30 | 16.63±1.03 | 2 (0.57) |
> | PromptBook | 53.94±3.96 | 31.34±1.82 | 9 (0.27) | 52.63±1.09 | 33.29±1.52 | 9 (0.26) | 50.49±0.51 | 33.49±2.07 | 9 (0.25) |
> | **Ca$^{2}$P (ours)** | **61.58±1.86** | **5.82±0.57** | **1 (0.81)** | **57.23±0.23** | **6.32±0.51** | **1 (0.79)** | **55.89±0.85** | **6.29±1.25** | **1 (0.78)** |
>
> | **TEACh** | *Open-Composition* ||| *Open-Perturbation* ||| *Open-Evolution* |||
> |-----------|----------------------|------|------|----------------------|------|------|----------------------|------|------|
> | **Methods** | **SR (↑)** | **PSL (↓)** | **Rank (↓)** | **SR (↑)** | **PSL (↓)** | **Rank (↓)** | **SR (↑)** | **PSL (↓)** | **Rank (↓)** |
> | HELPER | 47.14±0.63 | 17.31±1.08 | 3 (0.55) | 44.05±0.29 | 17.39±1.02 | 4 (0.52) | 40.00±0.38 | 17.48±0.71 | 3 (0.52) |
> | LRLL | 48.99±1.73 | 17.82±0.82 | 2 (0.55) | 46.78±0.91 | 17.93±1.22 | 3 (0.52) | 41.85±0.02 | 18.03±0.39 | 2 (0.52) |
> | PromptBook | 47.08±2.24 | 36.92±1.47 | 10 (0.24) | 44.85±1.00 | 35.04±1.24 | 10 (0.22) | 41.07±0.32 | 36.82±0.49 | 10 (0.21) |
> | **Ca$^{2}$P (ours)** | **58.59±0.39** | **5.23±0.09** | **1 (0.79)** | **56.26±1.09** | **5.61±1.83** | **1 (0.78)** | **52.93±0.62** | **7.03±0.82** | **1 (0.76)** |
>
> | **RLBench** | *Open-Composition* ||| *Open-Perturbation* ||| *Open-Evolution* |||
> |-----------|----------------------|------|------|----------------------|------|------|----------------------|------|------|
> | **Methods** | **SR (↑)** | **PSL (↓)** | **Rank (↓)** | **SR (↑)** | **PSL (↓)** | **Rank (↓)** | **SR (↑)** | **PSL (↓)** | **Rank (↓)** |
> | HELPER | 35.42±1.32 | 11.48±0.23 | 7 (0.47) | 34.84±0.03 | 11.38±0.06 | 6 (0.48) | 31.12±1.08 | 12.16±1.42 | 8 (0.48) |
> | LRLL | 36.11±0.00 | 10.83±0.82 | 4 (0.49) | 33.82±0.02 | 12.01±0.11 | 8 (0.46) | 32.45±0.94 | 12.17±2.01 | 6 (0.48) |
> | PromptBook | 35.93±1.12 | 23.49±0.63 | 10 (0.18) | 33.03±0.67 | 24.03±3.89 | 10 (0.17) | 31.65±3.32 | 25.93±2.09 | 10 (0.16) |
> | **Ca$^{2}$P (ours)** | **45.91±4.37** | **3.24±0.65** | **1 (0.73)** | **42.82±2.64** | **3.47±0.16** | **1 (0.71)** | **33.98±1.27** | **4.39±0.43** | **1 (0.67)** |
>
>
>
> A key distinction of Ca$^{2}$P is that it enables capabilities not supported by per-episode caching. By maintaining persistent and interpretable memory in the form of function-level KV states, it allows validated functional units to be reused across tasks and supports efficient task-specific adaptation through in-place modification within those KV states in dynamic embodied environments.
>
> [1] Liang et al., *Code as Policies: Language Model Programs for Embodied Control.* ICRA 2023.
>
> [2] Li et al., *Structured Chain-of-Thought Prompting for Code Generation.* TOSEM 2025.
>
> [3] Sarch et al., *Open-Ended Instructable Embodied Agents with Memory-Augmented Large Language Models.* EMNLP 2023.
>
> [4] Tziafas and Kasaei, *Lifelong Robot Library Learning: Bootstrapping Composable and Generalizable Skills for Embodied Control with Language Models.* ICRA 2024.
>
> [5] Arenas et al., *How to Prompt Your Robot: A Promptbook for Manipulation Skills with Code as Policies.* ICRA 2024.
>
> ---
>
> We have carefully addressed the concerns raised and clarified the scope, novelty, and advantages of Ca$^{2}$P over existing KV and memory-based approaches. We greatly appreciate your thoughtful review and hope these clarifications provide a clearer understanding of our contributions. If there is anything we may have any clarification that would further strengthen the paper, we would sincerely welcome your feedback and will respond promptly.

---

> ### Author Response · Authors · 2025-11-27
>
> Dear Reviewer cxo5,
>
> Thank you again for your thoughtful feedback. As the rebuttal period is nearing its end, we provide a brief summary to facilitate a more focused and efficient continuation of the discussion.
>
> Regarding the efficiency baselines you mentioned, we fully agree that existing toolbox methods-such as token caching or model quantization-already deliver substantial gains, and we emphasize that our method is complementary to these techniques rather than competing with them.
>
> At the same time, we would like to clarify that our work addresses a different problem setting than the one these toolbox methods were designed for. Token caching and quantization primarily target static or short-horizon text generation, whereas our approach is motivated by the unique challenges of open-domain embodied environments, where the agent must operate over long, unpredictable task streams and continuously update, reuse, and reorganize executable code policies during execution.
>
> Our goal, therefore, is not to replace existing toolbox optimizations but to provide a framework that addresses open-domain embodied tasks, which are not accounted for in the intended scope of current methods. We hope this distinction makes clear that our contribution lies in addressing a setting that the current toolbox does not directly target, while remaining fully synergistic with these efficiency techniques.
>
> ---
>
> ### **Summary of Your Core Concern**
>
> You noted that our contribution may appear to focus mainly on cache management for code generation-an area already well explored in coding agents-and that existing toolbox methods (e.g., token caching, model quantization) already provide strong efficiency benefits. You also asked us to clarify how our caching mechanism differs from the existing CaP implementation and what concrete advantages it provides.
>
> ---
>
> ### **Summary of Our Response**
>
> **1. Complementary to existing toolbox optimizations, not competing with them.**
> We agree that toolbox methods such as token caching and model quantization already provide substantial efficiency gains. Our method is fully compatible with these techniques and can be used alongside them. Our goal is not to replace these optimizations but to address a different category of challenges that they were not designed for.
>
> **2. Different problem setting: open-domain embodied environments.**
> Toolbox methods target static or short-horizon text generation with fixed or predictable prompts.
> In contrast, our setting is open-domain embodied control, where the agent operates over long, unpredictable task streams and must continuously update, reuse, and reorganize executable code policies during execution. These structural and long-horizon requirements fall outside the design assumptions of existing toolbox methods.
>
> **3. Persistent, editable function-level memory across tasks.**
> Unlike the per-episode caching in open-sourced CaP, which only benefits the current code-generation session, our method maintains persistent, executable function-level KV states that can be reused, composed, and modified in-place across tasks. This enables rapid adaptation to dynamic embodied environments without regenerating entire programs.
>
> **4. Clear empirical advantages across embodied benchmarks.**
> Across ALFRED, TEACh, and RLBench, our approach consistently outperforms toolbox-style KV reuse (e.g., PromptCache, RAGCache, CAG) and memory-augmented CaP variants (e.g., HELPER, LRLL, PromptBook). These gains reflect improvements not just in inference latency but in actual embodied task performance under open-composition, open-perturbation, and open-evolution conditions.
>
> ---
>
> We hope this summary clarifies how our contribution addresses a problem setting not targeted by existing toolbox optimizations while remaining fully synergistic with them. If anything remains unclear or if you would like further elaboration, we would be grateful for the opportunity to continue the discussion. Even if you join the discussion later, we will respond immediately and do our best to clarify any remaining concerns.

---

### Official Review · Reviewer_r7c6 · 2025-10-31

**Soundness:** 3
**Presentation:** 3
**Contribution:** 3
**Rating:** 8
**Confidence:** 2

**Summary:**

This paper tackles a practical issue with Code-as-Policies (CaP) for robot control—every new instruction forces the model to rewrite all the code, making it slow and sometimes inconsistent. The proposed approach, CA2P, reuses previously verified functions by caching key–value (KV) states and updating code in place instead of regenerating everything from scratch. It builds a function-level caching system that supports both code reuse and quick local edits. Tests on ALFRED, TEACh, and RLBench show higher success rates and up to 2.9× faster responses than standard CaP methods. Real-robot trials confirm the improvements in both speed and stability.

**Strengths:**

- Clear and well-structured technical design (two-tier cache: Function-Interface and Function-Code) with coherent equations and pseudocode.

- Writing is clear and figures are informative

- Extensive experimental setup across multiple environments, baselines, and metrics.

**Weaknesses:**

- In Algorithm 1, the try–except block appears tailored for simulation, where errors can be easily caught. It remains unclear how such failures would be handled in real-world deployments.

**Questions:**

N/A

---

> ### Author Response · Authors · 2025-11-18
> **Response to Weakness 1**
>
> Dear Reviewer r7c6
>
> Thank you for your insightful feedback. We understand the reviewer's concern as arising from the fact that, unlike in simulation, our method for detecting failure states in real-world robot execution was not described in sufficient detail. We appreciate the opportunity to clarify this aspect, and we will expand the explanation of our real-world robot framework in Appendix to make this process more explicit. The revised version will be uploaded along with other revisions, and we will notify you once it becomes available in the coming days.
>
> ---
>
> > **Weakness 1**: In Algorithm 1, the try–except block appears tailored for simulation, where errors can be easily caught. It remains unclear how such failures would be handled in real-world deployments.
>
> As a first step, although the implementation differs between simulation and real-world settings due to their inherent environmental differences, we made an effort to standardize the types of exceptions and the feedback format across both environments.
>
> We categorize exceptions into two types:
> (1) **interpreter-level exceptions** (e.g., `SyntaxError`, `NameError`) that are Python runtime errors working identically in both settings, and
> (2) **environment-level exceptions** (categorized as `RobotError`) such as grasp failures, workspace violations, and safety stops.
> While the *implementation* differs between simulation and real-world, the *exception interface* remains consistent, enabling cache-refactoring to handle them uniformly. The exception types are summarized in the table below.
>
> | **Type** | **Category** | **Details** |
> |----------|--------------|-------------|
> | **Interpreter-level** | SyntaxError        | Invalid Python syntax / Malformed code block / Non-code text in code region |
> | **Interpreter-level** | ClassNotFound      | Missing or unsupported Pygments lexer for given code block |
> | **Interpreter-level** | FileNotFoundError  | Missing configuration file (e.g., object metadata, orientation map, noise parameters) |
> | **Environment-level** | RobotError | Skill sequence execution failure / Path out of workspace / Object out of reach / Grasp/manipulation failure / Timeout while executing skill primitive / Invalid object type binding / Object handle does not exist / Missing object constraints in object mapping / Infeasible approach pose / Orientation constraint violation / Feedback-triggered safety stop or low-level failure |
>
> In simulation (ALFRED, TEACh, RLBench), we leverage built-in APIs that directly report execution failures. In real-world deployment with the Franka Emika Research 3, low-level safety exceptions are provided by the robot controller, but higher-level task failures require additional perception modalities as described in Appendix B.4. We implement environment-level exception detection using dual Intel RealSense D435 depth cameras, object detection models, and vision-language models.
>
> For example, to detect **grasp/manipulation failure during skill execution**, we proceed as follows: After the robot executes a `pick` action, we use the object detection model to localize the target object in the current RGB-D frame and compute its 3D position via point cloud registration. We then compare this actual object position with the expected gripper position. If the Euclidean distance exceeds a threshold (indicating the object was not successfully grasped or has fallen), we raise a `RobotError` with the failure description. Similarly, for higher-level semantic failures such as verifying whether a drawer is properly opened or a workspace is cleared, we query the VLM with the current observation and the expected state description. If the VLM reports a mismatch, the corresponding `RobotError` is raised.
>
> These mechanisms ensure that all failures, whether originating from Python execution or from perception-driven physical interaction, are funneled into the same exception interface and resolved through the same cache-refactoring. As a result, the try-except block in Algorithm~1 is not simulation-specific; it reflects a unified, environment-agnostic exception handling design that supports both simulated and real-world deployments in a consistent manner.
>
> ---
>
> We have carefully addressed the concerns raised and expanded the explanation of our real-world failure detection mechanism as recommended. We greatly appreciate your thoughtful review and hope these clarifications provide a clearer understanding of our approach. If anything remains unclear or inadequately resolved, please do not hesitate to inform us. We are fully committed to addressing any remaining concerns.

---

> ### Author Response · Authors · 2025-11-27
>
> Dear Reviewer r7c6,
>
> Thank you again for taking the time to review our work. As the rebuttal period is nearing its end, we provide a brief summary to facilitate a more focused and efficient continuation of the discussion. Below, we outline what we understand to be your core concern and how our response addresses it.
>
> ---
>
> ### **Summary of Your Core Concern**
>
> You noted that the try–except block in Algorithm 1 appears tailored to simulation, where failures can be programmatically captured. It was unclear how comparable failures would be detected and handled in real-world robot deployments, where errors are less neatly structured than in simulated environments.
>
> ---
>
> ### **Summary of Our Response**
>
> **1. Unified exception interface across simulation and real-world execution.**
> We clarified that although the underlying detection mechanisms differ between environments, we standardize all failures into a unified exception interface. Interpreter-level exceptions (e.g., SyntaxError, NameError, missing files) behave identically in both settings, and environment-level issues are categorized as `RobotError` to ensure consistent handling through cache-refactoring.
>
> **2. Real-world robot deployments include dedicated perception-based failure detection.**
> Unlike simulation, real-world failures cannot be retrieved directly from an API. We therefore implement perception-driven checks using RGB-D cameras, object detection models, and VLM-based state verification. These detect grasp failures, object misalignment, workspace violations, pose mismatches, and other interaction-related errors.
>
> **3. Real-world failures are routed into the same try–except pathway.**
> Regardless of origin-Python execution or physical interaction-failures are converted into the same exception format. This enables Algorithm 1’s try–except structure to function identically in simulation and on a real robot, making the mechanism environment-agnostic rather than simulation-specific.
>
> **4. Expanded paper clarification.**
> We added a clearer description of the real-world failure detection pipeline in the Appendix (Section B.4), along with the standardized exception taxonomy to prevent confusion about how physical execution failures map into the algorithmic interface.
>
> ---
>
> We hope this summary makes the distinction clearer and demonstrates that the try–except block in Algorithm 1 is not tied to simulation, but is part of a unified exception-handling design intended for both simulated and real-world deployments.
>
> If any part of this remains unclear or if you would like to discuss additional details, we would be very happy to clarify further. And even if you join the discussion later, we will respond immediately and address any remaining questions.

---

### Official Review · Reviewer_qB9j · 2025-11-01

**Soundness:** 3
**Presentation:** 2
**Contribution:** 3
**Rating:** 4
**Confidence:** 3

**Summary:**

This paper aims to address the high latency and inconsistency of the Code-as-Policies (CaP) paradigm, where large language models (LLMs) generate full control code from scratch for every embodied task. This paper presents $CA^{2}P$, a Cache-Augmented Code-as-Policies framework for embodied AI agents in open-domain environments. The core innovation is function-level key-value (KV) caching that repurposes native transformer attention caching to enable code reuse. The system maintains a two-tier cache (Function-Interface and Function-Code) indexed by function identifiers, supporting compositional programming (assembling new policies from cached functions) and cache-refactoring (editing cached functions via fill-in-the-middle). A cache management scheme based on recency, frequency, co-occurrence, and semantic diversity scores determines retention. Experiments on ALFRED, TEACh, and RLBench 8benchmarks, as well as real-world robot manipulation, show that $CA^{2}P$ achieves a superior trade-off between task success rate (SR) and policy synthesis latency (PSL), outperforming CaP baselines.

**Strengths:**

- The two-tier cache design (Function-Interface I and Function-Code C) is well-motivated for separating lightweight references from full implementations, enabling efficient compositional programming without redundant attention computation

- Evaluation spans three simulation benchmarks (ALFRED, TEACh, RLBench) with different task characteristics plus real-world manipulation. Thorough baseline comparisons and ablation studies are conducted

**Weaknesses:**

- The locality score $l(f_k)$ in Equation (3) is central to the method, but its components are defined only at a high level. $l_{freq}$ ("usage frequency") and $l_{asso}$ ("conditional association") are not given precise mathematical definitions, making it difficult to reimplement the scoring function exactly. The weights $\alpha, \beta, \gamma$ in Equation (3) are set to 0.4, 0.3, and 0.3, but this choice is presented without justification. No ablation or sensitivity analysis is provided.

- The "code cache warm-up" analysis (Fig 4) explicitly states it starts from an "empty cache". However, Appendix D.1 states: "All cache-based baselines and $CA^{2}P$ begin with the same initial KV cache states derived from basic success code policies" for the main benchmark results. This implies the main results in Table 1 use a pre-populated cache, not one warmed-up from empty. This is a crucial distinction that is not clarified in the main paper.

- The "open-domain" framing is somewhat overstated—all benchmarks provide predefined API sets (Tables 6-9) with fixed primitives; true open-domain deployment would require handling novel APIs or learning from demonstrations, which is not addressed.

**Questions:**

- For the real-world experiments (Table 2), the paper states the cache is built by "first solves simpler tasks in RLBench". Is the performance therefore dependent on this pre-population step?

- In Algorithm 1, when is generate() versus edit() called? How are exceptions E detected and categorized? What triggers cache updates beyond task success?

---

> ### Author Response · Authors · 2025-11-18
> **Response to Weakness 1**
>
> Dear Reviewer qB9j,
>
> Thank you for your insightful and constructive review. We genuinely appreciate the time and care you devoted to evaluating our work, and your comments were highly valuable in refining both the clarity and completeness of the paper. Below, we address each of your concerns with precise and comprehensive responses. A revised version-including the additional experiments and updates-will be uploaded shortly, and we will inform you as soon as it becomes available in the coming days.
>
> ---
>
> > **Weakness 1**: The locality score $\ell(f_k)$ in Equation (3) is central to the method, but its components are defined only at a high level. $\ell_{freq}$ ("usage frequency") and $\ell_{asso}$ ("conditional association") are not given precise mathematical definitions, making it difficult to reimplement the scoring function exactly. The weights in Equation (3) are set to 0.4, 0.3, and 0.3, but this choice is presented without justification. No ablation or sensitivity analysis is provided.
>
> We appreciate the reviewer’s insightful comments. Both points are important for improving the clarity of our locality scoring mechanism. Some definitions that were originally intended for the main text were inadvertently omitted during the main paper’s length adjustments, and we will include the full details in the revision. Below, we provide precise definitions for each component.
>
> **Clarification of locality score components.** The three components of the locality score, $\ell_{freq}$, $\ell_{asso}$, and $\ell_{sema}$, are not intended merely to preserve recently used functions. Each term captures a different aspect of the dynamics that arise in open-domain task streams. Their formal definitions are provided below.
>
> *$\ell_{freq}$ Usage frequency.* We compute how frequently each function is invoked over recent tasks:
> $$
> \ell_{\mathrm{freq}}(f_k) =
> \frac{\mathrm{count}(f_k)}{\max_j \mathrm{count}(f_j)}
> $$
>
> This reflects repetitive or sequential patterns that commonly appear in streaming tasks.
>
> *$\ell_{asso}$ Conditional association.* We measure conditional co-occurrence within an execution trace by computing the proportion of times $f_j$ is invoked after $f_k$:
> $$
> \ell_{\mathrm{asso}}(f_j \mid f_k) =
> \frac{\mathrm{cooccur}(f_k, f_j)}{\mathrm{count}(f_k)}
> $$
> This captures compositional and order-dependent relationships between functions, which are important in tasks where the execution sequence and composition matter.
>
> *$\ell_{sema}$ Semantic novelty.* We treat each function's implementation as a document and evaluate its perplexity using the CodeLLM:
> $$
> \ell_{\mathrm{sema}}(f_k) =
> \frac{\mathrm{PPL}(f_k)}{\max_j \mathrm{PPL}(f_j)}
> $$
> This promotes semantic diversity in the code cache by retaining functions that are less common or more distinct, which is useful for handling unpredictable task variations in open-domain environments.
>
> **Ablation on the contribution of each component.** Although Table 5 in the main text already includes an ablation on $\ell_{\text{sema}}$, we additionally provide extended ablations to address the reviewer’s concern by evaluating each component individually.
> For each ablation, we set one component’s weight to 1 and the remaining components’ weights to 0 (e.g., $(\alpha, \beta, \gamma) = (1, 0, 0)$), and we construct open-domain task streams that reflect specific task-pattern characteristics. The results show the following patterns:
>
> - **Repetitive task streams**: $\ell_{freq}$ alone is most effective.
> - **Order-dependent task streams**: $\ell_{asso}$ alone is most effective.
> - **Semantically diverse task streams**: $\ell_{sema}$ alone is most effective.
>
> These results will be included in the revision as a table.
>
> | **RLBench** | *Repetitive* ||| *Order-dependent* ||| *Semantic diverse* |||
> |-----------|----------------------|------|------|----------------------|------|------|----------------------|------|------|
> | **Methods** | **SR (↑)** | **PSL (↓)** | **HR (↑)** | **SR (↑)** | **PSL (↓)** | **HR (↑)** | **SR (↑)** | **PSL (↓)** | **HR (↑)** |
> | $\alpha$ | 36.43 | 3.34  | 67.04 | 34.34 | 3.32 | 61.04 | 35.12 | 3.40 | 60.23|
> | $\beta$ | 33.32 | 3.43 | 56.04 | 37.20 | 3.33 | 69.32 | 33.92 | 3.92 | 54.95 |
> | $\gamma$ | 34.42 | 3.30 | 61.26 | 33.33 | 3.14 | 56.58 | 38.12 | 3.60 | 68.29 |

---

> ### Author Response · Authors · 2025-11-18
> **Response to Weakness 1**
>
> **Justification of the final weights and sensitivity analysis.** To assess the sensitivity of our method to the relative weighting of the locality score components,
> We tested a range of weight settings for $\ell_{\mathrm{freq}}$, $\ell_{\mathrm{asso}}$, and $\ell_{\mathrm{sema}}$ in Eq. (3) of the main text. In particular, we evaluated configurations from balanced choices (e.g., $(\alpha,\beta,\gamma)=(0.4,0.3,0.3)$) to more imbalanced ones (e.g., $(0.6,0.2,0.2)$) to determine whether reasonable variations in these weights affect performance.
>
> As shown in the table below, the performance of our framework is not highly sensitive to the precise choice of $(\alpha,\beta,\gamma)$. Across all tested configurations, SR fluctuates by approximately 4.6\% and HR by about 10.7\%. Given that, in the RLBench results of Table 1 in the main text, our framework outperforms the strongest baselines by over 9.8\% in SR, these variations are comparatively small. This suggests that the locality score remains stable under reasonable adjustments to the weight allocation. More fine-grained trends can also be observed. Configurations that assign a weight of $0.6$ to a single component tend to yield slightly lower SR and HR, suggesting that overemphasizing any one signal (e.g., frequency or semantic diversity alone) can reduce cache utility in open-domain settings. Because open-domain tasks exhibit compositional structure and frequent changes in object states and goals, weight distributions that do not strongly favor a single component are generally more suitable. Among the balanced configurations, $(0.4,0.3,0.3)$ achieved the highest SR (45.81) and HR (74.03). Based on (1) the overall empirical trends observed across all tested configurations and (2) the small but consistent advantage of balanced settings, we fixed $(\alpha,\beta,\gamma) = (0.4,0.3,0.3)$ in all main experiments to avoid introducing unnecessary hyperparameter variability. We also observed similar trends across other benchmarks in our experiments, and we will include the corresponding results in the updated version.
>
> | **α** | **β** | **γ** | **SR (↑)** | **PSL (↓)** | **HR (↑)** |
> |------|------|------|-----------|-------------|-----------|
> | 0.4 | 0.3 | 0.3 | 45.81 | 3.32 | 74.03 |
> | 0.3 | 0.4 | 0.3 | 43.56 | 3.28 | 67.36 |
> | 0.3 | 0.3 | 0.4 | 43.44 | 3.38 | 68.20 |
> | 0.5 | 0.3 | 0.2 | 44.92 | 3.42 | 69.45 |
> | 0.3 | 0.5 | 0.2 | 42.45 | 3.38 | 66.71 |
> | 0.2 | 0.3 | 0.5 | 43.39 | 3.37 | 67.88 |
> | 0.5 | 0.2 | 0.3 | 45.18 | 3.39 | 71.82 |
> | 0.3 | 0.2 | 0.5 | 43.82 | 3.40 | 67.38 |
> | 0.2 | 0.5 | 0.3 | 42.43 | 3.35 | 66.11 |
> | 0.6 | 0.2 | 0.2 | 42.54 | 3.38 | 67.36 |
> | 0.2 | 0.6 | 0.2 | 41.20 | 3.42 | 63.36 |
> | 0.2 | 0.2 | 0.6 | 41.94 | 3.34 | 63.36 |

---

> ### Author Response · Authors · 2025-11-18
> **Response to Weakness 2**
>
> > **Weakness 2**: The "code cache warm-up" analysis (Fig 4) explicitly states it starts from an "empty cache". However, Appendix D.1 states: "All cache-based baselines and Ca$^2$P begin with the same initial KV cache states derived from basic success code policies" for the main benchmark results. This implies the main results in Table 1 use a pre-populated cache, not one warmed-up from empty. This is a crucial distinction that is not clarified in the main paper.
>
> We thank the reviewer for carefully examining not only the main text but also the Appendix, and for pointing out subtle details that help us identify areas requiring additional clarification. We clarify the relationship between the experiments referenced in the main text and the Appendix. The code cache warm-up experiment in Figure 4 corresponds to the analysis presented in Appendix D.3, whereas the main benchmark results in Table 1 correspond to the evaluation setup described in Appendix D.1. In Appendix D.1, the initial KV states are derived from external oracle policies for basic functions, and this prior information is provided equally across all comparison baselines. In particular, the KV caching methods receive the same functions encoded in KV-state form to maintain an equivalent initialization. In contrast, the experiment in Figure 4 is designed to observe each method’s warm-up characteristics from the beginning, so all methods-including those using KV caching-start with an empty cache. These experiments serve different purposes, and we explicitly state this distinction in the revision to prevent any potential confusion.

---

> ### Author Response · Authors · 2025-11-18
> **Response to Weakness 3**
>
> > **Weakness 3**: The "open-domain" framing is somewhat overstated-all benchmarks provide predefined API sets (Tables 6-9) with fixed primitives; true open-domain deployment would require handling novel APIs or learning from demonstrations, which is not addressed.
>
> We thank the reviewer for highlighting the broader notion of open-domain tasks, including open-ended settings that require expanding the API set itself and learning from demonstrations. In prior CaP-based and memory-augmented embodied agent works [1, 2, 3], however, open-domain commonly refers to an unbounded task space, meaning that agents must handle novel instructions, unseen task compositions, and unpredictable scenario variations while still operating over a fixed low-level API set. This convention is also adopted in recent work such as HELPER [4] and LRLL [5], both of which define open-domain tasks under the same assumption of a fixed low-level API set.
>
> We agree with the reviewer that a more embodiment-level perspective of open-domain, in which the API set itself evolves over time, is an important direction. Importantly, our framework is compatible with this broader setting because it operates through executable code policy synthesis rather than learning direct neural control policies. When new primitives are introduced, the framework can incorporate them as newly generated code functions, which can then be reused and refactored through our function-level KV caching mechanism. To further clarify this point, we added experiments illustrating two types of API extensions.
>
> **Regarding handling novel APIs: functional extension.** The first concerns functional extensions of the API set. This scenario is already reflected in our real-world robot manipulation experiments built upon RLBench, as shown in Table 2 of the main text. Certain primitives available in the real-world setup (such as `Pull`) did not exist in RLBench, yet our framework successfully incorporated these newly introduced APIs and generated the corresponding code functions to solve the tasks. We will include this example in the Appendix for additional clarity.
>
> | **Domain**   | **Available API** | **Action Expression** |
> |--------------|-------------------|------------------------|
> | RLBench      | `pull()` X        | `run_action(pick, ...)` + `run_action(move, ...)` (manually pulling) |
> | Real-world   | `pull()` O        | `execute_pull(...)` (abstracted API) |
>
>
> **Regarding handling novel APIs: depth extension.** The second type concerns extensions in the depth or abstraction level of the API. We varied the granularity of the available primitives, transitioning from low-level to high-level APIs, and evaluated whether our framework can effectively utilize primitives at different abstraction levels to synthesize code policies. The table below reports the performance measured when RLBench tasks were configured using either low-level or high-level APIs. Examples of the low-level APIs are shown first, and the high-level examples are presented afterward.
>
> ```
> Low-level API
>
> ...
> waypoints = self.interpolate_pose(current_pose_list, ...)
> destination_pose = deepcopy(self.target_pose)
> destination_pose[2] += self.gripper_offset
> waypoints += self.interpolate_pose(intermediate_pose, ...)
>
> for waypoint in waypoints:
>     move_group.go_to_pose_goal(waypoint[0], waypoint[1], ...)
>
> self.gripper.grasp(0.005, gripper_force)
> ...
> ```
>
> ```
> High-level API
>
> ...
> execute_go()
> execute_pick(gripper_force = 7, axis=2)
> ...
> ```
> | **RLBench** | *Open-Composition* ||| *Open-Perturbation* ||| *Open-Evolution* |||
> |-----------|----------------------|------|------|----------------------|------|------|----------------------|------|------|
> | **Methods** | **SR (↑)** | **PSL (↓)** | **HR (↑)** | **SR (↑)** | **PSL (↓)** | **HR (↑)** | **SR (↑)** | **PSL (↓)** | **HR (↑)** |
> | low-level | 43.21 | 4.12 | 68.02 | 40.29 | 4.92 | 65.10 | 36.82 | 6.12 | 61.38|
> | High-level | 45.91 | 3.24 | 73.65 | 42.82 | 3.47 | 70.73 | 33.98 | 4.39 | 70.29 |

---

> ### Author Response · Authors · 2025-11-18
> **Response to Weakness 3**
>
> **Regarding learning from demonstrations.** We appreciate the reviewer’s insightful comment on learning from demonstrations as a potential requirement for truly open-domain deployment. We agree that incorporating demonstrations, particularly instructional video demonstrations, is an important and promising direction for expanding the capabilities of code-based robotic control.
>
> In examining this suggestion, we found Demo2Code [6], a pioneering line of work on video-instructed robotic programming. Notably, Demo2Code still assumes a predefined API set, which is consistent with the benchmarks used in our paper. When considering the reviewer’s points on API extension and learning from demonstrations together, we realized that this naturally leads to a problem setting where the demonstration domain and the deployment domain differ in environment and embodiment. This discrepancy creates a cross-domain transfer problem that code-based agents must handle, and we believe such a setting provides a meaningful direction for extending our framework.
>
> While learning from demonstrations is highly compelling, it falls outside the scope of our current work. Our work focuses on robust and efficient code policy synthesis for open-domain tasks by leveraging function-level KV caching, which enables compositional programming through direct function reuse and supports in-place adaptation through cache-refactoring.
>
> Nevertheless, the reviewer’s suggestion is highly valuable. We believe our framework naturally provides a foundation for future work in which (1) code memory can be built from heterogeneous demonstration sources, (2) functional knowledge can be extended through skills extracted from demonstrations, and (3) cache-refactoring can support efficient adaptation even when the demonstrated embodiment or environment differs from the deployment domain. We sincerely appreciate the reviewer for prompting this direction. This discussion has been genuinely inspiring, and we view demonstration-driven code memory construction as an exciting extension of our framework for future research.
>
>
> [1] Choi et al. NeSyC: A Neuro-symbolic Continual Learner For Complex Embodied Tasks In Open Domains. ICLR 2025.
>
> [2] Chen et al. Language-Augmented Symbolic Planner for Open-World Task Planning. RSS 2024.
>
> [3] Cai et al. Open-world multi-task control through goal-aware representation learning and adaptive horizon prediction. CVPR 2023.
>
> [4] Sarch et al., ''Open-Ended Instructable Embodied Agents with Memory-Augmented Large Language Models,'' EMNLP 2023.
>
> [5] Tziafas \& Kasaei, “Lifelong Robot Library Learning: Bootstrapping Composable and Generalizable Skills for Embodied Control with Language Models,” ICRA 2024.
>
> [6] Wang et al. Demo2Code: From Summarizing Demonstrations to Synthesizing Code via Extended Chain-of-Thought. NeurIPS 2023.

---

> ### Author Response · Authors · 2025-11-18
> **Response to Question 1**
>
> > **Question 1**: For the real-world experiments (Table 2), the paper states the cache is built by "first solves simpler tasks in RLBench". Is the performance therefore dependent on this pre-population step?
>
> Thank you for the insightful question. We clarify that the RLBench-based pre-population step mentioned in Table 2 is minimal and not required for our framework to operate. It is included to establish a realistic cross-domain transfer scenario for evaluation, rather than to supply information that the framework depends on.
>
> During pre-population, we populate the code cache with only minimal initial building blocks in the form of function interfaces and implementations The real challenge, and the focus of Table 2, is that these cached functions must be rapidly adapted and recomposed using our framework’s mechanisms once the system transitions to the real robot setting. This setting introduces domain shift, visual variation, mechanical disturbances, and safety-related events such as gas hose disconnection.
>
> This design highlights the key contribution of our framework, namely, cache-refactoring of previously generated functions and low-latency synthesis without full regeneration. The intent of the experiment is not to indicate dependence on pre-population, but rather to show that even when the cache originates from a different domain, our framework can efficiently modify and compose cached functions to achieve reliable real-world execution. We will include these clarifications in the revision to avoid any potential misunderstanding.
>
> To further demonstrate the necessity of our function-level code caching, and in particular the cache-refactoring that enables targeted modifications within cached KV states during policy synthesis, we conducted an additional controlled comparison Under the same cross-domain transfer setting, from RLBench to real-world, we evaluated two alternative memory structures: (1) program-level code memory (non-functional, monolithic code blocks), and (2) function-level code memory. For each design, we evaluated two variants: one in which the memory is maintained in KV cache form, and one in which it is not.
>
> | **Real-world** | *Program-level* |            |             |            | *Function-level* |            |             |            |
> | ----------- | --------------- | ---------- | ----------- | ---------- | ---------------- | ---------- | ----------- | ---------- |
> | **Methods** | **SR (↑)**      | **GC (↑)** | **PSL (↓)** | **HR (↑)** | **SR (↑)**       | **GC (↑)** | **PSL (↓)** | **HR (↑)** |
> | Non KV Cache   | 44.44           | 53.33      | 8.04        | 74.50      | 77.78            | 77.78      | 6.21        | 87.25      |
> | KV Cache | 66.67           | 71.11      | 4.52        | 74.50      | 88.89            | 88.89      | 3.80        | 87.25      |
>
> Across all conditions, our function-level code memory with KV-state achieved the best performance in both efficiency and behavioral consistency. These results demonstrate that fine-grained, function-level KV caching, together with in-place KV-state editing, is essential for fast and reliable adaptation in open-domain tasks.

---

> ### Author Response · Authors · 2025-11-18
> **Response to Question 2**
>
> > **Question 2**: In Algorithm 1, when is generate() versus edit() called? How are exceptions E detected and categorized? What triggers cache updates beyond task success?
>
> We sincerely appreciate the reviewer’s thoughtful question and the opportunity to clarify this point. We address this question in three parts below.
>
> **Algorithm Explanation.** Algorithm 1 in the main text is organized into two parts.
> (A) Task Execution Loop outlines the overall procedure for handling a stream of embodied tasks, and  (B) CA$^2$P Built-in Methods provides the detailed logic for the three core functions: `generate()`, `edit()`, and `update()`.
>
> In (A) Task Execution Loop, the agent enters a scenario loop (line 2) and repeatedly resets the environment to obtain a new observation $o_t$ and instruction $\tau$ (line 3). Line 4 calls `generate(oₜ, τ)` whenever a new instruction is given, synthesizing an executable policy $\pi_{\text{exec}}$ via compositional programming using cached functions from the Function-Interface tier $\mathcal{I}$. The synthesized policy is then executed step-by-step (lines 5-14). Line 11 calls `edit(oₜ₊₁, τ, E, π_exec)` on-demand during execution whenever an exception $E$ is raised (line 10), enabling in-place editing of the faulty code span through cache-refactoring. Once a task completes successfully (line 16), line 17 invokes `update(π_exec)` to decompose the validated policy into function-level entries and update the two-tier code cache $\mathcal{H}$ according to the cache management scheme.
>
> | Function        | When Called                                 | Purpose                                                |
> |-----------------|----------------------------------------------|--------------------------------------------------------|
> | `generate()`    | Line 4 — at the start of each task           | Synthesize new policy from cached functions (compositional programming) |
> | `edit()`        | Line 11 — when exception occurs during execution | Correct erroneous code span in-place (cache-refactoring) |
> | `update()`      | Line 17 — after successful task completion   | Decompose and cache validated functions (cache management) |

---

> ### Author Response · Authors · 2025-11-18
> **Response to Question 2**
>
> **Exception Detect and Categorize.** We categorize exceptions into two types:
> (1) **interpreter-level exceptions** (e.g., `SyntaxError`, `NameError`) that are Python runtime errors working identically in both settings, and  (2) **environment-level exceptions** (categorized as `RobotError`) such as grasp failures, workspace violations, and safety stops. While the *implementation* differs between simulation and real-world, the *exception interface* remains consistent, enabling cache-refactoring to handle them uniformly. The exception types are summarized in the table below.
>
> | **Type**              | **Category**      | **Details**                                                                                                                                                                                                                                                                                                                                                                           |
> | --------------------- | ----------------- | ------------------------------------------------------------------------------------------------------------------------------------------------------------------------------------------------------------------------------------------------------------------------------------------------------------------------------------------------------------------------------------- |
> | **Interpreter-level** | SyntaxError       | Invalid Python syntax / Malformed code block / Non-code text in code region                                                                                                                                                                                                                                                                                                           |
> | **Interpreter-level** | ClassNotFound     | Missing or unsupported Pygments lexer for given code block                                                                                                                                                                                                                                                                                                                            |
> | **Interpreter-level** | FileNotFoundError | Missing configuration file (e.g., object metadata, orientation map, noise parameters)                                                                                                                                                                                                                                                                                                 |
> | **Environment-level** | RobotError        | Skill sequence execution failure / Path out of workspace / Object out of reach / Grasp/manipulation failure / Timeout while executing skill primitive / Invalid object type binding / Object handle does not exist / Missing object constraints in object mapping / Infeasible approach pose / Orientation constraint violation / Feedback-triggered safety stop or low-level failure |
>
> In simulation (ALFRED, TEACh, RLBench), we leverage built-in APIs that directly report execution failures. In real-world deployment with the Franka Emika Research 3, low-level safety exceptions are provided by the robot controller, but higher-level task failures require additional perception modalities as described in Appendix B.4. We implement environment-level exception detection using dual Intel RealSense D435 depth cameras, object detection models, and vision-language models.
>
> For example, to detect **grasp/manipulation failure during skill execution**, we proceed as follows: After the robot executes a `pick` action, we use the object detection model to localize the target object in the current RGB-D frame and compute its 3D position via point cloud registration. We then compare this actual object position with the expected gripper position. If the Euclidean distance exceeds a threshold (indicating the object was not successfully grasped or has fallen), we raise a `RobotError` with the failure description. Similarly, for higher-level semantic failures such as verifying whether a drawer is properly opened or a workspace is cleared, we query the VLM with the current observation and the expected state description. If the VLM reports a mismatch, the corresponding `RobotError` is raised.

---

> ### Author Response · Authors · 2025-11-18
> **Response to Question 2**
>
> **Cache Updates Beyond Task Success.** In the current implementation, cache updates are performed only upon successful task completion, consistent with Algorithm 1 in which `update()` is triggered after a success. This approach guarantees that only validated and reliable functions influence the locality score and are stored in the two-tier code cache.
>
> Although our current implementation updates the cache only after full task success, a neurosymbolic extension would allow us to identify function-level correctness even within failed executions through pre- and postcondition verification [1, 2]. By combining LLM-based semantic correctness checks with rule-based symbolic validation, the framework could extract richer feedback signals similar to goal-condition success rate (GC), enabling more fine-grained and informative updates to the code cache.
>
> Another extension is considering functions that repeatedly correlate with execution failures. The KV cache can be augmented with negative evidence that marks these functions as failure-associated. Incorporating such annotations would down-weight their reuse probability without fully evicting them, thereby reducing repeated failure patterns and improving robustness in long-horizon execution.
>
> [1] Wu et al., ''Lemur: Integrating Large Language Models in Automated Program Verification,'' ICLR 2024.
>
> [2] Ahn et al., ''Towards Reliable Code-as-Policies: A Neuro-Symbolic Framework for Embodied Task Planning,'' NeurIPS 2025.
>
> ---
> We have thoroughly examined your comments from multiple perspectives and carefully interpreted the issues raised in the weaknesses section, incorporating additional ablation studies as recommended. We sincerely appreciate your thoughtful review, and we hope that these revisions offer a clearer and more comprehensive understanding of our contributions. Should any part of our responses or revisions fall short of your expectations, we would greatly appreciate your guidance, and we will gladly provide additional clarification or updates as needed.

---

> > ### Comment · Reviewer_qB9j · 2025-11-27
> >
> > I thank the authors for their detailed and comprehensive response. The additional experiments and clarifications have effectively addressed my concerns. I have raised my score accordingly.

---

> > > ### Author Response · Authors · 2025-11-27
> > >
> > > We sincerely thank you for the time and effort you dedicated to reviewing our work. We greatly appreciate your thoughtful assessment and are pleased that your main concerns have been fully addressed. Your feedback has been invaluable in further improving the clarity and overall quality of our paper.
> > >
> > > Best regards,
> > > The Authors

---

### Official Review · Reviewer_NSpR · 2025-11-01

**Soundness:** 4
**Presentation:** 3
**Contribution:** 3
**Rating:** 6
**Confidence:** 4

**Summary:**

Authors propose to utilize caching in order to improve the latency of code as policies framework for LLM based robotic control. Authors perform experiments with real world robot.

**Strengths:**

- Good empirical performance.
- Clear contribution as identified knowledge gap in code as policies framework.
- Well engineered solution.

**Weaknesses:**

- Related work does not have any citations. Please rewrite, the purpose of related work is to cite previous works.
- In respect to these hyperparameters, such as  $\alpha$,  $\beta$, $\gamma$ and others, I did not see experiments where those values were varied. This is strange considering the amount of results in presented in the paper. Maybe authors can explain this?

**Questions:**

- In line 203, why $\alpha$,  $\beta$, $\gamma$ needs to sum to one?

---

> ### Author Response · Authors · 2025-11-18
> **Response to Weakness 1**
>
> Dear Reviewer NSpR,
>
> Thank you for your thoughtful and constructive review. We appreciate your careful assessment and the important points you raised, which helped us improve the clarity and completeness of the paper. We have provided detailed and direct responses to all of your concerns below, and the revised version will be uploaded along with the additional experiments and other revisions. We will notify you once it becomes available in the coming days.
>
> ---
>
> > **Weakness 1**: Related work does not have any citations. Please rewrite, the purpose of related work is to cite previous works.
>
> Thank you for pointing this out. We acknowledge that the current version of the related work section in the main text provides only a conceptual overview and does not yet include citations. As noted, the detailed discussion of related work, together with all corresponding citations, is provided in Appendix A. Given that the rebuttal permits up to ten pages, we will revise the related work section in the main text to include representative citations and more clearly position our contributions relative to existing studies.
>
> Before providing the full revision, we first address the reviewer’s key concern by briefly outlining how we plan to update the section. Below, we summarize the categories of related work and list representative prior studies for each.
> - **Large language models for embodied control.** Recent advances in embodied control increasingly leverage large language models (LLMs) for task planning [1, 2]. A growing line of work further employs code-writing LLMs (CodeLLMs) to generate executable control code policy that invokes perception and motor APIs for motion-level control [3, 4]. In this work, we extend the Code-as-Policies (CaP) framework to improve robotic programming, overcoming the latency and inconsistency that arise from full regeneration of code policies in dynamic environments.
> - **Memory-based embodied agents.** Memory-augmented embodied agents maintain structured experience buffers, including observations, trajectories, and latent embeddings, to support long-horizon reasoning, task adaptation, and generalization [5, 6]. In the context of CaP, only a few studies have explored how memory can enhance code-writing capabilities [7, 8]. Unlike prior memory-based approaches that rely primarily on text-level representations for generalization, Ca$^{2}$P introduces function-level KV caching tailored for function reuse, enabling rapid and reliable cache-augmented code policy synthesis.
> - **Key-value caching.** Recent work leverages cached attention states to accelerate generation and proposes adaptive strategies for combining multiple cache segments in dynamic contexts [9]. In addition, infilling methods such as PIE [10] and EFIM [11] support fill-in-the-middle generation by inserting updated code into cached sequences. Inspired by these advances, Ca$^{2}$P repurposes KV caching for embodied control, supporting compositional programming and refactoring to reduce the need for full regeneration and enable efficient, task-specific code policy synthesis.
>
> We will ensure that the main text accurately reflects these references and explanations. We also reiterate that a more detailed related work section, including all citations and extended discussion, is provided in Appendix A, and we would appreciate the reviewer’s consideration of this context.
>
> [1] Brohan et al., *Do as I Can, Not as I Say: Grounding Language in Robotic Affordances.* CoRL 2023.
>
> [2] Song et al., *LLM-Planner: Few-Shot Grounded Planning for Embodied Agents with Large Language Models.* ICCV 2023.
>
> [3] Liang et al., *Code as Policies: Language Model Programs for Embodied Control.* ICRA 2023.
>
> [4] Burns et al., *GenCHiP: Generating Robot Policy Code for High-Precision and Contact-Rich Manipulation Tasks.* IROS 2024.
>
> [5] Shinn et al., *Reflexion: Language Agents with Verbal Reinforcement Learning.* NeurIPS 2024.
>
> [6] Yang et al., *Embodied Multi-Modal Agent Trained by an LLM from a Parallel TextWorld.* CVPR 2024.
>
> [7] Sarch et al., *Open-Ended Instructable Embodied Agents with Memory-Augmented Large Language Models.* EMNLP 2023.
>
> [8] Tziafas & Kasaei, *Lifelong Robot Library Learning: Bootstrapping Composable and Generalizable Skills for Embodied Control with Language Models.* ICRA 2024.
>
> [9] Yao et al., *CacheBlend: Fast Large Language Model Serving for RAG with Cached Knowledge Fusion.* EuroSys 2025.
>
> [10] He et al., *Let the Code LLM Edit Itself When You Edit the Code.* ICLR 2025.
>
> [11] Guo et al., *EFIM: Efficient Serving of LLMs for Infilling Tasks with Improved KV Cache Reuse.* Euro-Par 2025.

---

> ### Author Response · Authors · 2025-11-18
> **Response to Weakness 2**
>
> > **Weakness 2**: In respect to these hyperparameters, such as $\alpha$, $\beta$, $\gamma$ and others, I did not see experiments where those values were varied. This is strange considering the amount of results in presented in the paper. Maybe authors can explain this?
>
> Thank you for pointing out this important question regarding hyperparameter sensitivity. We agree that understanding the effect of the weighting parameters (e.g., $\alpha$, $\beta$, $\gamma$) and other hyperparameters is important for interpreting the behavior of Ca$^{2}$P.
> Below, we clarify our design choices and provide additional sensitivity analyses.
>
> First, we provide the rationale behind our choices of $\alpha$, $\beta$, and $\gamma$ together with additional experiments examining the effect of varying these weights. These values were selected based on pilot studies and straightforward empirical observations, and we report the corresponding sensitivity results below. Second, we describe the rationale behind the remaining hyperparameter settings.
>
> **Pilot study on the locality score weights.** To assess how sensitive the overall performance is to the relative weighting of the components of the locality score, we experimented with a variety of settings for the values of $\ell_{\mathrm{freq}}$, $\ell_{\mathrm{asso}}$, and $\ell_{\mathrm{sema}}$ in Eq. (3) of the main text. In particular, we tested configurations ranging from balanced settings (e.g., $(\alpha, \beta, \gamma)=(0.4, 0.3, 0.3)$) to more imbalanced ones (e.g., $(0.6, 0.2, 0.2)$) to examine whether reasonable variations in these values affect performance.
>
> As shown in the table below, the performance of Ca$^{2}$P is not highly sensitive to the exact choice of $(\alpha, \beta, \gamma)$. Across all tested configurations, SR varies within approximately 4.6% and HR within 10.7%. Considering that, in the RLBench results of Table 1 in the main text, Ca$^{2}$P outperforms the strongest baselines by over 9.8% in SR, these fluctuations are comparatively small. This indicates that the locality score remains relatively stable under reasonable changes to the weight allocation. More fine-grained trends can also be observed. Configurations that assign a weight of $0.6$ to a single component tend to exhibit slightly lower SR and HR, suggesting that overly emphasizing one signal (e.g., frequency or semantic diversity alone) can reduce cache utility in open-domain settings. Because open-domain tasks involve compositional structure and dynamic changes in object states and goals, weight distributions that avoid heavily favoring a single component are generally more appropriate. Among the balanced configurations, $(0.4, 0.3, 0.3)$ achieved the highest SR (45.81) and HR (74.03). Considering (1) the overall empirical trends observed across all tested configurations and (2) the small but consistent advantage of balanced configurations, we fixed $(\alpha, \beta, \gamma) = (0.4, 0.3, 0.3)$ in all main experiments to avoid introducing unnecessary hyperparameter variability. We also observed that these performance trends were consistent across other benchmarks during our experiments, and we will include the corresponding results in the updated version.
>
>
> | **α** | **β** | **γ** | **SR (↑)** | **PSL (↓)** | **HR (↑)** |
> |------|------|------|-----------|-------------|-----------|
> | 0.4 | 0.3 | 0.3 | 45.81 | 3.32 | 74.03 |
> | 0.3 | 0.4 | 0.3 | 43.56 | 3.28 | 67.36 |
> | 0.3 | 0.3 | 0.4 | 43.44 | 3.38 | 68.20 |
> | 0.5 | 0.3 | 0.2 | 44.92 | 3.42 | 69.45 |
> | 0.3 | 0.5 | 0.2 | 42.45 | 3.38 | 66.71 |
> | 0.2 | 0.3 | 0.5 | 43.39 | 3.37 | 67.88 |
> | 0.5 | 0.2 | 0.3 | 45.18 | 3.39 | 71.82 |
> | 0.3 | 0.2 | 0.5 | 43.82 | 3.40 | 67.38 |
> | 0.2 | 0.5 | 0.3 | 42.43 | 3.35 | 66.11 |
> | 0.6 | 0.2 | 0.2 | 42.54 | 3.38 | 67.36 |
> | 0.2 | 0.6 | 0.2 | 41.20 | 3.42 | 63.36 |
> | 0.2 | 0.2 | 0.6 | 41.94 | 3.34 | 63.36 |

---

> ### Author Response · Authors · 2025-11-18
> **Response to Weakness 2**
>
> **Rationale for other fixed hyperparameters.** To determine the remaining fixed hyperparameters, we conducted small-scale tests and empirical inspections aimed at identifying stable and safe operating regions rather than performing exhaustive tuning. Importantly, the selected values were set to ensure that all methods operate normally within these ranges, avoiding failures caused by factors unrelated to their underlying algorithms. This ensures that performance differences arise from the methods themselves rather than instability in the evaluation setup.
>
> *GPU memory threshold.* We set the GPU memory threshold to 85%, which we found to be a stable and safe value on our RTX 4090 setup. This value was kept fixed across all experiments to maintain consistent evaluation conditions.
>
> *Min free GPU before generation (GB).* We set the minimum free GPU memory to 1.5 GB on our 24 GB device to account for the small fluctuations in memory usage that occur during generation and to prevent out-of-memory errors across all methods. Because this is a system-level configuration, we simply kept the same value for all experiments.
>
> *Number of retrieved blocks.* For the number of Top-$N$ blocks used, we fixed $N=2$. Retrieving more than two blocks often introduced unrelated skills, while using only one block lacked sufficient coverage. Since additional blocks can be retrieved later during the replanning (cache refactoring) stage, two initial blocks worked reliably in practice.
>
> *Fill-in-the-middle(FIM) repair: max tokens.* During cache refactoring, which uses FIM to repair faulty code segments, we limited the newly generated tokens to 128 because the modified portions are typically only a few tokens to a couple of lines, and this size consistently covers the required edits. Larger limits increased latency and irrelevant generation, while smaller limits occasionally prevented complete repairs, making 128 the most reliable choice.
>
> Across these settings, our objective was to establish stable and consistent evaluation conditions rather than optimize performance through hyperparameter tuning. We will include these clarifications, along with the corresponding observations, in the updated version.

---

> ### Author Response · Authors · 2025-11-18
> **Response to Question 1**
>
> > **Question 1**: In line 203, why $\alpha$, $\beta$, $\gamma$ needs to sum to one?
>
> We appreciate the detailed question and recognize that this clarification is necessary for Ca$^{2}$P. We provide additional explanation below.
>
> In line with common practice in prior work, weighting coefficients such as $\alpha$, $\beta$, and $\gamma$ are often constrained to sum to one. This ensures a consistent overall scale for the combined term, stabilizes optimization by preventing unintended magnitude changes in the objective, and allows hyperparameter tuning to focus solely on the relative proportions, which improves both interpretability and practical controllability [1, 2]. Following this convention, we enforce $\alpha + \beta + \gamma = 1$ so that the weighted aggregation of the three normalized locality components remains within $[0,1]$ under any weight configuration. This constraint preserves a fixed absolute range for the score while letting us adjust only the relative contribution of each component. To clarify this choice, we further explain below how the components interact to form the overall locality score.
>
> Our management scheme extends conventional cache locality concepts to better suit embodied agents. We reformulate temporal and spatial locality in this context, and additionally introduce a semantic locality component tailored for open-domain environments and function-level key-value caching. Specifically, we define three component scores that capture different aspects of locality: $\ell_{\mathrm{freq}}$ models temporal locality based on usage frequency, $\ell_{\mathrm{asso}}$ captures spatial locality based on conditional association with co-invoked functions, and $\ell_{\mathrm{sema}}$ reflects semantic locality based on language model perplexity.The corresponding component scores are computed as follows:
>
> $$
> \ell_{\mathrm{freq}}(f_k) =
> \frac{\mathrm{count}(f_k)}{\max_j \mathrm{count}(f_j)}
> $$
>
> $$
> \ell_{\mathrm{asso}}(f_j \mid f_k) =
> \frac{\mathrm{cooccur}(f_k, f_j)}{\mathrm{count}(f_k)}
> $$
>
> $$
> \ell_{\mathrm{sema}}(f_k) =
> \frac{\mathrm{PPL}(f_k)}{\max_j \mathrm{PPL}(f_j)}
> $$
>
> Since each component is normalized to $[0,1]$, using the simple constraint $\alpha + \beta + \gamma = 1$ keeps the weighted sum on a fixed scale and lets the score reflect only the relative emphasis among components.
> The weighted locality score is then computed as a weighted sum of the normalized components:
>
> $$
>     \ell(f_k) =
>     (1 - \ell_{\mathrm{curr}}(f_k)) \cdot
>     \left(
>     \alpha \cdot \ell_{\mathrm{freq}}(f_k)
>     + \beta \cdot \sum_j \ell_{\mathrm{asso}}(f_j \mid f_k)
>     + \gamma \cdot \ell_{\mathrm{sema}}(f_k)
>     \right)
>     + \ell_{\mathrm{curr}}(f_k)
> $$
>
> Constraining the weights to sum to one ensures that the final score $\ell(f_k)$ stays within $[0,1]$ and avoids unnecessary variation in magnitude across different weight settings for identical inputs. While the total sum of weights is fixed, their relative ratios can be adjusted depending on the deployment scenario. In this work, we adopt a fixed weight combination that offers stable performance under diverse open-domain conditions, but the framework naturally supports future extensions that adapt the contribution of each locality component according to environment-specific characteristics.
>
> [1] Naz et al., *Multi-Attribute Caching: Towards efficient cache management in Content-Centric Networks.* IEEE CCNC 2016.
>
> [2] Mahni et al., *Multicriteria File-Level Placement Policy for HPC Storage.* ACM SAC 2025.
>
> ---
>
> We have carefully addressed the concerns raised in the weaknesses section and clarified the hyperparameter choices and design as recommended. We greatly appreciate your thoughtful review and hope these updates provide a clearer understanding of our work. If there are any aspects that remain insufficiently addressed or require further clarification, please kindly let us know-we would be more than happy to revise or elaborate further.

---

> ### Author Response · Authors · 2025-11-27
>
> Dear Reviewer NSpR,
>
> Thank you again for your thoughtful review. As the rebuttal period is nearing its end, we provide a brief summary to facilitate a more focused and efficient continuation of the discussion. Below, we summarize what we understand to be your main concerns and how our responses address each point.
>
> ---
>
> ### **Summary of Your Core Concerns**
>
> **1. Lack of citations in the related work section.**
> You noted that the related work section in the main text contained no citations, which prevented the section from fulfilling its purpose of situating our contribution within prior work.
>
> **2. No sensitivity analysis for key hyperparameters ($\alpha$, $\beta$, $\gamma$, etc.).**
> You raised the concern that, given the number of experiments in the paper, it was unexpected that we did not report experiments varying these hyperparameters.
>
> **3. Why $\alpha$ + $\beta$ + $\gamma$ = 1 is required.**
> You asked why the sum of the weighting coefficients is constrained to one, and what purpose this normalization serves.
>
> ---
>
> ### **Summary of Our Responses**
>
> **1. Citations have been added to the main related work section**
> We clarified that detailed citations already exist in Appendix A, but we agree this should have been reflected in the main text. We will revise the related work section to include representative citations and explicit positioning relative to prior work across three categories: LLMs for embodied control, memory-augmented embodied agents, and KV caching and infilling methods. Additionally, we expanded the descriptions for each category in our response to more clearly articulate how these prior lines of work relate to and differ from our contribution.
>
> **2. We conducted and included hyperparameter sensitivity experiments.**
> We added sensitivity results showing that performance varies only modestly across a broad range of $\alpha$, $\beta$, $\gamma$ values. The trends indicate that balanced configurations perform consistently well, motivating our choice to fix ($\alpha$, $\beta$, $\gamma$) without further tuning. Additional rationale behind other fixed hyperparameters (GPU thresholds, retrieval count, FIM token limit) is also provided and will be added to the revised appendix.
>
> **3. The sum-to-one constraint ensures stable scaling and interpretability.**
> We explained that constraining $\alpha$ + $\beta$ + $\gamma$ = 1 keeps the aggregated locality score on a consistent scale, ensures stable optimization behavior, and cleanly represents the _relative_ weighting of the normalized locality components. This is a standard practice in multi-component scoring systems and avoids confounding effects arising from uncontrolled magnitude changes.
>
> ---
>
> ### **Follow-up and Discussion Invitation**
>
> If any part of these summaries remains unclear or if you would like us to elaborate further on the citations, sensitivity results, or the rationale behind the weighting constraints, we would be very grateful for the opportunity to clarify. Even if you join the discussion at a later stage, we will respond immediately and address any remaining questions.

---

### Author Response · Authors · 2025-11-21
**Global Comment: Summary of Revisions in Response to Reviewer Feedback**

# Dear Area Chair, Reviewers and ICLR community,

We sincerely thank you for your thoughtful and highly constructive feedback. Your careful assessments have been invaluable in strengthening the clarity, positioning, and technical rigor of our paper. Below, we summarize the major revisions made in response to the reviewers' comments.

---
## Summary of Revisions
---
### 1. Revisions addressing Reviewer NSpR

Reviewer NSpR noted the need for clearer positioning within the related work and more explicit justification of our locality design and hyperparameter choices.

To address this:
- Explained in our replies is how each method is utilized or extended, and the missing citations are provided in Section 2.
- Expanded the description of the locality components, clarifying how each score is computed and how normalization is applied. The full details are included in Appendix C.2.
- Provided a detailed explanation of the hyperparameter selection process and the rationale for using a normalized weighted combination. The corresponding material is included in Appendix D.8.
---

### 2. Revisions addressing Reviewer qB9j

Reviewer qB9j requested clearer elaboration of the locality formulation and expressed concern about ambiguous terminology across experiments.

To address this:
- Expanded the locality formulation by clarifying the role of each component and explaining the necessity of the normalized weighted combination for consistent retrieval and cache-eviction behavior across task settings; the full details are provided in Appendix C.2 and Appendix D.8.
- Refined the descriptions of the code cache warm-up experiments and clarified that their behavior is independent of the main-phase evaluations. The corresponding updates have been added to Sections 4.1 and 4.3.
- Clarified that the failure-handling mechanism in Algorithm 1 is not simulation-specific by explaining how real-world failures (e.g., grasp failures, pose mismatches, safety interruptions) map into the same structured exception interface used during simulation. This material has been added to Appendix C.2.

---

### 3. Revisions addressing Reviewer r7c6

Reviewer r7c6 questioned whether the try–except structure in Algorithm 1 was overly tailored to simulation and how failures would be detected and handled in real-world robot execution.

To address this:
- Added a unified explanation showing how both simulation-driven and perception-driven failures map into the same structured exception interface.
- Clarified how real-world failures are detected via perception signals (e.g., grasp verification, pose mismatch, safety interruptions), ensuring the same exception-handling path applies consistently across simulated and physical environments.
- The unified design, along with the newly added detection-level table, is documented in Appendix C.2.

---

### 4. Revisions addressing Reviewer cxo5

Reviewer cxo5 requested an additional experiment evaluating efficiency trade-offs introduced by quantization.

To address this:
- Conducted the requested quantization experiment and evaluated its influence on latency, memory usage, and end-to-end task performance.
- Reported how quantization interacts with the cache retrieval mechanism to ensure that reduced numerical precision does not adversely affect policy synthesis.
- The full results and analysis are included in Appendix D.10.

---

We have carefully revised the paper to incorporate all reviewer suggestions while preserving the integrity and intent of the original contribution.
If any part of our revisions remains unclear or would benefit from further elaboration, please let us know - we would be happy to clarify or revise further.

**Sincerely,
The Authors**

---

### Author Response · Authors · 2025-11-30
**Additional Clarifications for Remaining Reviewer cxo5's Concern**

**Dear Area Chair and Reviewer cxo5,**

We would like to provide additional clarifications, including supporting experiments, to assist with the evaluation of the remaining concern raised by Reviewer cxo5. To this end, the following comment more explicitly articulates the concern and presents the concrete evidence requested by the reviewer.

---

## **1. Clarifying what “existing toolbox caching” means**

Reviewer cxo5 referenced _existing toolbox methods such as token caching_, so we first clarify what “caching” typically means in this context.

In the broader systems and LLM literature, _token caching_ usually refers to reusing previously produced tokens or code fragments **to avoid re-decoding or re-generating identical sequences**.
Representative examples include prefix caching, text-fragment reuse, template caching, or retrieval of previously validated snippets without regenerating them [1, 2].

It is worth noting that the “token caching’’ used in many toolbox systems reduces decoding by reusing existing text segments, not by storing key–value attention states.[3, 4] As such, this form of caching is independent of KV mechanisms and functions as a  text-level reuse strategy.
Thus, when Reviewer cxo5 mentioned “token caching,” it is reasonable to interpret it as:

> _“reusing pre-generated code fragments without regenerating them from scratch.”_

We explicitly adopt this broader interpretation here.

---

## **2. Why our diagnostic experiment directly addresses this concern**

To show how such general (non-KV) caching interacts with our method,
we conducted a diagnostic experiment comparing:

- **Non-cache:**
    Every line of code is generated from scratch.
    No prefix reuse, no snippet recall, no function reuse.
- **Cache:**
    Previously validated code fragments are reused directly,
    avoiding regeneration of those fragments - representing general token/text caching.
- **Program-level memory:**
    Code is stored and reused as a single monolithic block.
- **Function-level memory:**
    Code is decomposed into reusable function modules that can be recomposed.

The results (Real-world) are:

|**Real-world**|_Program-level_||||_Function-level_||||
|---|---|---|---|---|---|---|---|---|
|**Methods**|**SR (↑)**|**GC (↑)**|**PSL (↓)**|**HR (↑)**|**SR (↑)**|**GC (↑)**|**PSL (↓)**|**HR (↑)**|
|Non-Cache|44.44|53.33|8.04|74.50|77.78|77.78|6.21|87.25|
|Cache|66.67|71.11|4.52|74.50|88.89|88.89|3.80|87.25|

### **Interpretation relevant to Reviewer cxo5:**

1. **General caching (token/text-level) truly helps**
    High-level caching alone improves efficiency and stability - showing our framework is compatible with the caching style the reviewer referred to.
2. **Our contribution lies elsewhere**
    Even without caching, function-level memory outperforms program-level memory.
    → meaning the key novelty is _memory structure_, not a specific caching mechanism.

This explains why our work is not “just cache management” and why existing toolbox methods (token caching, text caching, prefix reuse) do not target the structural, long-horizon memory demands of open-domain embodied control.

---

## **3. Connection to quantization (as the reviewer also mentioned toolbox techniques)**

We also include quantization experiments (Appendix D.10).
These experiments confirm that our framework can be used together with toolbox optimizations such as:

- token/prefix caching
- quantization
- lightweight text-fragment caching
- KV-based acceleration

and that these methods do not conflict, but instead produce additive improvements.

Thus, our framework extends beyond, but remains compatible with, existing toolbox efficiency techniques - exactly addressing the reviewer’s concern.

---

If further clarification would assist the committee, we would be happy to provide it immediately.

Sincerely,
**The Authors**

---

[1] Gim et al., _Prompt Cache: Modular Attention Reuse for Low-latency Inference._ MLSys 2024.

[2] Lu et al., *TurboRAG: Accelerating Retrieval-Augmented Generation with Precomputed KV Caches for Chunked Text.* EMNLP 2025.

[3] Kwon et al., _Efficient Memory Management for Large Language Model Serving with PagedAttention._ SOSP 2023.

[4] Arenas et al., _How to Prompt Your Robot: A Promptbook for Manipulation Skills with Code as Policies._ ICRA 2024.

---

### Author Response · Authors · 2025-11-30
**Summary of Discussion 2**

## **3. Reviewer r7c6 - Real-world Failure Detection**

**Concern:**
The try–except structure in Algorithm 1 appeared tailored to simulation, raising concerns about real-world applicability.

**Our response:**

- We now describe a unified exception interface applicable to both simulation and physical execution, distinguishing interpreter-level and environment-level (RobotError) exceptions.
- A perception-driven failure detection pipeline-using RGB-D sensing, object detection, and VLM-based state verification-has been documented in **Appendix B.4**.
- A consolidated exception taxonomy and unified handling pathway are provided in **Appendix C.2**.

**Discussion period:**
The reviewer provided an overall positive assessment while noting that additional clarification was needed for real-world deployment. Accordingly, we clarified these points by presenting concrete exception types and illustrative examples. Because we had not yet received further questions by November 27, we proactively provided a structured summary of the review–rebuttal points in order to facilitate an efficient and focused continuation of the discussion. Our goal was to ensure that every concern was fully resolved within the discussion period.

---

## **4. Reviewer cxo5 - Contribution Scope and Comparison to Prior Approaches**

**Concerns:**
The contribution appeared limited to cache management, and distinctions from existing toolbox methods and CaP’s built-in caching were not sufficiently clear.

**Our response:**

- We clarified that our work addresses a fundamentally different problem setting: **open-domain embodied control**, which requires persistent, editable, function-level memory across long, unpredictable task streams. This differs markedly from the static or short-horizon text-generation scenarios targeted by existing toolbox optimizations.
- We expanded comparative analysis against toolbox-style KV caching (PromptCache, RAGCache, CAG), per-episode CaP caching, and memory-based code agents (HELPER, LRLL, PromptBook).
- We strengthened empirical evidence across ALFRED, TEACh, and RLBench to demonstrate that our framework offers consistent improvements in both success rate and policy synthesis latency.
- Additional quantization experiments were conducted (reported in **Appendix D.10**), confirming that our approach is fully complementary to such optimizations.

**Discussion period:**
Although the reviewer initially expressed a negative evaluation, the detailed scores (Soundness, Presentation, Contribution), along with the accompanying comments, indicated that the reviewer’s concerns stemmed from a partial misunderstanding of our contribution. For this reason, we continued preparing additional clarifications even after our initial rebuttal response. We plan to provide further clarifications on these points and will update them as soon as they are fully prepared.

Sincerely,
**The Authors**

---

### Author Response · Authors · 2025-12-03
**Summary of Discussion 1**

# Dear Area Chair,

We are truly sorry to hear about the recent OpenReview data leakage and sincerely hope that the issue will be resolved smoothly and without further difficulty.

We would also like to express our gratitude for your oversight throughout the review process. To support a clear and efficient final assessment, we respectfully provide below a concise and formal summary of the primary concerns raised by the reviewers, along with our responses addressing each of them.

---

## **1. Reviewer NSpR - Related Work and Locality Design**

**Concern:**
The related work in the main text lacked citations, and the locality formulation and hyperparameter choices were insufficiently supported.

**Our response:**

- We have added representative citations and clarified the positioning of our work within three major lines of literature in **Section 2**, with expanded descriptions now included.
- Precise mathematical definitions of all locality components and the rationale for normalized weighting are provided in **Appendix C.2**.
- A comprehensive hyperparameter sensitivity analysis has been added in **Appendix D.8**, showing the method’s stability across a broad range of weight configurations.

**Discussion period:**
Although the reviewer’s initial assessment was positive overall, we prepared and shared a concise summary of the key review–rebuttal points in order to facilitate an efficient and focused continuation of the discussion. Our intention was to ensure that all concerns were addressed thoroughly and transparently within the available discussion period.

---

## **2. Reviewer qB9j - Locality Specification, Warm-up Interpretation, and Open-domain Framing**

**Concerns:**
Locality score definitions were high-level; the relationship between warm-up and pre-populated cache experiments was unclear; the “open-domain” framing needed clarification; and further explanation was requested for Algorithm 1 and real-world experiments.

**Our response:**

- The locality score is now defined rigorously, with component-wise ablations included in **Appendix C.2**.
- We clarified the distinction between the warm-up experiment (empty cache, Fig. 4) and the pre-populated evaluation setting (Table 1), as detailed in **Appendix D.1** and **D.3**.
- Additional experiments demonstrating API-level extensions (low → high-level primitives, RLBench → real-robot transfer) have been added to justify our usage of the open-domain terminology.
- Detailed explanations of generate(), edit(), update(), exception handling, and perception-driven failure detection are included in **Appendix B.4** and **C.2**.

**Discussion period:**
The reviewer initially assigned a score of 4, placing the evaluation on the slightly negative side; however, during the discussion period, the score was raised to 6. The reviewer provided substantial and detailed feedback, demonstrating strong interest in the work. We made a concerted effort to address every concern without omission, and the reviewer was the first to re-engage once the discussion opened. On November 26th, the reviewer indicated that their concerns had been satisfactorily resolved and subsequently updated the score.

---

### Author Response · Authors · 2025-12-04
**Author Final Remarks**

We conclude the Author-Reviewer discussion period with a summary of the reviews and discussions, and sincerely thank the ICLR community for their time and constructive engagement.

## **Strengths**

- Clear contribution addressing a knowledge gap in code-as-policies — mentioned by `Reviewer NSpR, Reviewer qB9j, Reviewer r7c6`.
- Well-structured technical design (two-tier cache: Function-Interface & Function-Code) — mentioned by `Reviewer NSpR, Reviewer qB9j, Reviewer r7c6`.
- Extensive empirical evaluation across multiple environments and domains — mentioned by `Reviewer NSpR, Reviewer qB9j, Reviewer r7c6, Reviewer cxo5`.
- Clear writing and informative presentation — mentioned by `Reviewer r7c6, Reviewer cxo5`.

## **Raised Concerns and Our Responses**

- Locality score definitions — mentioned by `Reviewer NSpR, Reviewer qB9j`.
  - We provided precise mathematical definitions of all locality components with the rationale for normalized weighting and added component-wise ablations (Appendix C.2, D.8).

- Algorithm 1 clarity & real-world execution details — mentioned by `Reviewer qB9j, Reviewer r7c6`.
  - We added clearer explanations of the framework’s execution logic and real-world execution procedures, including the exception interface and perception-driven failure handling, and provided additional API-level extensions and real-world transfer experiments (RLBench → real-robot) to support the approach’s general applicability (Appendix B.4, C.2).

- Open-domain framing & differentiation from prior work — mentioned by `Reviewer NSpR, Reviewer qB9j, Reviewer cxo5`.
  - We clarified that the work targets open-domain embodied control and its scope by adding representative citations and positioning the work within existing literature (Section 2), added experiments demonstrating API-level extensions and real-world transfer, and expanded comparative analysis against toolbox methods (token caching, quantization), KV caching, and other related approaches.

## **Discussion Points**

- Detailed and comprehensive clarifications with additional experiments — mentioned by `Reviewer qB9j`, who raised the score from 4 to 6.

- A summary and added clarifications for efficient discussion — provided to `Reviewer NSpR, Reviewer r7c6, Reviewer cxo5`.

- A declaration of no involvement in or exploitation of the recent OpenReview data leakage.

---

We appreciate the constructive feedback, which has helped improve the clarity and contribution of our work, and we have made every effort to adequately address all remaining concerns raised by the reviewers. We also sincerely thank the Area Chair for their time, guidance, and effort throughout the review process.

Sincerely,
**The Authors**

---

### Meta-Review · Area_Chair_8z4P · 2025-12-30

**Summary:**

The core contribution is limited and largely centers on cache management for code generation, an area that has been widely explored in both research and deployed systems. Despite the authors’ rebuttals and clarifications, the novelty beyond existing toolbox methods and previous caching mechanisms remains incremental. As a result, I do not believe the work meets the bar for a substantial contribution to ICLR, and I recommend rejection.

**Reviewer Concerns:**

Reviewer NSpR was concerned about insufficient analysis of hyperparameters, while Reviewer qB9j raised issues regarding the local score, implicit inconsistencies in experiments, and an overstated open-domain framing. Reviewer r7c6 has low confidence in this paper and only questioned the real-world deployment, and Reviewer cxo5 expressed fundamental concerns about the contribution being incremental and overlapping with existing toolbox caching approaches.

**Reviewer Scores:**

Since most of the concerns of the negative reviewers (cxo5 and qB9j) are not well addressed yet, I believe all the reviewers would possibly remain at the same score.

---

### Decision · Program_Chairs · 2026-01-26

Reject